# Sustainable irrigation based on co-regulation of soil water supply and atmospheric evaporative demand

Jingwen Zhang [1,2 ✉], Kaiyu Guan [1,2,3 ✉], Bin Peng [1,2,3 ✉], Ming Pan[4,5], Wang Zhou[1,2], Chongya Jiang[1,2], Hyungsuk Kimm[1,2], Trenton E. Franz[6], Robert F. Grant [7], Yi Yang[1,2], Daran R. Rudnick[8], Derek M. Heeren[8], Andrew E. Suyker[6], William L. Bauerle[9] & Grace L. Miner[10]

Irrigation is an important adaptation to reduce crop yield loss due to water stress from both soil water deficit (low soil moisture) and atmospheric aridity (high vapor pressure deficit, VPD). Traditionally, irrigation has primarily focused on soil water deficit. Observational evidence demonstrates that stomatal conductance is co-regulated by soil moisture and VPD from water supply and demand aspects. Here we use a validated hydraulically-driven ecosystem model to reproduce the co-regulation pattern. Specifically, we propose a plant-centric irrigation scheme considering water supply-demand dynamics (SDD), and compare it with soil-moisture-based irrigation scheme (management allowable depletion, MAD) for continuous maize cropping systems in Nebraska, United States. We find that, under current climate conditions, the plant-centric SDD irrigation scheme combining soil moisture and VPD, could significantly reduce irrigation water use (−24.0%) while maintaining crop yields, and increase economic profits (+11.2%) and irrigation water productivity (+25.2%) compared with MAD, thus SDD could significantly improve water sustainability.

[1] Agroecosystem Sustainability Center, Institute for Sustainability, Energy, and Environment, University of Illinois at Urbana Champaign, Urbana, IL, USA. [2] College of Agricultural, Consumer and Environmental Sciences, University of Illinois at Urbana Champaign, Urbana, IL, USA. [3] National Center for Supercomputing Applications, University of Illinois at Urbana Champaign, Urbana, IL, USA. [4] Department of Civil and Environmental Engineering, Princeton University, Princeton, NJ, USA. [5] Center for Western Weather and Water Extremes, Scripps Institution of Oceanography, University of California San Diego, La Jolla, CA, USA. [6] School of Natural Resources, University of Nebraska-Lincoln, Lincoln, NE, USA. [7] Department of Renewable Resources, University of Alberta, Edmonton, Alberta, Canada. [8] Department of Biological Systems Engineering, University of Nebraska-Lincoln, Lincoln, NE, USA. [9] Department of Horticulture and Landscape Architecture, Colorado State University, Fort Collins, CO, USA. [10] Soil Management and Sugarbeet Research Unit, USDA-ARS, Fort Collins, CO, USA. ✉email: jingwenz@illinois.edu; kaiyug@illinois.edu; binpeng@illinois.edu

rrigated agriculture accounts for ~72% of the total water withdrawals from surface water and groundwater globally, while contributing 40% of total food production[1–3]. At the same time, agricultural irrigation has led to severe water scarcity issues at regional to global scales due to the expansion of irrigated areas and increase of irrigation amount[4,5]. For example, the U.S. used an estimated 162.8 km³ of water to irrigate 25.7 million ha in 2015, accounting for 42% of total freshwater withdrawals[6]. Increasing areas of croplands with intensified irrigation have caused groundwater depletion in High Plains, Central Valley, and Mississippi Embayment aquifers in the U.S., which highlights the urgency of more sustainable water use for irrigation, especially, under climate change[7–9].

Understanding plant water relations[10] is the prerequisite for sustainable water use for irrigation (Fig. 1). Plant growth is regulated by the balance of water supply and demand in the soil-plant-atmosphere-continuum (SPAC). Water supply is represented by available water in the soil for plant uptake, while water demand is controlled by atmospheric aridity that passively drives water to move from plants into the atmosphere[11–13]. The atmospheric aridity is quantified by vapor pressure deficit (VPD), i.e., the difference between the saturated and actual vapor pressures at a given air temperature. Current irrigation practices have primarily focused on the soil water supply side, though we acknowledge that to calculate soil water balance to get soil moisture, evapotranspiration (ET) based on different methods is usually used in most methods[14]. Here we argue that the plant-centric irrigation schemes are crucial for sustainable irrigation based on the interplay between soil water supply and atmospheric evaporative demand via plant physiological regulations[15] (i.e., plant hydraulics and stomatal response).

Soil water deficit and high VPD both can reduce terrestrial ecosystem productivity[16–23]. In the SPAC, low soil moisture and high VPD can lead to plant water stress which drives plants to close their stomata to prevent excessive water loss[11,12,18,23–25] (Fig. 1a). At the same time, reduced stomatal conductance, reflecting the physiological regulation of the uptake of atmospheric carbon dioxide for photosynthesis, and water loss through transpiration[11,20,26,27] also limits carbon assimilation and increases the risks of crop yield loss[13]. A long-term increasing

trend of VPD[16,28] and high probability of concurrent soil water deficit with atmospheric aridity[21,29] have been projected globally under the climate change, further underscoring the need of including the physiological impact of high VPD in irrigation management.

This study has three objectives: (1) to investigate the co-regulation of the soil moisture and VPD on stomatal conductance of maize using field measurements and a validated process-based ecosystem model; (2) to propose a plant-centric irrigation scheme for sustainable irrigation based on the co-regulation pattern; (3) to test and compare the plant-centric irrigation scheme with soil moisture-based management allowable depletion (MAD) irrigation scheme under current climate and the representative concentration pathway 8.5 (RCP-8.5) scenario. The innovation of this study is to apply the co-regulation pattern into irrigation management, and we find the proposed method has demonstrated a large improvement over the existing soil moisture-only irrigation metrics and thus could have significant contributions to water sustainability.

## Results

**The co-regulation of soil moisture and VPD on stomatal conductance.** Stomatal conductance can be treated as one of the most effective metrics to quantify plant water stress considering both soil water supply (i.e., soil moisture) and atmospheric evaporative demand (i.e., VPD). Figure 2 showed the co-regulation pattern of soil moisture and VPD on stomatal conductance of maize based on observations (including those from greenhouse experiments and eddy-covariance sites) and process-based modeling under different climate conditions. Based on the contour fitted using a statistical model (see Methods), the whole regime can be classified into the co-regulated regime (i.e., inclined contours) and the VPD-dominated regime (i.e., horizontal contours). The greenhouse measurements of maize indicated that stomatal conductance increased with soil moisture and decreased with VPD in the co-regulated regime (large gradient of stomatal conductance with soil moisture and VPD, Fig. S1), while it was mainly driven by VPD in the VPD-dominated regime (Fig. 2a, b). The co-regulation of soil moisture and VPD on stomatal conductance was further confirmed with eddy-covariance measurements (Fig. 2c, d). Stomatal conductance was higher under higher soil moisture (more water supply) and/or lower VPD (less water demand). All these observed patterns could be reproduced by a validated hydraulically driven ecosystem model (ecosys) under maize cropping systems across 12 sites in Nebraska (an example site-GD in Fig. 2e, f, and Fig. S2) (see Methods). The co-regulation pattern indicated that plants can have water stress even at high soil moisture but under high VPD conditions. In contrast, plants may not have water stress when soil moisture was relatively low and VPD also happened to be low.

The differences in the size and shape of the co-regulation regimes mainly resulted from varying climate conditions and soil properties (Fig. 2 and S2). In addition, different cultivars of maize and approaches to obtain stomatal conductance may also cause the differences in Fig. 2. Specifically, the impacts of soil and climate properties on the co-regulation regimes were investigated using ecosys and a relative importance method[30] across 12 sites in Nebraska with a large rainfall gradient (Fig. 3). We further quantified the relative contributions of soil moisture and VPD to the variations of ecosys-simulated stomatal conductance at the daily scale. All the Spearman partial rank correlation coefficients between stomatal conductance and soil moisture/VPD across 12 sites were significant ($p < 0.001$), demonstrating the significant controls from soil moisture and VPD on stomatal conductance. The estimated relative importance metrics across 12 sites

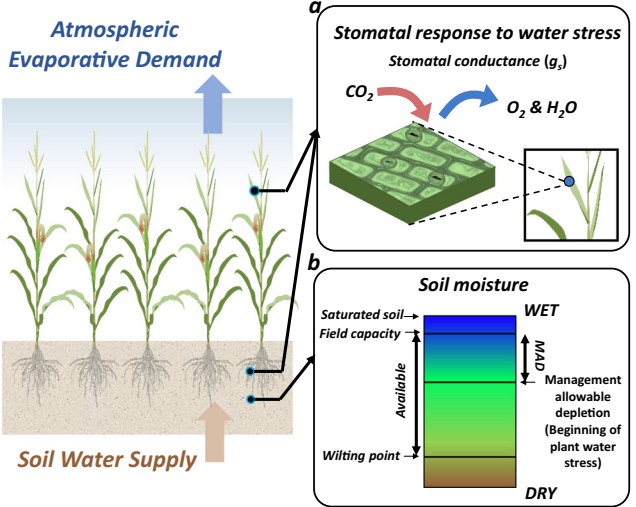

**Fig. 1 Conceptual schemes of two methods to quantify plant water stress. a** Plant-centric: regulating stomatal conductance ($g_s$) by considering both soil water supply and atmospheric evaporative demand. **b** Soil-moisture-based: management allowable depletion (MAD) solely considering soil water supply[15].

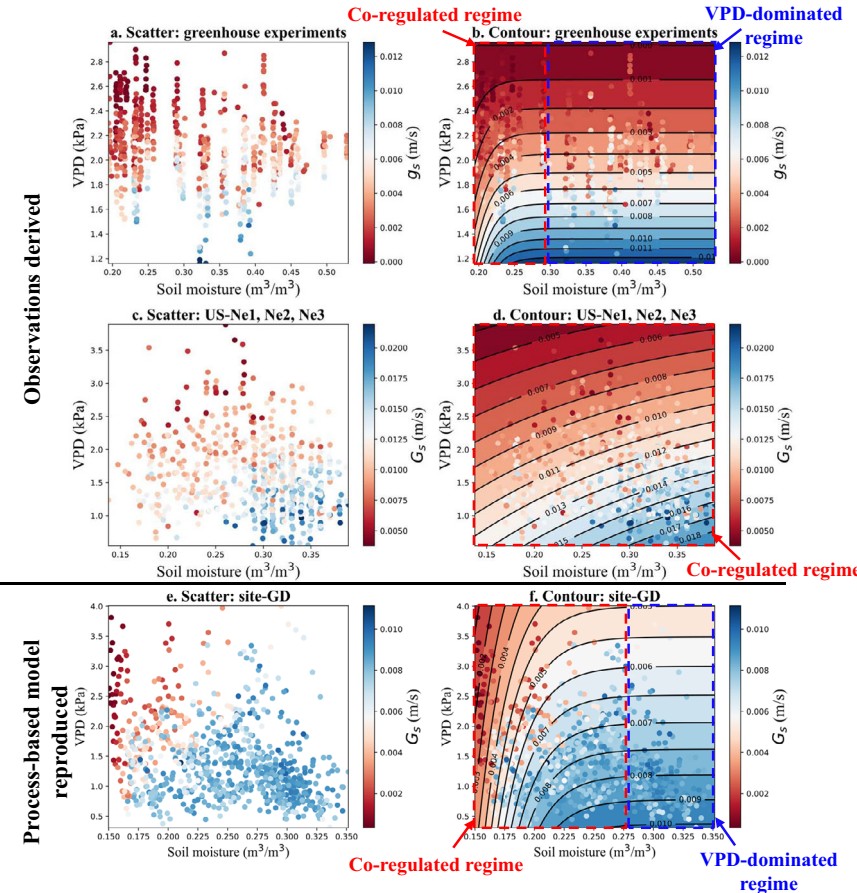

**Fig. 2 The co-regulation pattern of soil moisture and VPD on stomatal conductance of maize at the daily scale. a, b** Measurements of soil moisture, VPD, and leaf-level stomatal conductance ($g_s$) of maize from greenhouse experiments. **c, d** Measurements of soil moisture, VPD, and derived canopy-level stomatal conductance ($G_s$) using Penman-Monteith equation in the peak growing season (July and August) from 2001 to 2012 at three eddy covariance sites (US-Ne1, US-Ne2, and US-Ne3) under maize cropping systems. **e, f** Simulations of soil moisture, VPD, and $G_s$ from the *ecosys* model in the peak growing season (July and August) from 2001 to 2019 at site-GD in Nebraska under continuous maize cropping systems. The contours shown in c, d and f are fitted using a statistical model (see Methods) with $R^2$ of 0.64, 0.54, and 0.62, respectively.

indicated that the contributions of soil moisture to stomatal conductance significantly increased with the aridity index ($p < 0.001$) as there were more limitations from soil water supply to stomatal responses in drier regions (Fig. 3b and S4). The significant positive relationship between the Spearman partial rank correlation coefficients ($G_s$-Soil moisture) and aridity index ($p < 0.0001$) further confirmed this (Fig. S3a). In contrast, the contributions of VPD to stomatal conductance significantly decreased with the increasing aridity index ($p < 0.0001$) (Fig. 3b and S4), leading to no VPD-dominated regimes at site-T1S1, site-EastBayard, and site-Mitchell (Fig. S2j–l). For example, under the current climate (2001–2019), soil moisture dominated the stomatal conductance variations (88.3%) at an extreme dry and sandy site-T1S1 (aridity index: 2.22 and sand fraction: 77%); while soil moisture and VPD made comparable contributions (50.1 and 49.9%) to the stomatal conductance variations at a wetter site-Mead (aridity index: 1.38 and sand fraction: 10%) (Fig. 3). These results indicated that VPD had a non-negligible impact on stomatal conductance, especially in wetter regions. The contributions of soil moisture usually exceeded those of VPD to stomatal conductance, with an exception at site-Lowell (more contributions from VPD than soil moisture) which was sandy but got relatively higher annual rainfall. Compared with current climate conditions, the contributions of soil moisture to stomatal conductance at 12 sites increased under the RCP-8.5 scenario, as precipitation distribution was more biased

towards nongrowing season (Fig. S5), leading to more stomatal limitations from soil water supply during the growing season (Fig. 3b).

**A plant-centric irrigation scheme based on water supply-demand dynamics.** We proposed a plant-centric irrigation scheme based on water supply-demand dynamics (SDD), i.e., the co-regulation pattern of soil moisture and VPD on stomatal conductance. Stomatal conductance increased with soil moisture due to the limitations from water supply until reaching a critical stomatal conductance under specified VPD conditions. The soil moisture corresponding to the critical stomatal conductance was treated as the transition point of supply and demand limitations[18,31] (Fig. 4b, c, d). Taking site-GD in Nebraska as an example, the critical stomatal conductance was determined to be 0.007 m s$^{-1}$ (see Methods and Table S2). The first part of the SDD soil moisture thresholds, varying with VPD, were determined using the fitted contours of the critical stomatal conductance in the co-regulated regime until entering the VPD-dominated regime, i.e., the blue contour of critical $G_s = 0.007$ m s$^{-1}$ (part 1 in Fig. 4a). Stomatal conductance was dominated by VPD in the VPD-dominated regime, as soil moisture had little effect on stomatal conductance. Thus, the other part of the SDD soil moisture threshold was determined as the boundary of the VPD-dominated regime, i.e. the constant blue MAD threshold

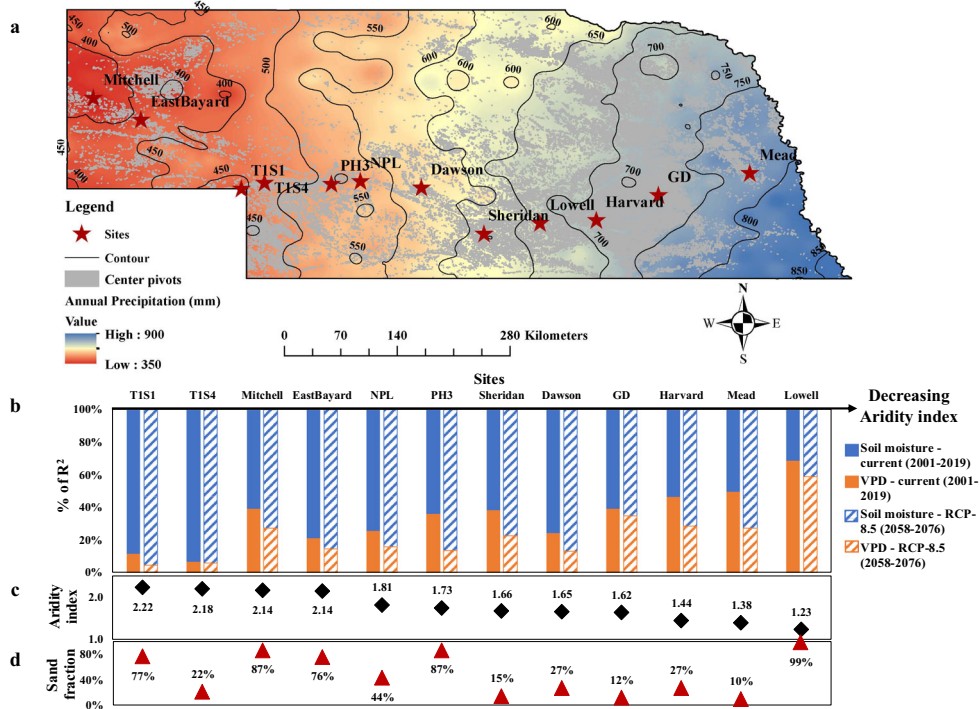

**Fig. 3 Spatial variability of the co-regulation pattern of soil moisture and VPD on stomatal conductance of maize in Nebraska, United States. a** Location of 12 sites with a large rainfall gradient (from 800 mm in the east to 400 mm in the west) in Nebraska. **b** The relative importance metrics of soil moisture and VPD as drivers of variability in stomatal conductance (from *ecosys*) across 12 sites in Nebraska under current climate (2001-2019) and RCP-8.5 scenario (2058- 2076). **c, d** Variation of aridity index and sand fraction across 12 sites in Nebraska. Aridity index was calculated as the ratio of potential evapotranspiration[31] (PET, mm) and precipitation (P, mm) during growing season from 2001 to 2019, i.e. PET/P.

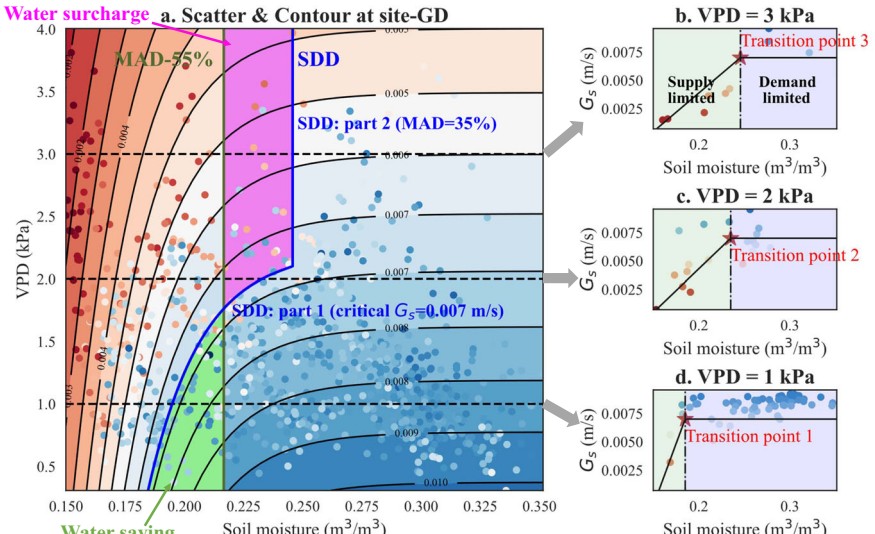

**Fig. 4 Schematic diagram of plant-centric irrigation scheme based on water supply-demand dynamics at an example site- GD. SDD denotes the proposed plant-centric irrigation scheme based on water supply-demand dynamics (SDD). a** Scatters of soil moisture, VPD, and canopy-level stomatal conductance ($G_s$) at site-GD were simulations from the *ecosys* model without irrigation impacts to reproduce the co-regulation pattern. The SDD soil moisture thresholds (blue curve including part 1, i.e. contour of critical $G_s = 0.007$ m/s, and part 2, i.e. the constant blue MAD threshold with 35%), which were the transition points between water supply (light green region) and demand (light blue region) limitations, varied with VPD. **b** the scatters of soil moisture and $G_s$ when VPD equaled 3 kPa with the transition point 3; **c** the scatters of soil moisture and $G_s$ when VPD equaled 2 kPa with the transition point 2; and **d** the scatters of soil moisture and $G_s$ when VPD equaled 1 kPa with the transition point 1. MAD-55% (olive line), performed as the benchmark, denoted the traditional soilmoisture-based management allowable depletion (MAD) irrigation scheme with constant soil moisture threshold (45% of soil available water holding capacity).

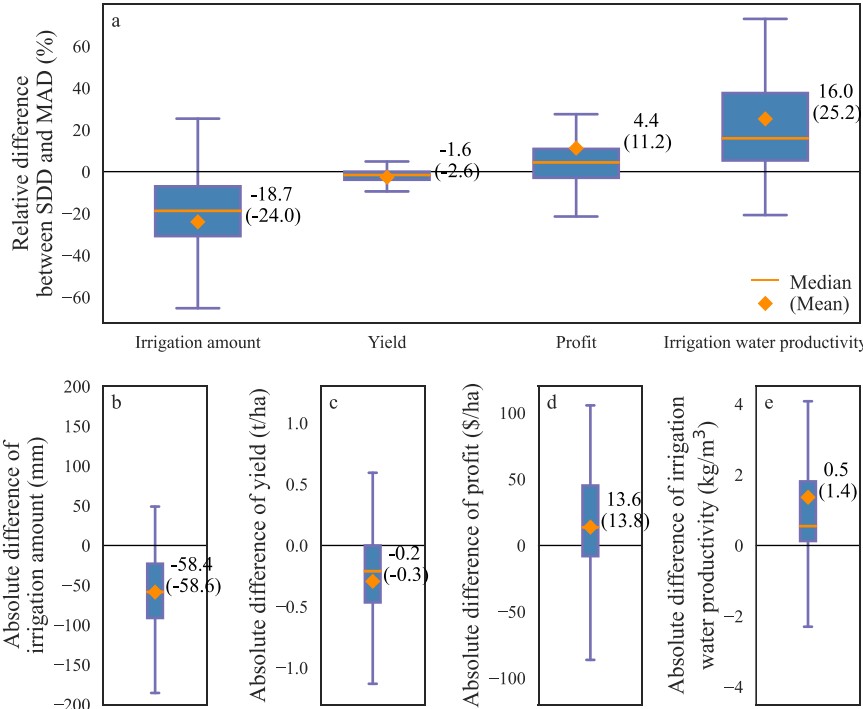

**Fig. 5 Performances of plant-centric SDD and soil-moisture-based MAD irrigation schemes across 12 sites in Nebraska under current climate (2001-2019). a** The box plots of relative differences (see Methods) in irrigation amount, yield, profit, and irrigation water productivity (ratio between yield and irrigation) between SDD and MAD across 228 site-years. **b, c, d** and **e** The box plots of absolute differences in irrigation amount, yield, profit, and irrigation water productivity between SDD and MAD across 228 site-years. SDD and MAD irrigation schemes both used universal parameters across 12 sites in Nebraska. Boxes showed 25th–75th percentiles, and the orange line and diamond denoted the median and mean for each box, respectively.

with 35% (part 2 in Fig. 4a). Consequently, irrigation was triggered when soil moisture got lower than the SDD soil moisture threshold under specified VPD conditions. The innovation of the SDD irrigation scheme was the dynamic soil moisture threshold based on the co-regulation of soil moisture and VPD on stomatal conductance. We compared its performance with a simple conventional soil moisture-based irrigation scheme (MAD-55%, olive line in Fig. 4a). All parameters of the SDD (i.e., critical stomatal conductance under the low VPD conditions and the soil moisture threshold under high VPD conditions) and MAD (i.e., soil moisture threshold) irrigation schemes were optimized to be either site-specific or universal for all the sites using simulated profit, which took total irrigation cost and economic gain from crop yields into account (see Methods, Fig. S6 and Table S2, S3).

SDD differed from MAD in two zones, i.e., the "water-saving" zone (green area in Fig. 4a) and the "water surcharge" zone (magenta area in Fig. 4a). More specifically, irrigation can be delayed under low VPD conditions in the "water-saving" zone as less water supply was required from the soil moisture to meet low atmospheric water demand. Conversely, irrigation was more easily triggered to prevent water stress arising from high atmospheric water demand in the "water surcharge" zone. The net effect on irrigation water use and crop production of SDD was determined by the relative occurrence frequency of environmental conditions in the "water-saving" and "water surcharge" zones. If the "water-saving" occurred more frequently than the "water surcharge", the SDD irrigation scheme contributed to water sustainability. On the contrary, if the "water surcharge" happened more often than the "water saving", the SDD irrigation scheme required more irrigation water use than the MAD scheme but also ensured no water stress under high VPD conditions.

**Contributions to water sustainability**. We systematically simulated application and outcomes of the SDD and MAD irrigation schemes at 12 sites in Nebraska under current (2001–2019) and future (RCP-8.5, 2058–2076) climate conditions. Under current climate condition, SDD with universal parameters (same set of parameters across 228 site-years) could significantly reduce irrigation water use ($-24.0\%$, $-58.6$ mm), maintain crop yields (marginal differences), and thus increase economic profits ($+11.2\%$, $+\$13.8$ ha$^{-1}$) and irrigation water productivity ($+25.2\%$, $+1.4$ kg m$^{-3}$) compared with MAD (Fig. 5 and S7). If site-specific parameters were applied, SDD could still make large contributions to save irrigation water use ($-17.8\%$, $-39.1$ mm), increase economic profits ($+5.7\%$, $+\$23.1$ ha$^{-1}$) and irrigation water productivity ($+21.4\%$, $+1.2$ kg m$^{-3}$) without penalizing crop yields (Figs. S8 and S9). It is worth noting that MAD is already a highly efficient method for irrigation. In reality strictly enforcing MAD is hard to achieve due to the need for soil moisture information from either sensors or sophisticated modeling, and most practical solutions (such as based on rainfall or ET) have much lower efficiency than MAD. Thus the significant benefit of SDD over MAD demonstrated here provided a testimony for the improved performance of SDD. The benefits of the SDD irrigation scheme over MAD varied with climate conditions (e.g. aridity index) and soil properties (e.g., sand fraction) (Fig. S10). The absolute differences in irrigation water use and irrigation water productivity between the SDD and MAD irrigation schemes significantly decreased with increasing aridity index ($p < 0.05$) and also slightly decreased with increasing sand fraction; while the difference in crop yields between the SDD and MAD maintained stable and negligible with aridity index and sand fraction. It indicated that the SDD irrigation scheme made larger contributions to water sustainability in wetter and/or less

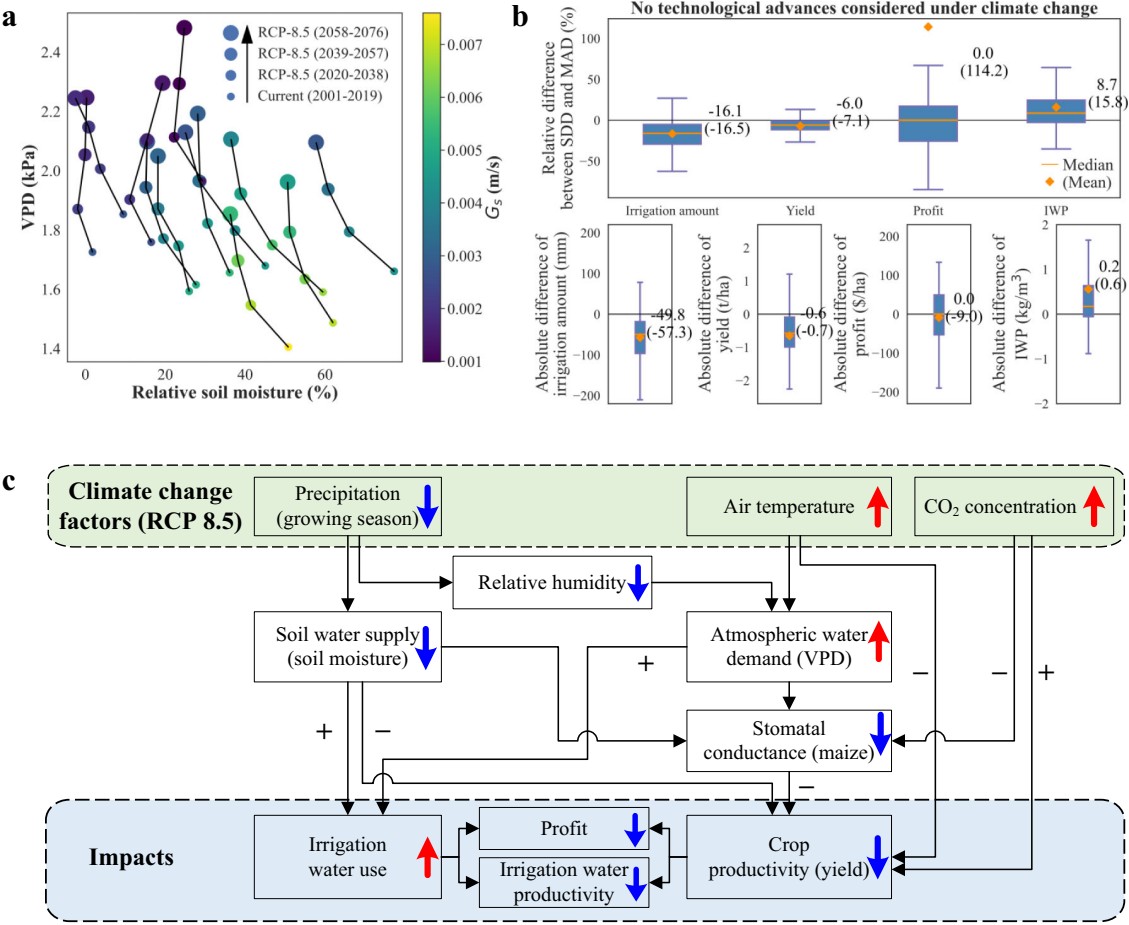

**Fig. 6 The impacts of climate change on irrigation water use and crop productivity. a** The simulated shifts of soil water supply (soil moisture), atmospheric evaporative demand (VPD), and stomatal conductance ($G_s$) without irrigation under current climate (2001–2019) and over three continuous time periods (2020–2038, 2039–2057, and 2058–2076) under RCP-8.5 scenario. Each point showed the averages of daily soil moisture, VPD, and $G_s$ during the peak growing season (July and August) in each scenario at 12 sites in Nebraska. Relative soil moisture, the ratio (in percentage) of available water to field capacity, was applied to avoid the impact of heterogeneous soil properties at 12 sites. **b** The performances of the plant-centric SDD and soil-moisture-based MAD irrigation schemes with universal parameters under the RCP-8.5 scenario (2058–2076) (similar to Fig. 5) without considering technological advances on crop yields. **c** The schematic diagram of the main processes by which climate change affects irrigation water use and crop productivity.

sandy regions than drier and/or more sandy regions. One of the reasons was that VPD in wetter regions was relatively lower than those in drier regions (Fig. S5), thus "water saving" occurred more frequently in wetter regions, resulting in more contributions to water sustainability. On the other hand, VPD had a larger impact on stomatal conductance in wetter regions than that in drier regions (Fig. 3b and S4a). Thus, we concluded that the SDD irrigation scheme could contribute more to water sustainability and economic profits at regions with lower VPD and/or larger constraints from VPD on agricultural drought.

Under climate change conditions (RCP-8.5, 2058–2076), more concurrent soil water deficit and atmospheric dryness led to decreased stomatal conductance (Fig. 6a). If we do not consider the technological advances[32], such as seeds, irrigation, and fertilizer improvements, on crop yields, i.e., no technology yield trend, irrigation water use of MAD irrigation scheme under RCP-8.5 scenario (2058–2076) increased by 16.1%, accompanied with significant reductions in crop yield (−24.5%), economic profit (−54.1%), and irrigation water productivity (−29.2%), when compared with the current climate condition (2001–2019) (Fig. S11). This indicated that more irrigation water use was needed to relieve more severe soil water deficit under future climate conditions. However, more intensive irrigation still

cannot resolve the yield loss from rising air temperature (ca. 4 °C temperature elevation during peak growing season in Nebraska) and VPD (Fig. S5), and decreased stomatal conductance under climate change, even after considering the benefits of elevated [$CO_2$] for net carbon assimilation of maize under water-limited conditions[33] (Fig. 6c). Compared with MAD, SDD could still significantly reduce irrigation water use (−16.5%, −57.3 mm) and increase irrigation water productivity (+15.8%, 0.6 kg m$^{-3}$), while it made negligible contributions to economic profits under future climate conditions (Fig. 6b). These results demonstrated that the SDD irrigation scheme could lead to more sustainable irrigation water use under future climate conditions.

## Discussion
Our study proposed and implemented the plant-centric SDD irrigation scheme based on the plant water supply-demand dynamics, i.e., the co-regulations of soil moisture and VPD on stomatal conductance. This co-regulation mechanism has been widely demonstrated by observational evidence[11,12,18,24,25] but has seldom been applied in irrigation management directly[34]. Although some developed irrigation practices were plant-based

and/or plant-soil hybrid[34,35], such as evapotranspiration (ET)-based[36,37] and canopy temperature-based[34,38] irrigation, the plant-centric SDD irrigation scheme based on supply-demand dynamics was the first application in leveraging the co-regulation mechanism from plant physiology. Our modeling results indicated that the plant-centric SDD scheme may make significant contributions to water sustainability, compared with the existing highly efficient MAD scheme with optimized constant threshold, under both the current and future climate conditions. It should be noted that the traditional MAD irrigation scheme could be further optimized through many other settings, such as the growing stage-specific thresholds, which is beyond the scope of this research. Multi-year field experiments comparing the SDD and MAD irrigation schemes are now underway at two sites in Nebraska starting from 2021: one site in eastern Nebraska with a variable rate irrigation system and another site at North Platte in western-central Nebraska with center pivot irrigation system. Data collected from these experiments will be used to further validate the results reported here.

Adopting the SDD irrigation scheme could provide economic incentives to producers. Our study has shown that SDD could lead to about $13.8 ha$^{-1}$ increase of profit when compared with MAD under current climate conditions. Moreover, producers could further increase their net revenue by the emerging water right trading. For example, producers could sell or lease their saved water rights from the agriculture sector (with lower water value) to the industry sector (with higher water value) and to the federal government, state agencies, and/or nongovernmental organizations for environmental conservation purposes[39,40].

The SDD irrigation scheme can potentially be deployed in fields equipped with various irrigation systems, such as center pivots and drip irrigation systems. Practically, a critical challenge of applying the SDD irrigation scheme is to determine the dynamic soil moisture threshold, which varies with VPD based on the co-regulation of soil moisture and VPD on stomatal conductance. In general, stomatal conductance is highly dynamic, and it is also difficult to directly measure stomatal conductance with gas exchange instruments routinely due to the high labor and time costs. SDD also requires the information of soil moisture and VPD (daily mean values) to estimate daily irrigation requirements. Using sensors for both soil moisture and VPD may be viable if the costs of sensors are sufficiently low, which may not be the case in the near term. Fortunately, the application of *ecosys* model in this research indicated that the co-regulation pattern of soil moisture and VPD on stomatal conductance for a specific site could be quantified through a data-model fusion method[15]. Specifically, advanced satellite fusion algorithms[41,42] have been developed to integrate multi-source satellite data to generate accurate field-level and high-frequency (e.g., daily) vegetation data, such as ET[43,44], leaf area index (LAI)[45], and photosynthesis (i.e., gross primary productivity, GPP)[46,47]. These field-level variables could be helpful to robustly drive and constrain advanced agroecosystem models (e.g., *ecosys* used in this study) to simulate both crop dynamics (e.g., stomatal conductance) and hydrological conditions (e.g., soil moisture)[48]. This approach is a promising direction to enable the implementation of the SDD irrigation scheme at every field based on models constrained by satellite-derived, field-level ecohydrological variables[48]. Additionally, real-time weather forecasts could be applied to provide forecasted ecohydrological variables for the SDD irrigation scheme, allowing producers to initiate irrigation days in advance in order to increase soil water supply and/or to avoid plant water stress due to the anticipated decrease in soil moisture and/or increase in VPD. We thus hope the SDD irrigation scheme could be potentially applied at the field scale and contribute to water sustainability in the future.

## Methods

**Field measurements**. We used two sets of field measurements of soil moisture, VPD, and stomatal conductance of maize at the daily scale to illustrate a proof-of-concept for the co-regulation of soil moisture and VPD on stomatal conductance.

The first set was measurements from greenhouse experiments of maize (seed: Dekalb hybrid DKC52-04) at Colorado State University during the 2013 growing season (planted on June 10, 2013)[49]. There were two treatments (well-watered, WW, and water-stressed, WS) with five plants per treatment. The soil of the greenhouse experiments was the air-dried soilless substrate (8.8 kg) consisting of a 1:1.3 by volume ratio of Greens GradeTM, Turface® Quick Dry® and Fafard 2SV in 26 L pots[49]. The soil moisture measurements came from soil moisture sensors (Decagon5TM sensors) installed in the middle of the pots (~6 inches from top). The greenhouse measurements of leaf-level stomatal conductance and soil moisture were performed in approximately 2-week intervals beginning in the vegetative stage and continuing until plant senescence (DOY 198–199, 210–211, 217–218, 233–234, 247), with 11 replicates for each plant under two treatments (WW and WS). The environmental variables, such as relative humidity and air temperature, were continuously measured in minutes. Other detailed experimental setups can be found in Miner and Bauerle (2017)[49].

The second set was eddy-covariance measurements of maize cropping systems (seed: Pioneer 33P67/33B51) from 2001 to 2012 at three AmeriFlux sites (US-Ne1, Ne2, and Ne3). US-Ne1 and Ne2 were irrigated sites, with a continuous maize cropping system during 2001–2012 for US-Ne1 and with a maize-soybean rotation cropping system during 2001-2009 and then a continuous maize cropping system during 2010-2012 for US-Ne2. US-Ne3 was rainfed with a maize-soybean rotation cropping system during 2001–2012. The soil at the three AmeriFlux sites was a deep silty clay loam consisting of four soil series: Yutan, Tomek, Filbert, and Filmore. There are three replicates with the soil moisture sensors (theta probes: ML2, Dynamax Inc.) installed horizontally with the profile of soil depth (10, 25, 50, and 100 cm) in the US-Ne1 and US-Ne2, and four replicates with soil moisture sensors (theta probes: ML2, Dynamax Inc.) installed horizontally with the profile of soil depth (10, 25, 50, and 100 cm) in the US-Ne3 (http://csp.unl.edu/public/ G_moist.htm). The soil moisture data used here was from the top soil layer (10–25 cm). The canopy-level stomatal conductance ($G_s$) was derived by inverting the Penman-Monteith equation[50] (Equations 1 and 2) from the eddy-covariance measurements at the hourly scale[18,24,51], and the averaged value near midday (from 12:00 to 14:00) was applied as the daily canopy-level stomatal conductance to remove the diurnal cycle. This inversion was only conducted during peak growing season (July and August) to avoid the impact of LAI[24]. The impact of evaporation from canopy interception and of low incoming shortwave radiation was removed by data filtering[24], i.e., excluding the data within 2 days following every precipitation and irrigation event, and periods of low incoming shortwave radiation conditions (<500 Wm$^{-2}$).

$$\lambda E = \frac{\Delta(R_n - G) + \rho c_p g_a VPD}{\Delta + \gamma(1 + g_a/G_s)} \quad (1)$$

$$G_s = g_a \gamma \Big/ \left\{ \frac{\Delta(R_n - G) + \rho c_p g_a VPD}{\lambda E} - (\Delta + \gamma) \right\} \quad (2)$$

where $\Delta$ is the slope of the water vapor deficit; $R_n$ and $G$ are net radiation and soil heat flux, respectively; $\rho$ is the air density; $c_p$ is the specific heat capacity for dry air; $g_a$ is aerodynamic conductance; $\gamma$ is the psychometric constant; and $\lambda E$ is evapotranspiration.

**Advanced process-based model (*ecosys*)**. We further used an advanced process-based agroecosystem model, *ecosys*, to reproduce the co-regulation pattern at the daily scale and to investigate the performances of different irrigation schemes with continuous maize cropping systems across 12 sites in Nebraska with a large rainfall gradient under current climate (2001–2019) and RCP-8.5 scenario (2058–2076) (Fig. 3 and Table S1). *Ecosys* simulates coupled energy, water, carbon, and nutrient cycles[52–56], and has been extensively tested in various agricultural ecosystems[52–54,56,57]. *Ecosys* could simulate all major agricultural management practices, such as tillage[58], crop rotation[58], fertilizer[59], and irrigation[60].

*Physical processes*. *Ecosys* uses a multilayered soil-root-canopy system to get hourly two-stage convergent solutions for crop carbon assimilation, water uptake, and energy fluxes[52–55,60,61]. The first stage focuses on the convergence of canopy temperature for the first-order canopy energy balance in Equation 3, including net radiation $R_n$, latent heat flux $LE$ (including from evaporation, $LE_v$ in Equation 4a, and transpiration, $LE_c$ in Equation 4b), sensible heat flux $H$, and soil heat flux $G$. Canopy latent heat is controlled by aerodynamic resistance ($r_a$)[52,53] and canopy stomatal resistance ($r_c$). Canopy stomatal resistance is postulated by two controlling mechanisms: canopy photosynthesis (leaf level driven by rates of carboxylation vs. diffusion, Equation 5) and canopy turgor potential ($\psi_t$, canopy level constrained by water status, Equation 6). The second stage focuses on the convergence of canopy water potential ($\psi_c$) and thereby $r_c$ (=$1/g_s$) at which transpiration ($E_c$) based on canopy energy balance (left term in Equation 7) equals root water uptake from multiple soil layers (right term in Equation 7). Root water uptake is calculated using water potential differences between soil ($\psi_s$) and root ($\psi_r$), and the soil and root hydraulic resistances ($\Omega_s$ and $\Omega_r$) in each rooted soil

layer.

$$R_n + LE + H + G = 0 \tag{3}$$

$$LE_v = L(e_a - e_c)/r_a \tag{4a}$$

$$LE_c = L(e_a - e_c)/(r_a + r_c) \tag{4b}$$

$$r_{c(min)} = (C_b - C_i')/(1.56V_c') \tag{5}$$

$$r_c = r_{c(min)} + (r_{c(max)} - r_{c(min)})e^{-5.0\psi t} \tag{6a}$$

$$\psi_t = \psi_c - \psi_\pi \tag{6b}$$

$$(e_a - e_c)/(r_a + r_c) = \sum_l \sum_r (\psi_c - \psi_{s,l})/(\Omega_{s,r,l} + \Omega_{r,r,l} + \sum_x \Omega_{a,r,l,x}) + X_c \partial \psi_c / \partial t \tag{7}$$

where $r_{c(min)}$ is the minimum $r_c$ at $\psi_c = 0$ MPa; $r_{c(max)}$ is canopy cuticular resistance to vapor flux; $C_b$ is [CO$_2$] in canopy air; $C_i'$ is [CO$_2$] in canopy leaves at $\psi_c = 0$ MPa; $V_c'$ is the potential canopy CO$_2$ fixation rate at $\psi_c = 0$ MPa; $\psi_\pi$ is canopy osmotic potential; $e_a$ is atmospheric vapor density at air temperature ($T_a$) and ambient humidity; $e_c$ is canopy vapor density at canopy temperature ($T_c$) and $\psi_c$; $\Omega_{s,r,l}$ is radial resistance to water transport from soil to surface of roots or mycorrhizae; $\Omega_{r,r,l}$ is radial resistance to water transport from the surface to axis of roots or mycorrhizae; $\Omega_{a,r,l,x}$ is axial resistance to water transport along axes of primary ($x = 1$) or secondary ($x = 2$) roots or mycorrhizae; $l$ is soil or canopy layer; $r$ is root or mycorrhizae; and $X_c$ is canopy capacitance.

Photosynthesis at the leaf-level is calculated using the Farquhar model for C3 plants and the Farquhar model plus mesophyll-bundle sheath carbon exchange for C4 plants with specific azimuth, leaf inclination, exposure of light conditions (i.e. sunlit and shaded leaves), and canopy height. Canopy photosynthesis is the sum of the photosynthesis of all individual leaves. The carbohydrate is then allocated for maintenance respiration ($R_m$) in both the shoot and root, and the remainder for growth respiration ($R_g$), and dry mass (DM) formation. The phenologically-driven plant carbon allocation ratio among shoot and root organs is impacted by the number of phyllochron intervals and the water and nutrient status of the plant. DM of shoots is partitioned to seven organs (leaf, sheath, stalk, soluble reserves, husk, cob, and grain) with dynamic partitioning ratios varying with growing stages. Seed number and kernel mass are set during postanthesis growth stages to determine the yield upon harvest. More details about the biophysical and biochemical processes in *ecosys* can be referred to the supplement of Grant, et al (2020)[61].

*Model setup. Ecosys* was validated at three AmeriFlux sites (US-Ne1, Ne2, and Ne3, https://ameriflux.lbl.gov/)[60] and 12 sites across Nebraska (Table S5, Fig. S12–S15).

Three AmeriFlux sites (US-Ne1, Ne2, and Ne3) had the complete data from 2001 to 2012 to test model performance, including the meteorological variables (such as surface air temperature, downward shortwave radiation, wind speed, and precipitation), eddy-covariance fluxes (such as GPP, net ecosystem exchange-NEE, and LE) from the FLUXNET2015 dataset[62] (http://fluxnet.fluxdata.org/data/fluxnet2015-dataset/) and detailed ground-based crop growth observations (i.e. planting/harvest date, irrigation/fertilization records, LAI, and yield) from the Carbon Sequestration Program (CSP) at University of Nebraska-Lincoln's Agricultural Research and Development Center (http://csp.unl.edu/Public/sites.htm)[63]. The daily and monthly model simulations of GPP, ET, LAI, and yield matched well with the eddy-covariance and ground-based observations (Table S5, Fig. S12–S15).

The hourly meteorological variables, including surface air temperature, humidity, wind speed, precipitation, and downward shortwave radiation, from 1979 to 2019 (1979–2000 set for model spin-up and 2001–2019 set as current climate conditions) across 12 sites in Nebraska were obtained from the North American Land Data Assimilation System (NLDAS-2). For climate change, the ensemble predictions of air temperature, precipitation, and carbon dioxide concentration [CO$_2$] at the same irrigated sites in Nebraska were ensemble average projections from 15 Coupled Model Intercomparison Project phase 5 (CMIP5) models under Representative Concentration Pathway 8.5 (RCP-8.5)[64] (Table S4). RCP-8.5 was selected as the future scenario to investigate the high warming conditions with the highest greenhouse gas (GHG) emissions[65]. *ecosys* model simulation under the current climate (2001–2019) was extended through three additional 19-year cycles from January 1, 2020 to December 31, 2076 under the RCP-8.5 scenario using the incremental change scheme without the acclimation of parameters in *ecosys*, and the fourth cycle 2058–2076 was selected as the future climate to investigate the climate change effects. It should be noted that the acclimation of parameters in *ecosys* could be applied to maintain high maize yields with increased air temperature under climate change, like the high maize yields under hot, semiarid to arid climate with intensive irrigation in the Texas Panhandle. As we aimed to investigate the impacts of environmental variables to maize under climate change, no acclimation of parameters in *ecosys* is suggested under climate change.

The soil information (i.e., bulk density, field capacity, wilting point, soil texture, saturated hydraulic conductivity, soil organic carbon, pH, and cation exchange capacity) was acquired from the Gridded Soil Survey Geographic Database (gSSURGO) dataset[66]. There were 12 soil layers (the depth of the bottom for each layer: 0.01, 0.05, 0.1, 0.15, 0.18, 0.28, 0.35, 0.59, 0.92, 1.32, 1.6, and 2.00 m) with a maximum depth of 2.0 m. The planting date of the continuous maize cropping systems at the 12 sites was obtained from the USDA NASS weekly Crop Progress Reports (2001–2019) with the fertilizer (18 g N m$^{-2}$ and 5 g P m$^{-2}$ per year) applied two days before planting, and the crops were harvested on October 31. Other land management practices were set as the same across the 12 irrigated sites in Nebraska, including planting density (8.4 plants m$^{-2}$), tillage practice (no tillage), and crop type (continuous maize cropping systems). The auto-irrigation scheme in *ecosys* with the widely used soil-based MAD-50% was applied to determine the irrigation scheduling at 12 sites to test the model performance. The NASS county-level irrigated maize yield (2010–2019) at the counties where the 12 sites were located was used as the observations. The probability density function with Gaussian kernel density estimation of irrigated yields from *ecosys* model at 12 sites showed good agreement with that of the National Agricultural Statistics Service (NASS) county-level irrigated yields during the period from 2010 to 2019 (Fig. S15h), which further validated the reliability of the *ecosys* model.

**Empirical nonlinear statistical model.** There were linear and nonlinear empirical models available for modeling stomatal conductance[20,24,67]. As the exponent of VPD was close to one for croplands[25] and the linear function between the stomatal conductance and soil water potential led to the logistic function between the stomatal conductance and soil moisture, we used an empirical nonlinear statistical model to describe the co-regulation of soil moisture and VPD on stomatal conductance (Equation 8), including two sub-functions[24]. The first one denoted an inverse proportional relationship between the VPD and stomatal conductance[25,68], and the other represented the logistic function between soil moisture and stomatal conductance[69]. The impacts of [CO$_2$], nutrient, radiation, and temperature on stomatal conductance were not considered in this research.

$$G_s = f(VPD, \theta) = \left( \frac{a_1}{VPD - a_2} + a_3 \right) \times \left( \frac{b_1}{1 + \exp(b_2(\theta - b_3))} \right) \tag{8}$$

The field measurements of soil moisture, VPD, and stomatal conductance at the daily scale from the greenhouse experiments and three AmeriFlux sites (US-Ne1, Ne2, and Ne3) were applied to investigate the co-regulation of soil moisture and VPD on stomatal conductance (Equation 8) as observational evidences. With the validated *ecosys* model at three AmeriFlux sites (US-Ne1, Ne2, and Ne3) and 12 sites across Nebraska (Table S5, Fig. S12–15), the model simulations of daily soil moisture, VPD, and canopy-level stomatal conductance during the peak growing season (July and August) were applied to investigate the co-regulation patterns across 12 sites in Nebraska. It needs to be noted that the fitted parameters under the current climate (2001–2019) should be updated under RCP-8.5 scenario, as maize may respond differently under climate change due to increased air temperature and VPD.

In addition, Lindeman, Merenda, and Gold method (LMG)[30] was used to identify the relative importance of soil moisture and VPD on the stomatal conductance. LMG decomposed the determination coefficients of a linear regression ($R^2$) to the contributions of soil moisture and VPD, i.e., to quantify the variation of stomatal conductance that can be explained by soil moisture and VPD, while taking the correlation between the soil moisture and VPD into account.

**Agricultural irrigation management.** We proposed the plant-centric irrigation scheme based on water supply-demand dynamics (SDD). By using *ecosys* model simulations, we further compared its performance with the widely used soil moisture-based irrigation scheme, i.e. management allowable depletion (MAD) under both current (2001–2019) and future climate conditions (RCP-8.5 scenario, 2058–2076). The universal (under current climate and RCP-8.5 scenario) and site-specific (under current climate) parameters of SDD and MAD irrigation schemes across 12 sites in Nebraska were optimized for maximizing economic profit (Equation 9). For the irrigation setting in the *ecosys* model, irrigation was triggered when soil moisture was lower than the soil moisture threshold of SDD (varying with VPD) and MAD (constant) at the daily scale during the growing season. Soil moisture was the weighted average of soil water content in different soil layers within the root zone, which was varying with dynamic root growth but no more than the top 9 soil layers with a depth of 0.92 m to reduce the impacts of deep wet soil. For simplification, we ignored the constraints on irrigation amount and duration from irrigation infrastructures. Irrigation amount was determined by water required to fill current soil water content to field capacity. Each irrigation event lasting 24 h was incorporated into the *ecosys* model at a daily time step in real time. Furthermore, we used two indexes, economic profit and irrigation water productivity, to evaluate the performances of SDD and MAD irrigation schemes. Economic profit was the net revenue based on marketable yields and costs, including irrigation costs and fixed costs of production (Equation 9). Irrigation water productivity ($IWP$)[70] was the ratio between the marketable yields and irrigation amount during the growing season (Equation 10). To reduce the impacts of parameters in performance assessment, we used the recorded parameters at Nebraska, United States in 2019 (Table S3). Relative and absolute differences in irrigation amount, yield, profit, and irrigation water productivity between SDD and MAD irrigation schemes under current climate (2001–2019) and RCP-8.5 scenario

(2058–2076) were calculated (Equations 11 and 12).

$$\text{Profit} = \text{Revenue} - \text{Costs} = y \times p_{maize} - \frac{I}{\lambda} \times \Gamma_{irrigation} - K_{fixed} \qquad (9)$$

$$IWP = \frac{y}{I} \qquad (10)$$

$$\text{Relative difference} = \frac{SDD - MAD}{MAD} \times 100\% \qquad (11)$$

$$\text{Absolute difference} = SDD - MAD \qquad (12)$$

where $y$ is maize yield (t ha$^{-1}$); $p_{maize}$ is the price of maize (\$ t$^{-1}$); $I$ is the irrigation amount (mm); $\Gamma_{irrigation}$ is the price of irrigation (\$ m$^{-3}$); $\lambda$ is the irrigation application efficiency of the center pivots; $K_{fixed}$ is the fixed costs of production (\$ ha$^{-1}$), including the costs of seeds, fertilizer, storage, and so on; and $IWP$ is the irrigation water productivity (kg m$^{-3}$).

## Data availability

Data supporting the conclusions of this study are properly cited and publicly available. The field measurements from greenhouse experiments are available upon requests from the co-authors (W.L.B. and G.L.M). Extra data are available upon request from the corresponding authors.

## Code availability

The source code of the *ecosys* model used in this study is publicly available from https://github.com/jinyun1tang/ECOSYS.

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

## Acknowledgements

We acknowledge the support from USDA National Institute of Food and Agriculture Foundational Program Cyber-physical systems (2019-67021-29312) and NSF CAREER award (1847334) managed through the NSF Environmental Sustainability Program.

## Author contributions

J.Z. and K.G. conceived the study; J.Z., K.G. and B.P. designed research; J.Z. collected data, analyzed data; J.Z., K.G., and B.P. wrote the paper with the inputs from all other co-authors; W.L.B. and G.L.M collected the greenhouse experiment data; H.K., T.E.F., D.R.R., D.M.H. and A.E.S. collected and processed the eddy-covariance data; W.Z., M.P., C. J., Y. Y. and R.F.G. helped with the calibration and validation of the process-based model (*ecosys*); and R.F.G. is the author of the process-based model (*ecosys*).

## Competing interests

The authors declare no competing interest.
