## [Peer Review File · Nature Communications]

REVIEWER COMMENTS

Reviewer #1 (Remarks to the Author):

Title: Combining soil water supply and atmospheric evaporative demand for sustainable irrigation

In this paper, the authors build a modeling framework for irrigation management that is “plant-centric” (SDD), meaning that it focuses on the two key hydrologic drivers of stomatal conductance—soil moisture (water supply) and vapor pressure deficit (atmospheric demand). The model is compared to a more commonly applied irrigation management scheme that depends only on soil moisture (MAD). The authors apply the SSD and MAD schemes at 12 maize sites spanning an aridity gradient in Nebraska in both current and future climate conditions. The authors find that the SSD irrigation scheme reduces water use compared to MAD while maintaining yields in current climate conditions. In future climates, it similarly reduces water use compared to MAD.

Overall, I found the methods and results very compelling. I like how the authors provided leaf-level and field-level (eddy covariance) observations to support the stomatal co-regulation hypothesis in addition to revealing co-regulation in the ecosystem model. I also like how the SDD model was evaluated using both site-specific and across-site calibration— it really highlighted the applicability of the SDD model.

Major comments:

(#1) In Fig. 2 (and Fig. S2), why are there differences in the size and shape/strength of co-regulation zones? Is it because the plants are actually regulating their stomata differently? I think this is important to discuss this in the section on co-regulation.

(#2) The authors might think more about the structure of the paper. There is a “Results” section, but there are key results discussed prior to this section, including predictions with climate change (L117-L139).

(#3) Related to the point above... I also wonder how the results shown in Fig. 3 and described in lines L117-139 relate to the main results of the paper regarding irrigation schemes? While I think the results in Fig. 3 are really interesting, I think they detract from the main results of the paper. I see the observations and process-based model (Fig. 2) as being clear motivation for creating the plant-centric SDD irrigation model. I think if the paper went from motivation to model creation, it would flow much better. (Maybe Fig. 3 is another paper? Or moved to the SI? Or somehow integrated into the main results of the paper?)

Minor comments:

Abstract: Very clear. I think some of the sentences may work better in the present tense, e.g., “Specifically, we proposed ...” sounds awkward because you are currently proposing a method in the paper.

Introduction:

L58-62, very long sentence, I recommend revising for clarity.

L63, “but” to “and”

L63, “, which also has a large impact on plant water relations” is a bit redundant

L66-67, Is it really the interplay between these three things, or the interplay between water supply and atmospheric demand via plant physiological regulation?

Last paragraph, can these three objectives be distilled more (specifically, obj 1 and obj 3)? When first reading this paragraph, it’s hard to get a clear roadmap because of the sub-objectives.

Co-regulation section:

L102, "Figure 2 shows the co-regulation pattern of <DAILY (?)> soil moisture and VPD..." is it daily? I think the time-scale needs to be mentioned here/in this paragraph/in the paper in general.

L108, are the greenhouse measurements at the leaf-level?

L113, is the "228 site-years" the same set of data as the "12 sites (e.g. L117). If so, it would be helpful to refer to them using the same language

L114, do you mean, "The co-regulation pattern indicates that plants can have water stress even at high soil moisture IF VPD is also high"?

L117, are you analyzing contributions of VPD and SM to daily conductances? I think specifying the time scale is important for a couple reasons... stomata are not responding to daily averages and the physical coupling between VPD and SM is very time scale dependent (with stronger coupling at longer time scales). So the time-scale at which these relationships are resolved is a simplification/assumption, which is fine, but should be stated.

L126, Are the relationships in Fig. S3 & S4 statistically significant? Since there are already lines of best fit, it might be useful to add summary statistics to the figures or captions.

Fig. 2, I'm assuming each dot represents one day... is this right?

SDD section:

L184 and Fig. 4, I think the colors should be changed for the water saving and water surcharge zones, since they are in the Gs colormap.

Description of Fig. 4, I think the text should include a conceptual description of the three sub-panels in Fig. 4 (VPD = 1kPa...). This would help explain the SDD model.

Results:

Very nicely written and easy to follow.

Fig. 6b, I was confused with the title of this subplot

Discussion:

L275, "The proposed SDD scheme is pushing the boundaries in scientific irrigation management through applying the above fundamental theories in the Soil-Plant-Atmosphere-Continuum." I'm not sure this sentence adds anything... I'd delete it.

L296, "Fortunately, we find that the co-regulation pattern of soil moisture and VPD on stomatal conductance for a specific site can be quantified through a data-model fusion method." I am confused by the "we find"...

L298-311, I think this section could be written a little differently, with a more speculative tone. As it is written now, it almost sounds like the authors have tested this modeling framework.

Methods:

L319, I think a better description of the "greenhouse measurements" is needed. I am not sure the spatial or temporal scale.

L322, what is the depth of the soil moisture measurements?

Reviewer #2 (Remarks to the Author):

Review of "293458_0_art_file_5239341_qmwmrg.pdf" titled: "Combining soil water supply and atmospheric evaporative demand for sustainable irrigation"

OVERALL:

The manuscript describes the use of a general ecosystem model (ecosys) to simulate impacts of two different irrigation scheduling schemes across a range of climate roughly from eastern Nebraska to western Nebraska. The irrigation scheduling schemes are one based on management allowed depletion (MAD) of soil water with a constant MAD threshold and one based on considering both soil

water depletion and stomatal conductance with the soil water depletion threshold (MAD level) varying in concert with simulation stomatal conductance. The schemes are applied using simulation modeling to present data and to future climate and purport to show a difference in outcomes depending on which scheme was used.

The ecosystem model has been widely used but is not “rigorously validated” as claimed in the abstract. The one cited prior use of the model with irrigated maize in Nebraska was based on one year of data at one location (Grant et al., 2007). The present study uses three locations in Nebraska but yield simulation is biased and inaccurate (r^2 of 0.37) with a regression slope of ~ 0.78 and intercept of ~ 40 bushels per acre.

The paper makes claims of innovation and being the first to apply the theory of plant water relations to irrigation management, including co-regulation, but this is not so. The paper ignores a plethora of earlier work, some ongoing, that recognized and utilized both soil water deficit and plant response to soil water deficit and atmospheric demand (see specific comments).

The total reliance on evapotranspiration (ET) sensed by eddy covariance presents the results from being definitive because of the widely recognized tendency of eddy covariance methods to underestimate ET by 15% to 30%. There appear to be no mass balance-based ET data used, whether from weighing lysimetry or from deep soil water content sensing such as with the neutron probe.

The data used to develop stomatal conductance relationships with soil water content and vapor pressure deficit were all from three sites within a 14-km radius near Mead, Nebraska in far eastern Nebraska. US-Ne3 is a rainfed site at Mead where the cropping is a maize-soybean rotation. US-Ne1 and US-Ne2 are also near Mead but feature irrigated maize-soybean. None of the data are from the more arid sites and different soils that would be found in central or western Nebraska. Therefore, projections of the co-regulation across the eleven other sites stretching from eastern to western Nebraska appears to be based entirely on computer simulation. To be sure, the methods leave this point somewhat obscurely described.

The stomatal conductance was derived from inversion of the “Penman-Monteith equation” using ET from eddy covariance data at the three eastern Nebraska sites but the version of the Penman-Monteith formulation used is not referenced and details of the inversion are not given. It is important to give details of this inversion because stomatal conductance changes with VPD, solar irradiance, and leaf water potential and because the specific formulation of Penman-Monteith used can affect the results. How were leaf water potential and solar irradiance factored in? The Penman-Monteith formulation allows for solar irradiance but not for leaf water potential. This is far from an obvious inversion although it has been done before. Lascano and others have demonstrated a two-source (canopy and soil surfaces) implicit ET model and have shown how it can be inverted to compute stomatal conductance (Lascano et al., 2010a,b). Because it computes canopy and air temperatures separately, the Lascano et al. approach is more accurate than is inversion of the Penman-Monteith formulation under conditions where canopy and air temperatures vary substantially, which is the case in more arid, and highly irrigated regions. Because the Penman-Monteith assumes equality of air and canopy temperatures, inverting the Penman-Monteith will give approximately the same results as an implicit solution in a more humid climate (e.g., eastern Nebraska) where air and canopy temperatures tend to be similar but will give a different and less accurate stomatal conductance in regions where VPD and canopy-air temperature differences are both greater (e.g., western Nebraska). In addition, there is some evidence that eddy covariance bias increases in more arid environments, which would add more to the bias of the results of this study.

There appears to be a disconnect between the source of ET data at the 12 sites across Nebraska (site-specific eddy covariance data) and the source of forcing meteorological data, which was not site-specific but came from NLDAS-2 and was therefore interpolated.

The results indicate that irrigation would “not resolve the yield loss from rising temperatures” but considering corn yields in places with much greater VPD and higher temperatures, one wonders how realistic these simulations are. For example, In the northwest corner of the Texas Panhandle, county wide corn yields approach 300 bushels per acre and are better than those in much of the corn belt. This in in a region with intensive irrigation due to its hot, semiarid to arid climate, and lack of precipitation. In this region, irrigation quite adequately resolves yield loss under conditions that are already severe.

The use of a constant threshold across soil types and throughout the season as is done here is a simplistic and unrealistic way to use the MAD concept in irrigation management. Early season irrigation management and late season irrigation management typically do not strictly follow the MAD concept for good reasons. Reasons include holding off on early irrigation to force root growth into deeper soil layers and reducing irrigation during vegetative growth (by adjusting the MAD level). Also, the depth of soil considered in computing soil moisture varies with root growth in sophisticated irrigation management and may be greater than the 0.92 m used in this study.

The description of the applicability of the SDD irrigation scheme is all in the realm of possibility but have not been demonstrated in real-world tests so are conjecture. The difficulty in applying a complex process-based simulation model such as ecosys inhibits even research scientists from using such models, not to mention irrigation managers. Throughout, the manuscript tends to certainty when the subjunctive tense would be more appropriate. The description of things as rigorous when they are susceptible to bias is not acceptable. A more humble stance would get the authors further, particularly because the study is primarily a modeling exercise and does not test the approach by applying it to irrigation management in real fields.

SPECIFIC COMMENTS:

L1-2: Title is difficult to understand.

L34: Based on the contents of the manuscript, I do not think it is defensible to refer to the model as “rigorously validated”.

L73-73: This is a meaningless statement. One could just as easily write the converse, which is that in some regions, like the U.S. corn belt, constraints from soil water deficit on crop growth can be even larger than that from VPD. Suggest deleting it.

L78-79: The co-regulation of stomatal conductance by soil water availability and evaporative demand is well known and does not need verification as stated in the first objective. What needs verification is the use of the process-based ecosystem model in this context. The model was not designed with irrigation management in mind. Recommend rephrasing this objective statement.

L87-90: This is not the first such effort at applying the theory of plant water relations to irrigation management. Earlier work exists that employs a combined soil water sensing and modeling approach to irrigation management. Suggest you discuss that work, including:

Lascano, R.J. 2000. A general system to measure and calculate daily crop water use. *Agronomy J.* 92, 5, 821-832. <https://doi.org/10.2134/agronj2000.925821x>

Lascano, R. J., Baumhardt, R. L., Hicks, S. K., Evett, S. R., and Heilman, J. L. 1996. Daily measurement and calculation of crop water use. pp. 225-230. In C. R. Camp, E. J. Sadler, and R. E. Yoder (eds.) *Proc. International Conference. Evapotranspiration and Irrigation Scheduling*, San Antonio, TX.

Figure 1: Suggest adding an arrow from the rhizosphere to the "a" box. Otherwise, the caption does not completely jive with the figure.

L143-144: Give details of how stomatal conductance was derived from the Penman-Monteith equation

L157: The concept of potential evapotranspiration ceased to be used in irrigation science some decades ago in favor of the concept of reference evapotranspiration. Is the “potential evapotranspiration” written here supposed to be reference evapotranspiration? If it was computed using Penman-Monteith (e.g., the ASCE 2005 Penman- Monteith standard) then it is reference ET, not potential ET.

L169 and Figure 4: I find figure 4 difficult to interpret. The labels for SDD: part 1 and SDD: part 2 do not seem to be adequately explained.

L175-179: What is innovative is the use of a process-based model to set the varying threshold. It has long been recognized that water use rate should impact the soil water content threshold that is used to trigger irrigation. This was the case when the neutron probe was used for irrigation management. Suggest rephrasing and omitting the statement that MAD based irrigation scheduling uses a constant threshold. That is a simplistic view of MAD based irrigation management.

L208: Shouldn't this be "simulated" not "tested"? Rephrasing to "We systematically simulated application and outcomes of the SDD and MAD irrigation schemes..." might be more to the point. A test would involve comparing the simulated results with some objective measure. There does not appear to be an objective measure that was obtained at any of the 12 sites. Indeed, the present study appears not to involve any actual test in the field.

L220: Suggest removing "Besides". It does not seem to serve a purpose in this exposition.

L218-220: I do not see that this statement follows from the previous one. Why is the word "Thus" used here?

L223-224: This phrase is confusing. It appears that it is incompletely worded.

L233-237: The climate change scenarios for Nebraska that I have seen do not predict such large changes in irrigation demand.

L238-240: Just considering corn yields in places with much greater VPD and higher temperatures, one wonders how realistic these simulations are. For example, In the northwest corner of the Texas Panhandle, county wide corn yields approach 300 bushels per acre and are better than those in much of the corn belt. This is in a region with intensive irrigation due to its hot, semiarid to arid climate, and lack of precipitation. In this region, irrigation quite adequately resolves yield loss under conditions that are already more severe.

L250-251: The relative difference is undefined. Is it (result for SDD) minus (result for MAD), or is it (result for MAD) minus (result for SDD)?

L274-275: It is only partially true that a co-regulation mechanism has never been applied in irrigation management. See the works of Wanjura, Mahan, Upchurch, Lascano, O'Shaughnessy, Andrade, etc. who go about it in other ways, combining different models and sensing of canopy temperature (as influenced by stomatal conductance) and soil water content (see review of this work by Evett et al. (2020)). The present manuscript should discuss the prior work along these lines and then draw the distinctions between the present work and the past and some ongoing work, rather than fail to recognize the earlier and ongoing work. The difference between the present paper and earlier ones is that in the present paper the focus is directly on stomatal conductance whereas in earlier works the focus has been on the properties related to changes in stomatal conductance. In many cases those properties are more easily sensed in the field than is stomatal conductance. See the following papers for plentiful citations of earlier work:

Evett, S.R., S.A. O'Shaughnessy, M.A. Andrade, P.D. Colaizzi, R.C. Schwartz, H.S. Schomberg, K.C. Stone, E.D. Vories, and R. Sui. 2020. Theory and development of a VRI decision support system: The USDA-ARS ISSCADA approach. *Trans. ASABE*. 63(4): <https://doi.org/10.13031/trans.13922>

Also, see:

O'Shaughnessy, S.A., M. Kim, M.A. Andrade, P.D. Colaizzi, and S.R. Evett. 2020. Site-specific irrigation of grain sorghum using plant and soil water sensing feedback - Texas High Plains. *Agric. Water Manage.* 240 (2020) 106273. <https://doi.org/10.1016/j.agwat.2020.106273>

L278: Suggest changing "show" to "indicate", and suggest changing "can" to "may". The results are not definitive.

L281: Suggest changing "can" to "could". The purported benefits are not proven in this report. Use of the subjunctive in this and other statements if appropriate.

L304-309: This description of the application of the SDD irrigation scheme is all in the realm of possibility but these have not been demonstrated so are conjecture, so please use the subjunctive tense – "Could" rather than "can", "could enable" rather than "enables".

L309-310: At least this claim of practicality is presented as a belief, not a fact. However, applying a complex SPAC model such as ecosys to field situations is quite difficult. Much easier but still effective prior solutions to irrigation management have failed to be widely adopted yet could be enabled more

widely through the use of satellite remote sensing and associated simulation modeling. The SDD approach has yet to be demonstrated to be an effective tool, much less widely adopted.

L322-323: Which version of the Penman-Monteith formulation was used? Please give details of this inversion from ET data to stomatal conductance. Stomatal conductance changes with VPD, solar irradiance, and leaf water potential. How were leaf water potential and solar irradiance factored in? The Penman-Monteith formulation allows for solar irradiance but not for leaf water potential. This is far from an obvious inversion although it has been done before. Lascano and others have demonstrated a two-source (canopy and soil surfaces) implicit ET model and have shown how it can be inverted to compute stomatal conductance (Lascano et al., 2010a,b). Because it computes canopy and air temperatures separately, the Lascano et al. approach is more accurate than is inversion of the Penman-Monteith formulation under conditions where canopy and air temperatures vary substantially, which is the case in more arid, and highly irrigated regions. Because the Penman-Monteith assumes equality of air and canopy temperatures, inverting the Penman-Monteith will give approximately the same results as an implicit solution in a more humid climate (e.g., eastern Nebraska) but will give a different and less accurate stomatal conductance in regions where VPD and canopy-air temperature differences are both greater. In addition, there is some evidence that eddy covariance bias increases in more arid environments, which would add more to the bias of the results of this study.

Lascano, R.J., C.H.M. van Bavel, and S.R. Evett. 2010a. A field test of recursive calculation of crop evapotranspiration. *Trans. ASABE*. 53(4):1117-1126. <https://doi.org/10.13031/2013.32601>

Lascano, R.J., and S.R. Evett. 2010b. Calculation of canopy resistance with a recursive evapotranspiration model. Pp. 20-23 In Symposium 2.1.1 of 19th World Congress of Soil Science, Soil Solutions for a Changing World, 1-6 August 2010, Brisbane, Australia.

L335: Suggest changing "validated" to "tested". No model is ever "valid" but some are thoroughly tested with statistical reports indicating how well the simulated results compared with some objective measures.

L337: Grant et al. (2007) indeed applied ecosys to an irrigated situation but did not describe how irrigation was handled by the model. Although ecosys has been used for a wide range of ecosystems, ranging from peat bogs, to forests, to agriculture, it appears that it has not been tested for simulation of ET and yield of maize and soybean beyond the one study. In the Grant et al. (2007) study, about 17% of variations in eddy covariance LE fluxes for irrigated maize were not explained by the model, while irrigated grain yield was observed was close to that simulated. With only one year of maize data, the Grant et al. (2007) report is not a "rigorous" test. ecosys is currently being used in an AgMIP model intercomparison study using multiple years of LE data from eddy covariance and weighing lysimeter that may shed more light on its ability to simulate ET.

L378: I recommend omitted the work "rigorously". A test against eddy covariance ET data is not rigorous because eddy covariance ET can underestimate crop ET by up to 30% with common percentages of underestimation being around 15%. Are there soil water balance-based ET data against which to compare the eddy covariance ET data? I do not think so.

L385-386: Figures S12 through S14 are nice figures but it would be more useful to the reader to place the statistical results in a table where they could be reviewed more easily. The use of the correlation coefficient (r) rather than the coefficient of determination (r^2) tends to inflate the perception of goodness of fit. The r^2 for yield is only 0.37, not convincing (the adjusted r^2 is closer to 0.35). Also, the RMSE and bias values are not very useful without mean values for comparison. More importantly, the regression slope and intercept values should be shown in the table along with the other statistics. In the case of yield, a regression on the data in the figure shows that ecosys tends to overestimate yield at the low end, which is consistent with figure S15b.

L387-390: If I understand correctly, meteorological data used to run ecosys were obtained from NLDAS-2 but ET data were obtained from site-specific eddy covariance systems. Is that correct? If so, then there is a problem mixing meteorological data that isn't from on-site sensors with ET data that are from on-site eddy covariance systems. This can be particularly important in the case of precipitation.

Use past tense when describing what was done. In this case, write "were" instead of "are". This applies to lines 387 through 409 at the very least and really to the entire methods section.

L406 and Figure S4b: I can't agree that the data match "well." In particular, the ecosys data are

strongly skewed towards smaller yields than the NASS data with reduced probability of yields greater in the range from ~180 bu/acre to ~230 bu/acre. Could this bias the results towards stronger negative effects of climate change?

L416-417: This filtering has the potential to bias the results. Particularly in more arid, irrigated environments, late afternoon and early evening ET can be a substantial portion of daily ET. See Tolk et al. (2006).

Tolk, J.A., T.A. Howell, and S.R. Evett. 2006. Nighttime evapotranspiration from alfalfa and cotton in a semiarid climate. *Agron. J.* 98:730–736. doi:10.2134/ agronj2005.0276

L418: Is "screen out" what is meant here? It seems that "identify" is what is meant. If not, then this confused me.

L425-and later: Use past tense when describing what was done. The use of present tense here causes confusion and misunderstanding.

L431-433: This is a simplistic and unrealistic way to use the MAD concept in irrigation management. Early season irrigation management and late season irrigation management typically do not strictly follow the MAD concept for good reasons. Reasons include holding off on early irrigation to force root growth into deeper soil layers and reducing irrigation during vegetative growth (by adjusting the MAD level). Also, the depth of soil considered in computing soil moisture varies with root growth in sophisticated irrigation management and may be greater than 0.92 m.

Reviewer #3 (Remarks to the Author):

The study by Zhang et al. presented an important update to agricultural irrigation schemes. They first used observations from a controlled experiment and flux towers to demonstrate the relative importance of VPD and soil moisture. Then, they used a process based model to test how much water can be saved without sacrificing productivity. Their findings could be of great interest from the landholders and local councils. I find the paper to be well-written with clear logic. Below are some minor points I suggest the authors to consider.

The proposed plant-driven irrigation scheme is not evaluated with real data. I would imagine there will be farms in the same areas with different irrigation intensity. The authors could demonstrate the advantage by using the production data from farming using different irrigation intensity. Although the change of intensity is not necessarily the same as the plant driven scheme. Using real data is usually more convincing than just model predictions. Having said that, I know little about USA farm management so the idea may not be realistic.

The empirical model (Equation 6) is quite different from what often used in other studies. For example, Sulman et al. (2016) used a linear model:

$$G_s = C_1 + C_2 \cdot \ln(\text{VPD}) + C_3 \cdot \text{SWC} + C_4 \cdot \ln(\text{VPD}) \cdot \text{SWC}$$

where C_1 - C_4 are fitted parameters. This fitting could obtain the impacts (i.e., slopes) of VPD and SWC on G_s . However, the model used here is a multiple of VPD and SWC and thus difficult to separate the sensitivity of G_s to each factor. This study then use the R^2 rather than regression coefficient. I feel the authors could consider using a formula that separate the impact of each driver better. The linear model by Sulman could work or a more sophisticated model (e.g., Whitley et al., 2013) might be even better. I'm not saying the authors need to use those models but it makes sense to justify the model selection in the method or introduction.

My last point is with the 'profit' (Figures 5 and 6). The difference between estimated benefit of the new and old schemes has large variance (i.e., 25% quantile crossing zero). I wonder whether the variance is driven by the variation in sites and climate. If so, it may be better for the authors to present when/where the new scheme is better than the old one.

Detailed comments

L157 (figure 3 caption). The MAP used here is only for the growing season. What about the impact of snow melt at early spring. My understanding is some region heavily depend on snow melt but not sure about these sites and region.

L393. The authors chose RCP8.5 which is the most extreme pathway that is unlikely to be true. Some justification is needed for the choice.

References

Sulman, B. N., Roman, D. T., Yi, K., Wang, L., Phillips, R. P., & Novick, K. A. (2016). High atmospheric demand for water can limit forest carbon uptake and transpiration as severely as dry soil. *Geophysical Research Letters*, 43(18), 9686–9695. <https://doi.org/10.1002/2016GL069416>

Whitley, R., Taylor, D., Macinnis-Ng, C., Zeppel, M., Yunusa, I., O'Grady, A., Froend, R., Medlyn, B., & Eamus, D. (2013). Developing an empirical model of canopy water flux describing the common response of transpiration to solar radiation and VPD across five contrasting woodlands and forests. *Hydrological Processes*, 27(8), 1133–1146. <https://doi.org/10.1002/hyp.9280>

Dear all,

We thank the editor, Associate Editor, and three reviewers for their comments and helpful suggestions to improve this manuscript. We have revised the manuscript and addressed these comments point by point, and we believe that we have significantly improved the manuscript. We hope that this revised manuscript could fulfill editors and reviewers' high standards for *Nature Communications*.

Reviewer's comments appear in black, our response appears in blue, and the revised text in the manuscript appears in orange.

We look forward to your further feedback!

- Jingwen Zhang, Kaiyu Guan, Bin Peng, on behalf of all the authors

Dr Kasey Bolles, Associate Editor

Thank you again for submitting your manuscript "Combining soil water supply and atmospheric evaporative demand for sustainable irrigation" to Nature Communications. We have now received reports from 3 reviewers and, after careful consideration, we have decided to invite a major revision of the manuscript.

As you will see from the reports copied below, the reviewers raise important concerns. We find that these concerns limit the strength of the study, and therefore we ask you to address them with additional work. Without substantial revisions, we will be unlikely to send the paper back to review. In particular, reviewers agree that the results and discussion are speculative. We agree with reviewers that the paper would be strengthened by application of the proposed SDD scheme to a real-world environment and a refined spatial analysis of where SDD would be an improvement over MAD. Reviewer 2 raises several concerns about the modeling approach that need to be addressed thoroughly, not limited to: model validation, sources of bias, and refined application of the MAD method for comparison to the proposed method.

Reply: Thank you for handling the review and helpful suggestions. We have revised the manuscript following the comments and suggestions from three reviewers. Please see the detailed reply and revised manuscript. We have summarized the three major comments from the reviewers, and we responded to them below as a summary and provided more details in the responses after.

(1) Regarding **the validated performance of *ecosys* model**, we have updated the **validated performance of *ecosys* model** with maize-cropping systems at three AmeriFlux sites (US-Ne1, Ne2, and Ne3) during the period from 2001-2012 and 12 sites across Nebraska during the period from 2010-2019 (Table S5, Fig. S12-15). The R^2 of crop yields at three AmeriFlux sites (US-Ne1, Ne2, and Ne3) were 0.50 during the period 2001-2012. The regression line of maize yields is close to a 1-to-1 relationship. The difficulty of the application of the SDD irrigation scheme is to determine the dynamic threshold level, which varies with VPD based on the co-regulation of soil moisture and VPD on stomatal conductance. Once the dynamic threshold level of SDD was determined, SDD can be tested for irrigation with the measurements of soil moisture and VPD. As explained in the Discussion section, data-model fusion is a promising direction to achieve this goal.

(2) Regarding **the spatial analysis of the SDD's performance**, we have added more discussion on this point in the revised manuscript. The profit is calculated based on revenue (maize yields and price) and costs (including irrigation costs and other fixed costs). Thus, the profit difference between the proposed SDD and traditional MAD irrigation scheme is mainly determined by the differences of maize yields and irrigation water use, and also the price of maize and irrigation. Based on the historical price in Nebraska 2019, we have investigated the impacts of climate (i.e. aridity index) and soil (i.e. sand fraction) properties to the differences of irrigation amount, crop yields, profits, and irrigation water productivity between SDD and MAD (Fig. S10).

(3) Regarding **the application of SDD to the real-world environment**, we are now doing a 3-year field experiment to test SDD irrigation scheme at two sites in Nebraska starting from this year (one in eastern at Mead and one in central-western Nebraska). Though we are not reporting real-world testing results of SDD in the current manuscript, we believe our study on combining soil water supply and atmospheric evaporative demand for sustainable irrigation based on the validated hydraulically-driven ecosystem model, *ecosys*, is rigorous and has its unique and broader value to the community.

For other concerns, please check the detailed reply and revised manuscript.

REVIEWER COMMENTS

Reviewer #1 (Remarks to the Author):

In this paper, the authors build a modeling framework for irrigation management that is “plant-centric” (SDD), meaning that it focuses on the two key hydrologic drivers of stomatal conductance—soil moisture (water supply) and vapor pressure deficit (atmospheric demand). The model is compared to a more commonly applied irrigation management scheme that depends only on soil moisture (MAD). The authors apply the SDD and MAD schemes at 12 maize sites spanning an aridity gradient in Nebraska in both current and future climate conditions. The authors find that the SDD irrigation scheme reduces water use compared to MAD while maintaining yields in current climate conditions. In future climates, it similarly reduces water use compared to MAD.

Overall, I found the methods and results very compelling. I like how the authors provided leaf-level and field-level (eddy covariance) observations to support the stomatal co-regulation hypothesis in addition to revealing co-regulation in the ecosystem model. I also like how the SDD model was evaluated using both site-specific and across-site calibration—it really highlighted the applicability of the SDD model.

Reply: Thank you for the positive review.

Major comments:

(#1) In Fig. 2 (and Fig. S2), why are there differences in the size and shape/strength of co-regulation zones? Is it because the plants are actually regulating their stomata differently? I think this is important to discuss this in the section on co-regulation.

Reply: Thank you for the comments. The differences in the size and shape/strength of the co-regulation regimes in Fig. 2 and Fig. S2 mainly resulted from different climate and soil properties.

For climate, plants have different responses to water stress under different climate conditions. The climate conditions of greenhouse experiments during the 2013 growing season at Colorado State University were different from the climates at three eddy covariance sites (750-800 mm/y) and site-GD (700-750 mm/y) in Nebraska (Fig. 2). There was a dramatic rainfall gradient across 12 sites in Nebraska (from 800 mm in the east to 400 mm in the west) (Fig. 3a and S2).

For soil, different soil types led to the different degree of difficulty for root water uptake under water stress. The soil of the greenhouse experiments was the air-dried soilless substrate consisting of a 1:1.3 by volume ratio of Greens Grade™, Turface® Quick Dry® and Fafard 2SV. The soil at three eddy covariance sites was deep silty clay loams consisting of four soil series: Yutan, Tomek, Filbert, and Filmore. The soil information at 12 sites across Nebraska with different sand fractions was acquired from the Gridded Soil Survey Geographic Database (gSSURGO) dataset (Table S1 and Fig. 3d).

The relative importance analysis of soil moisture and VPD on stomatal conductance can be used to explain these differences. Larger contributions from VPD to the variations of stomatal conductance resulted in larger VPD-dominated regimes and smaller co-regulation regimes. Thus, based on the LMG results (Fig. 3b) and significant trend with aridity index (Fig. S4a), the regions with wetter climate (smaller aridity index) had larger VPD-dominated regimes, such as site-Mead in eastern Nebraska (Fig. S2a). As there is no significant trend with sand fraction (Fig. S4b), the

impacts to the co-regulation patterns from soil properties were less than those from climate properties.

In addition, different cultivars of maize and approaches to obtain stomatal conductance may also cause the differences in the size and shape of co-regulation regimes in Fig. 2. **For cultivars of maize**, maize may respond differently to water stress due to the genetic diversity. The maize seeds of the greenhouse experiments (Dekalb hybrid DKC52-04) were different from the maize seeds at three eddy covariance sites (US-Ne1, US-Ne2, US-Ne3) (Pioneer 33P67/33B51), while *ecosys* model did not have the option to set the cultivar of maize, as the crop phenology parameters were determined using the observations at US-Ne1, US-Ne2, US-Ne3 and 12 sites across Nebraska. **For the approaches to obtain stomatal conductance**, the greenhouse experiments used the measured leaf-level stomatal conductance from the Li-6400 gas exchange system; the eddy covariance sites used the canopy-level stomatal conductance from the inverted Penman-Monteith equation; while the *ecosys* model used the simulated canopy-level stomatal conductance based on water, energy, and carbon balance.

We have added the discussion of the differences of the co-regulation regimes in the revised manuscript (line 112-118), and added more information about the field measurements in the Methods section (line 332-360). Please see the revised texts in the manuscript below:

Line 112-118: The differences in the size and shape of the co-regulation regimes mainly resulted from varying climate conditions and soil properties (Fig. 2 and S2). In addition, different cultivars of maize and approaches to obtain stomatal conductance may also cause the differences in Fig. 2. Specifically, the impacts of soil and climate properties on the co-regulation regimes were investigated using *ecosys* model and a relative importance method^{29,30} across 12 sites in Nebraska with a large rainfall gradient (Fig. 3). We further quantified the relative contributions of soil moisture and VPD to the variations of *ecosys*-simulated stomatal conductance at the daily scale.

Line 330-358: The first set was measurements from greenhouse experiments of maize (seed: Dekalb hybrid DKC52-04) at Colorado State University during the 2013 growing season (planted on June 10, 2013)⁵¹. There were 2 treatments (well-watered, WW, and water-stressed, WS) with five plants per treatment. The soil of the greenhouse experiments was the air-dried soilless substrate (8.8 kg) consisting of a 1:1.3 by volume ratio of Greens GradeTM, Turface[®] Quick Dry[®] and Fafard 2SV in 26 L pots⁵¹. The soil moisture measurements came from soil moisture sensors (Decagon^{5TM} sensors) installed in the middle of the pots (~6 inches from top). The greenhouse measurements of leaf-level stomatal conductance and soil moisture were performed in approximately 2-week intervals beginning in the vegetative stage and continuing until plant senescence (DOY 198-199, 210-211, 217-218, 233-234, 247), with 11 replicates for each plant under two treatments (WW and WS). The environmental variables, such as relative humidity and air temperature, were continuously measured in minutes. Other detailed experimental setups can be found in Miner and Bauerle (2017).

The second set was eddy covariance measurements of maize cropping systems (seed: Pioneer 33P67/33B51) from 2001 to 2012 at three AmeriFlux sites (US-Ne1, Ne2, and Ne3). US-Ne1 and Ne2 were irrigated sites, with a continuous maize cropping system during 2001-2012 for US-Ne1 and with a maize-soybean rotation cropping system during 2001-2009 and then a continuous maize cropping system during 2010-2012 for US-Ne2. US-Ne3 was rainfed with a maize-soybean rotation cropping system during 2001-2012. The soil at three AmeriFlux sites was a deep silty clay loam consisting of four soil series: Yutan, Tomek, Filbert, and Filmore. The soil moisture data used here was from the top soil layer (10-25 cm). The canopy-level stomatal conductance (G_s) was derived by inverting the Penman-Monteith equation⁵² (Eqs. 1 and 2) from

the eddy covariance measurements at the hourly scale^{18,23,53}, and the averaged value near midday (from 12:00-14:00) was applied as the daily canopy-level stomatal conductance to remove the diurnal cycle. This inversion was only conducted during peak growing season (July and August) to avoid the impact of LAI²³. The impact of evaporation from canopy interception and of low incoming shortwave radiation was removed by data filtering²³, i.e. excluding the data within two days following every precipitation and irrigation event, and periods of low incoming shortwave radiation conditions (<500 Wm⁻²).

(#2) The authors might think more about the structure of the paper. There is a “Results” section, but there are key results discussed prior to this section, including predictions with climate change (L117-L139).

Reply: We agree that the results should include the analysis of the co-regulation prior to this section. We have revised the “Results” to “Contributions to water sustainability” section.

(#3) Related to the point above... I also wonder how the results shown in Fig. 3 and described in lines L117-139 relate to the main results of the paper regarding irrigation schemes? While I think the results in Fig. 3 are really interesting, I think they detract from the main results of the paper. I see the observations and process-based model (Fig. 2) as being clear motivation for creating the plant-centric SDD irrigation model. I think if the paper went from motivation to model creation, it would flow much better. (Maybe Fig. 3 is another paper? Or moved to the SI? Or somehow integrated into the main results of the paper?)

Reply: Referring to the reply to your first comment, the results in Fig. 3 can be treated as the analysis of the reasons for the differences in the size and shape of the co-regulation regimes across 12 sites in Nebraska. The relative importance analysis aimed to quantify the relative contributions of soil moisture and VPD to the variations of *ecosys*-simulated stomatal conductance, with the trend analysis to aridity index (climate properties) and sand fraction (soil properties) in Fig. S4. We have added some connections between the co-regulation pattern and relative importance analysis in the revised manuscript. Please see the revised texts in the manuscript below:

Line 112-118: The differences in the size and shape of the co-regulation regimes mainly resulted from varying climate conditions and soil properties (Fig. 2 and S2). In addition, different cultivars of maize and approaches to obtain stomatal conductance may also cause the differences in Fig. 2. Specifically, the impacts of soil and climate properties on the co-regulation regimes were investigated using *ecosys* model and a relative importance method^{29,30} across 12 sites in Nebraska with a large rainfall gradient (Fig. 3). We further quantified the relative contributions of soil moisture and VPD to the variations of *ecosys*-simulated stomatal conductance at the daily scale.

Minor comments:

Abstract: Very clear. I think some of the sentences may work better in the present tense, e.g., “Specifically, we proposed ...” sounds awkward because you are currently proposing a method in the paper.

Reply: We have revised the past tense of some sentences to the present tense in the abstract.

Introduction:

L58-62, very long sentence, I recommend revising for clarity.

Reply: Thank you for the suggestion. We have revised it. Please see the revised texts in the manuscript below:

Line 58-63: Plant growth is regulated by the balance of water supply and demand in the Soil-Plant-Atmosphere-Continuum (SPAC). Water supply is represented by available water in the soil for plant uptake, while water demand is controlled by atmospheric aridity that passively drives water to move from plants into the atmosphere¹¹⁻¹³. The atmospheric aridity is quantified by vapor pressure deficit (VPD), i.e. the difference between saturated and actual vapor pressures at a given air temperature.

L63, “but” to “and”

Reply: We have revised it.

L63, “, which also has a large impact on plant water relations” is a bit redundant

Reply: We have deleted it.

L66-67, Is it really the interplay between these three things, or the interplay between water supply and atmospheric demand via plant physiological regulation?

Reply: We agree that plant physiological regulation connects soil water supply and atmospheric evaporative demand in the soil-plant-atmosphere-continuum (SPAC). We have revised it to make it clear. Please see the revised texts in the manuscript below:

Line 65-57: Here we argue that plant-centric irrigation schemes are crucial for sustainable irrigation based on the interplay between soil water supply and atmospheric evaporative demand via plant physiological regulations¹⁵ (i.e. plant hydraulics and stomatal response).

Last paragraph, can these three objectives be distilled more (specifically, obj 1 and obj 3)? When first reading this paragraph, it’s hard to get a clear roadmap because of the sub-objectives.

Reply: We have revised this paragraph to make a clear storyline. Please see the revised texts in the manuscript below:

Line 77-82: This study has three objectives: (1) to investigate the co-regulation of soil moisture and VPD on stomatal conductance of maize using field measurements and a validated process-based ecosystem model; (2) to propose a plant-centric irrigation scheme for sustainable irrigation based on the co-regulation pattern; (3) to test and compare the plant-centric irrigation scheme with soil-moisture-based management allowable depletion (MAD) irrigation scheme under current climate and the representative concentration pathway 8.5 (RCP-8.5) scenario.

Co-regulation section:

L102, “Figure 2 shows the co-regulation pattern of <DAILY (?)> soil moisture and VPD...” is it daily? I think the time-scale needs to be mentioned here/in this paragraph/in the paper in general.

Reply: Thank you for the comments and suggestions. Yes, soil moisture and VPD are the daily averaged values. Stomatal conductance is the daily averaged values near midday (from 12:00-14:00) to remove the diurnal cycle. We have added this information in the caption of Fig. 2 (line 143), and Methods in Field measurements (line 327-329) and Advanced process-based model (*ecosys*) (line 365-369). Please see the revised texts in the manuscript below:

Line 143: **Fig. 2 The co-regulation pattern of soil moisture and VPD on stomatal conductance of maize at the daily scale.**

Line 327-329: **Field measurements.** We used two sets of field measurements of soil moisture, VPD, and stomatal conductance of maize at the daily scale to illustrate a proof-of-concept for the co-regulation of soil moisture and VPD on stomatal conductance.

Line 365-369: **Advanced process-based model (*ecosys*).** We further used an advanced process-based agroecosystem model, *ecosys*, to reproduce the co-regulation pattern at the daily scale and to investigate the performances of different irrigation schemes with continuous maize cropping systems across 12 sites in Nebraska with a large rainfall gradient under current climate (2001-2019) and RCP-8.5 scenario (2058-2076) (Fig. 3 and Table S1).

L108, are the greenhouse measurements at the leaf-level?

Reply: Yes. The stomatal conductance of the greenhouse measurements is leaf-level. We have introduced this information in the caption of Fig. 2. Please see the revised texts in the manuscript below:

Line 143-144: **a, b, Measurements of soil moisture, VPD, and leaf-level stomatal conductance (g_s) of maize from greenhouse experiments.**

L113, is the “228 site-years” the same set of data as the “12 sites (e.g. L117). If so, it would be helpful to refer to them using the same language

Reply: Yes. The 228 site-years denotes the 12 sites across Nebraska during the period 2001-2019. Please see the revised texts in the manuscript below:

Line 107-109: **All these observed patterns can be reproduced by a validated hydraulically-driven ecosystem model (*ecosys*) under maize cropping systems across 12 sites in Nebraska (an example site-GD in Fig. 2e, 2f, and Fig. S2) (see Methods).**

L114, do you mean, “The co-regulation pattern indicates that plants can have water stress even at high soil moisture IF VPD is also high”?

Reply: Under the co-regulation pattern, the stomatal conductance of maize decreases with increased VPD given high soil moisture conditions. The reason is that the stomata tends to close when VPD gets higher (larger water demand) to avoid the xylem embolism due to the excessive water flow.

L117, are you analyzing contributions of VPD and SM to daily conductances? I think specifying the time scale is important for a couple reasons... stomata are not responding to daily averages and the physical coupling between VPD and SM is very time scale dependent (with stronger coupling at longer time scales). So the time-scale at which these relationships are resolved is a simplification/assumption, which is fine, but should be stated.

Reply: Referring to your comment on L102, the analysis of the co-regulation using *ecosys* model is at the daily scale. Soil moisture and VPD are daily averaged values, while stomatal conductance is the averaged values near midday (from 12:00-14:00) to remove the diurnal cycle. We have added the information of the time scale in the revised manuscript. Please see the revised texts in the manuscript below:

Line 117-118: We further quantified the relative contributions of soil moisture and VPD to the variations of *ecosys*-simulated stomatal conductance at the daily scale.

L126, Are the relationships in Fig. S3 & S4 statistically significant? Since there are already lines of best fit, it might be useful to add summary statistics to the figures or captions.

Reply: Thank you for the comments and suggestions. We have added the statistics (the correlation coefficient and p-value) in Fig. S3 and S4. It needs to be noted that each point in Fig. S3 denotes the Spearman partial rank correlation coefficients. The p-values of the Spearman partial rank correlation coefficients (all the points) in Fig. S3 were less than 0.001, demonstrating the significant controls from soil moisture and VPD on stomatal conductance. We have added more details of Fig. S3 and S4 in the revised manuscript. Please see the revised texts in the manuscript below:

Line 118-129: All the Spearman partial rank correlation coefficients between stomatal conductance and soil moisture/VPD across 12 sites were significant ($p < 0.001$), demonstrating the significant controls from soil moisture and VPD on stomatal conductance. The estimated relative importance metrics across 12 sites indicated that the contributions of soil moisture to stomatal conductance significantly increased with the aridity index ($p < 0.001$) as there were more limitations from soil water supply to stomatal responses in drier regions (Fig. 3b and S4). The significant positive relationship between the Spearman partial rank correlation coefficients (G_s -Soil moisture) and aridity index ($p < 0.0001$) further confirmed this (Fig. S3a). In contrast, the contributions of VPD to stomatal conductance significantly decreased with the increasing aridity index ($p < 0.0001$) (Fig. 3b and S4), leading to no VPD-dominated regimes at site-T1S1, site-EastBayard, and site-Mitchell (Fig. S2j-S2l).

Fig. S3 Variation of the Spearman partial rank correlation coefficient (p-value of each point was less than 0.001) between G_s and soil moisture/VPD with **a**, aridity index and **b**, sand fraction across 12 sites in Nebraska under current climate (2001-2019). The line denoted the regression line with a 95% confidence interval. r and p denoted the correlation coefficient and p-value, respectively.

Fig. S4 Variation of the relative importance of soil moisture and VPD on G_s with **a**, aridity index and **b**, sand fraction across 12 sites in Nebraska under current climate (2001-2019) and RCP-8.5 scenario (2058-2076). The line denoted the regression line with a 95% confidence interval. r and p denoted the correlation coefficient and p-value, respectively.

Fig. 2, I'm assuming each dot represents one day... is this right?

Reply: Correct. Each dot represents one day as all the data are at the daily scale. The points in Greenhouse experiments were measurements with multiple replicates under different environmental conditions. We have added this information in the caption of Fig. 2 and Field measurements in Methods. Please see the revised texts in the manuscript below:

Line 143: **Fig. 2 The co-regulation pattern of soil moisture and VPD on stomatal conductance of maize at the daily scale.**

Line 336-342: The greenhouse measurements of leaf-level stomatal conductance and soil moisture were performed in approximately 2-week intervals beginning in the vegetative stage and continuing until plant senescence (DOY 198-199, 210-211, 217-218, 233-234, 247), with 11 replicates for each plant under two treatments (WW and WS). The environmental variables, such as relative humidity and air temperature, were continuously measured in minutes. Other detailed experimental setups can be found in Miner and Bauerle (2017).

SDD section:

L184 and Fig. 4, I think the colors should be changed for the water saving and water surcharge zones, since they are in the G_s colormap.

Reply: We have changed the colors of the “water saving” and “water surcharge” zones into green and magenta areas for clarification. Please see the revised texts in the manuscript below:

Line 185-186: SDD differed from MAD in two zones, i.e. the “water saving” zone (green area in Fig. 4a) and the “water surcharge” zone (magenta area in Fig. 4a).

Fig. 4 Schematic diagram of plant-centric irrigation scheme based on water supply-demand dynamics (SDD) at an example site-GD. a, Scatters of soil moisture, VPD, and canopy-level stomatal conductance (G_s) at site-GD were simulations from the *ecosys* model to reproduce the co-regulation pattern. The SDD soil moisture thresholds (blue curve), which were the transition points between water supply (light green region) and demand (light blue region) limitations, varied with VPD. b, the scatters of soil moisture and G_s when VPD equaled 3 kPa with the transition point 3; c, the scatters of soil moisture and G_s when VPD equaled 2 kPa with the transition point 2; and d, the scatters of soil moisture and G_s when VPD equaled 1 kPa with the transition point 1. MAD-55% (olive line), performed as the benchmark, denoted the traditional soil-moisture-based management allowable depletion (MAD) irrigation scheme with constant soil moisture threshold (45% of soil available water holding capacity).

Description of Fig. 4, I think the text should include a conceptual description of the three sub-panels in Fig. 4 (VPD = 1kPa...). This would help explain the SDD model.

Reply: We have added the description of three sub-panels in the captions of Fig. 4. Please see the revised texts in the manuscript below:

Line 198-209: Fig. 4 Schematic diagram of plant-centric irrigation scheme based on water supply-demand dynamics (SDD) at an example site-GD. **a**, Scatters of soil moisture, VPD, and canopy-level stomatal conductance (G_s) at site-GD were simulations from the *ecosys* model to reproduce the co-regulation pattern. The SDD soil moisture thresholds (blue curve), which were the transition points between water supply (light green region) and demand (light blue region) limitations, varied with VPD. **b**, the scatters of soil moisture and G_s when VPD equaled 3 kPa with the transition point 3; **c**, the scatters of soil moisture and G_s when VPD equaled 2 kPa with the transition point 2; and **d**, the scatters of soil moisture and G_s when VPD equaled 1 kPa with the transition point 1. MAD-55% (olive line), performed as the benchmark, denoted the traditional soil-moisture-based management allowable depletion (MAD) irrigation scheme with constant soil moisture threshold (45% of soil available water holding capacity).

Results:

Very nicely written and easy to follow.

Fig. 6b, I was confused with the title of this subplot

Reply: The technological advances, such as improved variety of seeds and fertilizers, and better farming practices, contribute to the enhancement of crop production globally (Najafi et al., 2018). As this manuscript aims to investigate the performance of SDD over traditional MAD irrigation schemes under current and future climates, we do not consider the impacts of the technological advances on crop yields, i.e. technology yield trend, under climate change. We have added more details in the revised manuscript to make it clearer. Please see the revised texts in the manuscript below:

Line 240-245: Without considering the technological advances³³, such as seeds and fertilizer improvements, on crop yields, i.e. no technology yield trend, irrigation water use of MAD irrigation scheme under RCP-8.5 scenario (2058-2076) increases by 16.1%, accompanied with significant reductions in crop yield (-24.5%), economic profit (-54.1%), and irrigation water productivity (-29.2%), when compared with the current climate condition (2001-2019) (Fig. S11).

Line 268-279: Fig. 6 The impacts of climate change on irrigation water use and crop productivity. **a**, The simulated shifts of soil water supply (soil moisture), atmospheric evaporative demand (VPD), and stomatal conductance (G_s) without irrigation under current climate (2001-2019) and over three continuous time periods (2020-2038, 2039-2057, and 2058-2076) under RCP-8.5 scenario. Each point showed the averages of daily soil moisture, VPD, and G_s during the peak growing season (July and August) in each scenario at 12 sites in Nebraska. Relative soil moisture, the ratio (in percentage) of available water to field capacity, was applied to avoid the impact of heterogeneous soil properties at 12 sites. **b**, The performances of the plant-centric SDD and soil-moisture-based MAD irrigation schemes with universal parameters under the RCP-8.5 scenario (2058-2076) (similar to Fig 5) without considering technological advances on crop yields. **c**, The schematic diagram of the main processes by which climate change affects irrigation water use and crop productivity.

Discussion:

L275, “The proposed SDD scheme is pushing the boundaries in scientific irrigation management through applying the above fundamental theories in the Soil-Plant-Atmosphere-Continuum.” I’m not sure this sentence adds anything... I’d delete it.

Reply: We have deleted it.

L296, “Fortunately, we find that the co-regulation pattern of soil moisture and VPD on stomatal conductance for a specific site can be quantified through a data-model fusion method.” I am confused by the “we find”...

Reply: We used “we find” here based on the results of this manuscript and our prior research (Zhang et al., 2021), as the validated *ecosys* model can be used to quantify the co-regulation of soil moisture and VPD on stomatal conductance. We have revised it in the manuscript to make it clearer. Please see the revised texts in the manuscript below:

Line 306-308: Fortunately, the application of *ecosys* model in this research indicates that the co-regulation pattern of soil moisture and VPD on stomatal conductance for a specific site could be quantified through a data-model fusion method¹⁵.

L298-311, I think this section could be written a little differently, with a more speculative tone. As it is written now, it almost sounds like the authors have tested this modeling framework.

Reply: Thank you for the suggestions. The data-model fusion method with BESS-STAIR ET and Noah-MP model has been tested for the reliable simulations of soil moisture (Yang et al., 2020), while other variables have not been tested. We have revised this section with the speculative tone. Please see the revised texts in the manuscript below:

Line 310-323: Specifically, advanced satellite fusion algorithms^{43,44} have been developed to integrate multi-source satellite data to generate accurate field-level and high-frequency (e.g. daily) vegetation data, such as evapotranspiration (ET)^{45,46}, leaf area index (LAI)⁴⁷, and photosynthesis (i.e. gross primary productivity, GPP)^{48,49}. These field-level variables could be helpful to robustly drive and constrain advanced agroecosystem models (e.g. *ecosys* used in this study) to simulate both crop dynamics (e.g. stomatal conductance) and hydrological conditions (e.g. soil moisture)⁵⁰. This approach is a promising direction to enable the implementation of the SDD irrigation scheme at every field based on models constrained by satellite-derived, field-level ecohydrological variables⁵⁰. Additionally, real-time weather forecasts could be applied to provide forecasted ecohydrological variables for the SDD irrigation scheme, allowing producers to initiate irrigation days in advance in order to increase soil water supply and/or to avoid plant water stress due to the anticipated decrease in soil moisture and/or increase in VPD. We thus hope the SDD irrigation scheme could be potentially applied at the field scale and contribute to water sustainability in future.

Methods:

L319, I think a better description of the “greenhouse measurements” is needed. I am not sure the spatial or temporal scale.

Reply: We have added a more detailed description of the greenhouse experiments. Please see the revised texts in the manuscript below:

Line 330-342: The first set was measurements from greenhouse experiments of maize (seed: Dekalb hybrid DKC52-04) at Colorado State University during the 2013 growing season (planted on June 10, 2013)⁵¹. There were 2 treatments (well-watered, WW, and water-stressed, WS) with five plants per treatment. The soil of the greenhouse experiments was the air-dried soilless substrate (8.8 kg) consisting of a 1:1.3 by volume ratio of Greens Grade™, Turface® Quick Dry® and Fafard 2SV in 26 L pots⁵¹. The soil moisture measurements came from soil moisture sensors (Decagon5™ sensors) installed in the middle of the pots (~6 inches from top). The greenhouse measurements of leaf-level stomatal conductance and soil moisture were performed in approximately 2-week intervals beginning in the vegetative stage and continuing until plant senescence (DOY 198-199, 210-211, 217-218, 233-234, 247), with 11 replicates for each plant under two treatments (WW and WS). The environmental variables, such as relative humidity and air temperature, were continuously measured in minutes. Other detailed experimental setups can be found in Miner and Bauerle (2017).

L322, what is the depth of the soil moisture measurements?

Reply: We have added more detailed information of the eddy covariance measurements, and the soil depth of the soil moisture measurements at three AmeriFlux sites was 10-25 cm. Please see the revised texts in the manuscript below:

Line 349-350: The soil moisture data used here was from the top soil layer (10-25 cm).

Reviewer #2 (Remarks to the Author):

The manuscript describes the use of a general ecosystem model (*ecosys*) to simulate impacts of two different irrigation scheduling schemes across a range of climate roughly from eastern Nebraska to western Nebraska. The irrigation scheduling schemes are one based on management allowed depletion (MAD) of soil water with a constant MAD threshold and one based on considering both soil water depletion and stomatal conductance with the soil water depletion threshold (MAD level) varying in concert with simulation stomatal conductance. The schemes are applied using simulation modeling to present data and to future climate and purport to show a difference in outcomes depending on which scheme was used.

Reply: Thank you for providing the comments and suggestions, which are very helpful to improve this manuscript. We have revised the manuscript following your comments and suggestions.

The ecosystem model has been widely used but is not “rigorously validated” as claimed in the abstract. The one cited prior use of the model with irrigated maize in Nebraska was based on one year of data at one location (Grant et al., 2007). The present study uses three locations in Nebraska but yield simulation is biased and inaccurate (r^2 of 0.37) with a regression slope of ~ 0.78 and intercept of ~ 40 bushels per acre.

Reply: Thank you for the comments. We fine tuned two parameters of *ecosys*, including fraction of leaf protein in bundle sheath chlorophyll (CHL4) and plant maturity group (GROUPX), at three AmeriFlux sites (US-Ne1, Ne2, and Ne3) manually during the period from 2001-2012. For 12 sites across Nebraska, we updated the pdf results with the same parameters from the original period 2015-2019 (recent 5 years) to the period 2010-2019 (recent 10 years). Then, we updated the validated performance of *ecosys* model with maize-cropping systems at three AmeriFlux sites (US-Ne1, Ne2, and Ne3) during the period from 2001-2012 and 12 sites across Nebraska during the period from 2010-2019 (Table S5, Fig. S12-15). The R^2 of crop yields at three AmeriFlux sites (US-Ne1, Ne2, and Ne3) were 0.50 during the period 2001-2012. The regression line of maize yields is close to a 1-to-1 relationship. In addition, we have deleted the word “rigorously” in the whole manuscript. Please see the revised texts in the manuscript below:

Line 413-424:

(2) **Model setup:** *ecosys* was validated at three AmeriFlux sites (US-Ne1, Ne2, and Ne3, <https://ameriflux.lbl.gov/>)⁶² and 12 sites across Nebraska (Table S5, Fig. S12-S15).

Three AmeriFlux sites (US-Ne1, Ne2, and Ne3) had the complete data from 2001 to 2012 to test model performance, including the meteorological variables (such as surface air temperature, downward shortwave radiation, wind speed, and precipitation), eddy-covariance fluxes (such as GPP, net ecosystem exchange-NEE, and LE) from the FLUXNET2015 dataset⁶⁶ (<http://fluxnet.fluxdata.org/data/fluxnet2015-dataset/>) and detailed ground-based crop growth observations (i.e. planting/harvest date, irrigation/fertilization records, LAI, and yield) from the Carbon Sequestration Program (CSP) at University of Nebraska-Lincoln’s Agricultural Research and Development Center (<http://csp.unl.edu/Public/sites.htm>)⁶⁷. The daily and monthly model simulations of GPP, ET, LAI, and yield matched very well with the eddy-covariance and ground-based observations (Table S5, Fig. S12-S15).

Line 438-454: The soil information (i.e. bulk density, field capacity, wilting point, soil texture, saturated hydraulic conductivity, soil organic carbon, pH, and cation exchange capacity) were acquired from the Gridded Soil Survey Geographic Database (gSSURGO) dataset⁷¹. There were 12

soil layers (the depth of the bottom for each layer: 0.01m, 0.05m, 0.1m, 0.15m, 0.18m, 0.28m, 0.35m, 0.59m, 0.92m, 1.32m, 1.6m, 2.00m) with a maximum depth of 2.0 m. The planting date of the continuous maize cropping systems at the 12 sites was obtained from the USDA NASS weekly Crop Progress Reports (2001-2019) with the fertilizer (18g N/m²/year and 5g P/m²/year) applied two days before planting, and the crops were harvested on October 31. Other land management practices were set as the same across the 12 irrigated sites in Nebraska, including planting density (8.4 plants/m²), tillage practice (no tillage), and crop type (continuous maize cropping systems). The auto-irrigation scheme in *ecosys* with the widely used soil-based MAD-50%⁷² was applied to determine the irrigation scheduling at 12 sites to test the model performance. The NASS county-level irrigated maize yield (2010-2019) at the counties where the 12 sites were located was used as the observations. The probability density function with gaussian kernel density estimation of irrigated yields from *ecosys* model at 12 sites showed good agreement with that of the National Agricultural Statistics Service (NASS) county-level irrigated yields during the period from 2010-2019 (Fig. S15h), which further validated the reliability of the *ecosys* model.

Table S5. The statistics indexes of *ecosys* simulated daily and monthly GPP, ET, LAI, and yield with flux towers/CPS observations with the maize cropping systems at three AmeriFlux sites (US-Ne1, US-Ne2, US-Ne3) during the growing seasons (June to September) of 2001-2012.

Variables	GPP					ET					LAI					Yield				
Indexes	RMSE (gC/m ²)	NRMSE (%)	Bias (gC/m ²)	NBias (%)	R ²	RMSE (mm)	NRMSE (%)	Bias (mm)	NBias (%)	R ²	RMSE (m ² /m ²)	NRMS E (%)	Bias (m ² /m ²)	NBias (%)	R ²	RMSE (t/ha)	NRMS E (%)	Bias (t/ha)	NBias (%)	R ²
Daily	3.82	29.6	0.26	2.0	0.79	1.04	27.7	-0.47	-12.5	0.73	1.10	32.6	-0.56	-16.6	0.74	1.31	13.2	0.12	1.2	0.50
Monthly	92.30	24.1	7.50	2.0	0.82	21.61	19.2	-14.14	-12.6	0.80	1.02	32.0	-0.58	-18.2	0.77					

* o_i and s_i are the observations and model simulations, respectively; RMSE is the Root Mean Square Error ($RMSE = \sqrt{\frac{1}{N} \sum_{i=1}^N (s_i - o_i)^2}$); NRMSE

is the Normalized Root Mean Square Error ($NRMSE = \frac{\sqrt{\frac{1}{N} \sum_{i=1}^N (s_i - o_i)^2}}{\bar{o}_i}$); Bias is the mean bias ($Bias = \frac{1}{N} \sum_{i=1}^N (s_i - o_i)$); NBias is the

normalized mean bias ($NBias = \frac{\frac{1}{N} \sum_{i=1}^N (s_i - o_i)}{\bar{o}_i}$); R^2 is the coefficient of determination ($R^2 = 1 - \frac{\sum_{i=1}^N (s_i - o_i)^2}{\sum_{i=1}^N (s_i - \bar{o}_i)^2}$).

Fig. S12 Performance of *ecosys* model on daily GPP, NEE, LE, and LAI with the statistical results (RMSE, Bias, and R²) for continuous maize cropping systems during the period 2001-2012 at US-Ne1.

Fig. S13 Performance of *ecosys* model on daily GPP, NEE, LE, and LAI with the statistical results (RMSE, Bias, and R²) for maize (yellow regions) and soybean (green regions) cropping systems during the period 2001-2012 at US-Ne2.

Fig. S14 Performance of *ecosys* model on daily GPP, NEE, LE, and LAI with the statistical results (RMSE, Bias, and R²) for maize (yellow regions) and soybean (green regions) rotation cropping systems during the period 2001-2012 at US-Ne3.

Fig. S15 The performance of *ecosys* model with maize cropping systems at three AmeriFlux sites (US-Ne1, Ne2, and Ne3) during the period from 2001-2012 and 12 sites with a dramatic rainfall gradient across Nebraska during the period from 2010-2019. The colorbar shows the normalized gaussian kernel density estimation of the scatters. Black dashed lines indicate the 1-to-1 relationship. The red line is the regression line with the slope and intercept. The probability density function of the maize yields at the 12 sites is the gaussian kernel density estimation.

The paper makes claims of innovation and being the first to apply the theory of plant water relations to irrigation management, including co-regulation, but this is not so. The paper ignores a plethora of earlier work, some ongoing, that recognized and utilized both soil water deficit and plant response to soil water deficit and atmospheric demand (see specific comments).

Reply: Thank you for the comments. Current irrigation scheduling research mainly includes conventional soil-based (such as MAD using neutron probe), plant-based (such as CWSI and TTT based on canopy temperature), and soil-plant hybrid (such as the ISSCADA-hybrid) irrigation scheduling. We agree that the plant-based and soil-plant hybrid irrigation scheduling considers the impacts of both soil water supply and atmospheric evaporative demand. The plant-based irrigation scheduling determines the irrigation decisions based solely on plant responses, while the plant-soil hybrid irrigation scheduling determines the irrigation decisions based on the plant and soil feedbacks. The difference between the proposed supply-demand dynamics (SDD) irrigation scheduling and the existing plant-based/hybrid scheduling is that SDD determines the dynamic soil moisture thresholds, varying with VPD, based on the co-regulation of soil moisture and VPD on stomatal conductance. Once the dynamic threshold level is determined, it can be applied for irrigation scheduling directly with soil moisture and VPD measurements, without the need of considering the plant feedback. We have added some discussion and related references in the Discussion section. Please see the revised texts in the manuscript below:

Line 82-85: The innovation of this study is to apply the co-regulation pattern into irrigation management, and we find the new method has demonstrated a large improvement over the existing soil-moisture-only irrigation metrics and thus could have significant contributions to water sustainability.

Line 282-289: Our study proposed and implemented the plant-centric SDD irrigation scheme based on the plant water supply-demand dynamics, i.e. the co-regulations of soil moisture and VPD on stomatal conductance. This co-regulation mechanism has been widely demonstrated by observational evidence^{11,12,18,23,24}, but has seldom been applied in irrigation management directly³⁵. Although some newly developed irrigation practices were plant-based and/or plant-soil hybrid^{35,36}, such as evapotranspiration (ET)-based^{37,38} and canopy temperature-based^{35,39} irrigation, the plant-centric SDD irrigation scheme based on supply-demand dynamics was the first application in leveraging the co-regulation mechanism from plant physiology.

The total reliance on evapotranspiration (ET) sensed by eddy covariance presents the results from being definitive because of the widely recognized tendency of eddy covariance methods to underestimate ET by 15% to 30%. There appear to be no mass balance-based ET data used, whether from weighing lysimetry or from deep soil water content sensing such as with the neutron probe.

Reply: Thank you for the comments. The eddy covariance ET measurements at US-Ne1, Ne2, and Ne3 were obtained from FLUXNET2015 Dataset (<https://fluxnet.org/data/fluxnet2015-dataset/>), which has been adjusted for energy balance closure (Pastorello et al., 2020). In addition, the eddy covariance flux measurements at US-Ne1, Ne2, and Ne3 implied a fairly good closure of the energy balance (Suyker and Verma, 2009), which is further confirmed by our own calculation of the energy closure. Thus, we believe the ET eddy covariance dataset at US-Ne1, Ne2, and Ne3 are reliable.

The data used to develop stomatal conductance relationships with soil water content and vapor pressure deficit were all from three sites within a 14-km radius near Mead, Nebraska in far eastern Nebraska. US-Ne3 is a rainfed site at Mead where the cropping is a maize-soybean rotation. US-Ne1 and US-Ne2 are also near Mead but feature irrigated maize-soybean. None of the data are from the more arid sites and different soils that would be found in central or western Nebraska. Therefore, projections of the co-regulation across the eleven other sites stretching from eastern to western Nebraska appears to be based entirely on computer simulation. To be sure, the methods leave this point somewhat obscurely described.

Reply: Referring to the model performance shown before, we believe that the *ecosys* model can be reliable for this study. We have added more details about how the co-regulation of soil moisture and VPD on stomatal conductance is developed in the Methods. Please see the revised texts in the manuscript below:

Line 466-472: The field measurements of soil moisture, VPD, and stomatal conductance at the daily scale from the greenhouse experiments and three AmeriFlux sites (US-Ne1, Ne2, and Ne3) were applied to investigate the co-regulation of soil moisture and VPD on stomatal conductance (Eq. 8) as observational evidences. With the validated *ecosys* model at three AmeriFlux sites (US-Ne1, Ne2, and Ne3) and 12 sites across Nebraska (Table S5, Fig. S12-15), the model simulations of daily soil moisture, VPD, and canopy-level stomatal conductance during the peak growing season (July and August) were applied to investigate the co-regulation patterns across 12 sites in Nebraska.

The stomatal conductance was derived from inversion of the “Penman-Monteith equation” using ET from eddy covariance data at the three eastern Nebraska sites but the version of the Penman-Monteith formulation used is not referenced and details of the inversion are not given. It is important to give details of this inversion because stomatal conductance changes with VPD, solar irradiance, and leaf water potential and because the specific formulation of Penman-Monteith used can affect the results. How were leaf water potential and solar irradiance factored in? The Penman-Monteith formulation allows for solar irradiance but not for leaf water potential. This is far from an obvious inversion although it has been done before. Lascano and others have demonstrated a two-source (canopy and soil surfaces) implicit ET model and have shown how it can be inverted to compute stomatal conductance (Lascano et al., 2010a,b). Because it computes canopy and air temperatures separately, the Lascano et al. approach is more accurate than is inversion of the Penman-Monteith formulation under conditions where canopy and air temperatures vary substantially, which is the case in more arid, and highly irrigated regions. Because the Penman-Monteith assumes equality of air and canopy temperatures, inverting the Penman-Monteith will give approximately the same results as an implicit solution in a more humid climate (e.g., eastern Nebraska) where air and canopy temperatures tend to be similar but will give a different and less accurate stomatal conductance in regions where VPD and canopy-air temperature differences are both greater (e.g., western Nebraska). In addition, there is some evidence that eddy covariance bias increases in more arid environments, which would add more to the bias of the results of this study.

Reply: Thank you for introducing **the recursive evapotranspiration method** for the possible application in this research (Lascano and Evett, 2010; Lascano et al., 2010). Let us clarify different approaches to obtain the co-regulation pattern at three AmeriFlux sites (US-Ne1, Ne2, and Ne3 in eastern Nebraska) and 12 sites across Nebraska.

For three AmeriFlux sites (US-Ne1, Ne2, and Ne3), the canopy-level stomatal conductance was obtained by the inversion of the Penman-Monteith equation. As you mentioned, the inversion of the Penman-Monteith equation gave pretty good estimations in more humid regions (e.g. eastern Nebraska). Thus, we believed that the inverted canopy-level stomatal conductance at three AmeriFlux sites (US-Ne1, Ne2, and Ne3) using the Penman-Monteith equation was reliable.

For 12 sites across Nebraska, as there were no eddy covariance measurements for the sites located at central and western Nebraska, the canopy-level stomatal conductance at 12 sites from eastern to western Nebraska was the model simulation from the *ecosys* model, rather than inverting the Penman-Monteith equation. As we have described before, *ecosys* is a mechanistic ecosystem model with fully connected balances and interactions for water, energy, carbon and nutrient cycles in the soil-vegetation-atmosphere continuum.

We have added more details of the inversion of the Penman-Monteith equation in the Methods section. Solar radiation was considered, while leaf water potential was not considered in the inverted Penman-Monteith equation. This approach has been widely used to obtain canopy-level stomatal conductance (Kimm et al., 2020; Novick et al., 2016; Xiao et al., 2020). Please see the revised texts in the manuscript below:

Line 343-363:

The second set was eddy covariance measurements of maize cropping systems (seed: Pioneer 33P67/33B51) from 2001 to 2012 at three AmeriFlux sites (US-Ne1, Ne2, and Ne3). US-Ne1 and Ne2 were irrigated sites, with a continuous maize cropping system during 2001-2012 for US-Ne1 and with a maize-soybean rotation cropping system during 2001-2009 and then a continuous maize cropping system during 2010-2012 for US-Ne2. US-Ne3 was rainfed with a maize-soybean rotation cropping system during 2001-2012. The soil at three AmeriFlux sites was a deep silty clay loam consisting of four soil series: Yutan, Tomek, Filbert, and Filmore. The soil moisture data used here was from the top soil layer (10-25 cm). The canopy-level stomatal conductance (G_s) was derived by inverting the Penman-Monteith equation⁵² (Eqs. 1 and 2) from the eddy covariance measurements at the hourly scale^{18,23,53}, and the averaged value near midday (from 12:00-14:00) was applied as the daily canopy-level stomatal conductance to remove the diurnal cycle. This inversion was only conducted during peak growing season (July and August) to avoid the impact of LAI²³. The impact of evaporation from canopy interception and of low incoming shortwave radiation was removed by data filtering²³, i.e. excluding the data within two days following every precipitation and irrigation event, and periods of low incoming shortwave radiation conditions ($<500 \text{ Wm}^{-2}$).

$$\lambda E = \frac{\Delta(R_n - G) + \rho c_p g_a VPD}{\Delta + \gamma(1 + g_a/G_s)} \quad (1)$$

$$G_s = g_a \gamma / \left\{ \frac{\Delta(R_n - G) + \rho c_p g_a VPD}{\lambda E} - (\Delta + \gamma) \right\} \quad (2)$$

where Δ is the slope of the water vapor deficit; R_n and G are net radiation and soil heat flux, respectively; ρ is the air density; c_p is the specific heat capacity for dry air; g_a is aerodynamic conductance; γ is the psychrometric constant; and λE is evapotranspiration.

There appears to be a disconnect between the source of ET data at the 12 sites across Nebraska (site-specific eddy covariance data) and the source of forcing meteorological data, which was not site-specific but came from NLDAS-2 and was therefore interpolated.

Reply: As there were no eddy covariance measurements for the sites located at central and western Nebraska, the canopy-level stomatal conductance at 12 sites from eastern to western Nebraska was the model simulation from *ecosys*, rather than inverting the Penman-Monteith equation with the existing ET and other meteorological data.

The results indicate that irrigation would “not resolve the yield loss from rising temperatures” but considering corn yields in places with much greater VPD and higher temperatures, one wonders how realistic these simulations are. For example, In the northwest corner of the Texas Panhandle, county wide corn yields approach 300 bushels per acre and are better than those in much of the corn belt. This is in a region with intensive irrigation due to its hot, semiarid to arid climate, and lack of precipitation. In this region, irrigation quite adequately resolves yield loss under conditions that are already severe.

Reply: Thank you for the comments. Let us introduce more about the setting of *ecosys* model under climate change. The incremental change scheme without the acclimation of parameters was selected in *ecosys* model with the ensemble predictions of air temperature, precipitation, and [CO₂] from 15 Coupled Model Intercomparison Project phase 5 (CMIP5) models under Representative Concentration Pathway 8.5 (RCP-8.5). All the parameters of *ecosys* model under RCP-8.5 (2058-2076) kept the same as those under current climate (2001-2019). Thus, the growing season under RCP-8.5 would be shortened with earlier anthesis date and shorter grain-filling period as higher temperature under RCP-8.5 speeds maturation, resulting in maize yield decline under climate change.

As you mentioned, maize in the Texas Panhandle could maintain high yield under hot, semiarid to arid climate due to intensive irrigation. There are several reasons for discussion: (1) the seeds used in Texas may be adapted to the hot, semiarid to arid climate, while the *ecosys* simulations under climate change used the same phenology parameters without acclimation from current climate, i.e. no genetic advances considered; (2) the existed cooling effects due to intensive irrigation in the field experiments was not considered in the *ecosys* model simulations under climate change; (3) the irrigation scheme with MAD under climate change was the same as that under current climate, i.e. no irrigation practice improvements considered.

Although we could apply the acclimation of parameters in *ecosys* under climate change to maintain the high maize yields under climate change, we think that the setting of the *ecosys* model without the acclimation of parameters under climate change is more reasonable. The reason is that we aimed to investigate the impacts of environmental variables under climate change, rather than other technological advances.

The use of a constant threshold across soil types and throughout the season as is done here is a simplistic and unrealistic way to use the MAD concept in irrigation management. Early season irrigation management and late season irrigation management typically do not strictly follow the MAD concept for good reasons. Reasons include holding off on early irrigation to force root growth into deeper soil layers and reducing irrigation during vegetative growth (by adjusting the MAD level). Also, the depth of soil considered in computing soil moisture varies with root growth in sophisticated irrigation management and may be greater than the 0.92 m used in this study.

Reply: Thank you for the comments and suggestions.

(1) We fully agree that the traditional MAD irrigation scheme could be optimized through many settings, such as the growing stage-specific MAD. As a matter of fact, the MAD threshold level presented in the current study was already optimized by maximizing the economic profits as the benchmarks. We admit that the constant MAD level is a simple approach for testing.

Furthermore, since the optimal MAD threshold is not the focus of this study, the major goal of this study is to incorporate the acknowledged co-regulation pattern in plant physiology into irrigation management, and to assess its efficacy for irrigation guidance. The proposed supply-demand dynamics (SDD)'s threshold level varied with VPD, and we compared its performance with the already optimized MAD level to investigate the benefits of incorporating the co-regulation patterns into irrigation.

(2) For the soil depth, there are 12 soil layers (the depth of the bottom for each layer: 0.01m, 0.05m, 0.1m, 0.15m, 0.18m, 0.28m, 0.35m, 0.59m, 0.92m, 1.32m, 1.6m, 2.00m) with a maximum depth of 2.0 m in *ecosys*. Soil moisture for irrigation is the weighted average of soil water content in different soil layers within the root-zone, which is varying with dynamic root-growth. Crops could absorb water from deeper roots in the soil, but it takes a long time for water transport from deeper roots to reach the stem and then leaves. Thus, we add the constraint of the maximum depth of 0.92 m to reduce the impacts of deep wet soil. We have added more details of the soil moisture used for irrigation. Please see the revised texts in the manuscript below:

Line 440-442: There are 12 soil layers (the depth of the bottom for each layer: 0.01m, 0.05m, 0.1m, 0.15m, 0.18m, 0.28m, 0.35m, 0.59m, 0.92m, 1.32m, 1.6m, 2.00m) with a maximum depth of 2.0 m.

Line 487-490: Soil moisture was the weighted average of soil water content in different soil layers within the root-zone, which was varying with dynamic root-growth but no more than the top 9 soil layers with a depth of 0.92 m to reduce the impacts of deep wet soil.

The description of the applicability of the SDD irrigation scheme is all in the realm of possibility but have not been demonstrated in real-world tests so are conjecture. The difficulty in applying a complex process-based simulation model such as *ecosys* inhibits even research scientists from using such models, not to mention irrigation managers. Throughout, the manuscript tends to certainty when the subjunctive tense would be more appropriate. The description of things as rigorous when they are susceptible to bias is not acceptable. A more humble stance would get the authors further, particularly because the study is primarily a modeling exercise and does not test the approach by applying it to irrigation management in real fields.

Reply: Thank you for the comments and suggestions. We are now doing a 3-year field experiment to test SDD irrigation scheme at two sites in Nebraska starting from this year (one in eastern at Mead and one in central-western Nebraska). Though we are not reporting real-world testing results of SDD in the current manuscript, we believe our study on combining soil water supply and atmospheric evaporative demand for sustainable irrigation based on the validated hydraulically-driven ecosystem model, *ecosys*, is rigorous and has its unique and broader value to the community. In addition, we have added more results about the validated performance of the advanced agroecosystem model, *ecosys*, to show its reliability. We have also revised the description of the simulated results with the subjunctive tense. Please check the revised manuscript.

The difficulty of the application of the SDD irrigation scheme is to determine the dynamic threshold level, which varies with VPD based on the co-regulation of soil moisture and VPD on stomatal conductance. Once the dynamic threshold level of SDD was determined, SDD can be tested for irrigation with the measurements of soil moisture and VPD. As explained in the Discussion section, data-model fusion is a promising direction to achieve this goal.

SPECIFIC COMMENTS:

L1-2: Title is difficult to understand.

Reply: Let us clarify the title. The soil-plant-atmosphere-continuum (SPAC) connects water supply from soil, water demand from atmosphere, and plant self-regulation, such as stomata response. Soil water supply denotes soil moisture available for root water uptake, while atmospheric evaporative demand represents the water demand in the atmosphere, quantified by VPD. This manuscript investigated the co-regulation of soil moisture and VPD on stomatal conductance, and proposed a new supply-demand dynamics (SDD) irrigation scheme. The results indicated that the new proposed SDD irrigation scheme could save irrigation water use without penalizing crop yields to relieve water scarcity, i.e. sustainable irrigation compared with traditional MAD. We have revised the title to make it clearer.

Title: Sustainable irrigation based on co-regulation of soil water supply and atmospheric evaporative demand

L34: Based on the contents of the manuscript, I do not think it is defensible to refer to the model as “rigorously validated”.

Reply: We have deleted the word “rigorously” in the whole manuscript. Referring to the reply to your main comment about *ecosys*, we have updated the validated performance of *ecosys* model with maize-cropping systems at three AmeriFlux sites (US-Ne1, Ne2, and Ne3) during the period from 2001-2012 and 12 sites across Nebraska during the period from 2010-2019 (Table S5, Fig. S12-15). The R^2 of crop yields at three AmeriFlux sites (US-Ne1, Ne2, and Ne3) were 0.50 during the period 2001-2012. The regression line of maize yields is close to a 1-to-1 relationship.

L73-73: This is a meaningless statement. One could just as easily write the converse, which is that in some regions, like the U.S. corn belt, constraints from soil water deficit on crop growth can be even larger than that from VPD. Suggest deleting it.

Reply: We have deleted it.

L78-79: The co-regulation of stomatal conductance by soil water availability and evaporative demand is well known and does not need verification as stated in the first objective. What needs verification is the use of the process-based ecosystem model in this context. The model was not designed with irrigation management in mind. Recommend rephrasing this objective statement.

Reply: We have revised the statements of the objectives. Please see the revised texts in the manuscript below:

Line 77-82: This study has three objectives: (1) to investigate the co-regulation of soil moisture and VPD on stomatal conductance of maize using field measurements and a validated process-based ecosystem model; (2) to propose a plant-centric irrigation scheme for sustainable irrigation based on the co-regulation pattern; (3) to test and compare the plant-centric irrigation scheme with soil-moisture-based management allowable depletion (MAD) irrigation scheme under current climate and the representative concentration pathway 8.5 (RCP-8.5) scenario.

L87-90: This is not the first such effort at applying the theory of plant water relations to irrigation management. Earlier work exists that employs a combined soil water sensing and modeling approach to irrigation management. Suggest you discuss that work, including:

Lascano, R.J. 2000. A general system to measure and calculate daily crop water use. *Agronomy J.* 92, 5, 821-832. <https://doi.org/10.2134/agronj2000.925821x>

Lascano, R. J., Baumhardt, R. L., Hicks, S. K., Evett, S. R., and Heilman, J. L. 1996. Daily measurement and calculation of crop water use. pp. 225-230. In C. R. Camp, E. J. Sadler, and R. E. Yoder (eds.) *Proc. International Conference. Evapotranspiration and Irrigation Scheduling*, San Antonio, TX.

Reply: Thank you for providing these important references. Current irrigation scheduling research mainly includes conventional soil-based (such as MAD using neutron probe), plant-based (such as ET-based and CWSI/TTT based on canopy temperature), and soil-plant hybrid (such as the ISSCADA-hybrid) irrigation scheduling. We agree that the plant-based and soil-plant hybrid irrigation scheduling considers the impacts of both soil water supply and atmospheric evaporative demand. The plant-based irrigation scheduling determines the irrigation decisions based solely on plant responses, while the plant-soil hybrid irrigation scheduling determines the irrigation decisions based on the plant and soil feedbacks. The difference between the proposed supply-demand dynamics (SDD) irrigation scheduling and the existing plant-based/hybrid scheduling is that SDD determines the dynamic soil moisture thresholds, varying with VPD, based on the co-regulation of soil moisture and VPD on stomatal conductance. Then, the dynamic soil moisture thresholds can be applied for irrigation scheduling directly with soil moisture and VPD measurements, without the need of considering the plant feedback. We have added this discussion and references in the Discussion sections. Please see the revised texts in the manuscript below:

Line 82-85: The innovation of this study is to apply the co-regulation pattern into irrigation management, and we find the new method has demonstrated a large improvement over the existing soil-moisture-only irrigation metrics and thus could have significant contributions to water sustainability.

Line 282-289: Our study proposed and implemented the plant-centric SDD irrigation scheme based on the plant water supply-demand dynamics, i.e. the co-regulations of soil moisture and VPD on stomatal conductance. This co-regulation mechanism has been widely demonstrated by observational evidence^{11,12,18,23,24}, but has seldom been applied in irrigation management directly³⁵. Although some newly developed irrigation practices were plant-based and/or plant-soil hybrid^{35,36}, such as evapotranspiration (ET)-based^{37,38} and canopy temperature-based^{35,39} irrigation, the plant-centric SDD irrigation scheme based on supply-demand dynamics was the first application in leveraging the co-regulation mechanism from plant physiology.

Figure 1: Suggest adding an arrow from the rhizosphere to the "a" box. Otherwise, the caption does not completely jive with the figure.

Reply: We have revised the Figure 1 following your suggestion.

Fig. 1 Conceptual schemes of two methods to quantify plant water stress. a, Plant-centric: regulating stomatal conductance (g_s) by considering both soil water supply and atmospheric evaporative demand. **b,** Soil-moisture-based: management allowable depletion (MAD) solely considering soil water supply.

L143-144: Give details of how stomatal conductance was derived from the Penman-Monteith equation

Reply: We have added the detailed information about the inversion of the Penman-Monteith equation to obtain the canopy-level stomatal conductance in Methods section (Line 343-363, shown before).

L157: The concept of potential evapotranspiration ceased to be used in irrigation science some decades ago in favor of the concept of reference evapotranspiration. Is the "potential evapotranspiration" written here supposed to be reference evapotranspiration? If it was computed using Penman-Monteith (e.g., the ASCE 2005 Penman-Monteith standard) then it is reference ET, not potential ET.

Reply: We used the Priestley-Taylor equation to calculate the potential evapotranspiration at the daily scale. We have added the reference for the Priestley-Taylor equation to avoid being misleading. Please see the revised texts in the manuscript below:

Line 157-159: Aridity index was calculated as the ratio of potential evapotranspiration³¹ (PET, mm) and precipitation (P, mm) during growing season from 2001 to 2019, i.e. PET/P.

L169 and Figure 4: I find figure 4 difficult to interpret. The labels for SDD: part 1 and SDD: part 2 do not seem to be adequately explained.

Reply: We have revised the interpretation of the SDD soil moisture threshold, Fig. 4 and its caption to make it clearer. Please see the revised texts in the manuscript below:

Line 168-177: Taking site-GD in Nebraska as an example, the critical stomatal conductance was determined to be 0.007 m/s (see Methods and Table S2). The first part of the SDD soil moisture thresholds, varying with VPD, were determined using the fitted contours of the critical stomatal conductance in the co-regulated regime until entering the VPD-dominated regime, i.e. the blue contour of critical $G_s = 0.007$ m/s (part 1 in Fig. 4a). Stomatal conductance was dominated by VPD in the VPD-dominated regime, as soil moisture had little effect on stomatal conductance. Thus, the other part of the SDD soil moisture threshold was determined as the boundary of the VPD-dominated regime, i.e. the constant blue MAD threshold with 35% (part 2 in Fig. 4a). Consequently, irrigation was triggered when soil moisture got lower than the SDD soil moisture threshold under specified VPD conditions.

Fig. 4 Schematic diagram of plant-centric irrigation scheme based on water supply-demand dynamics (SDD) at an example site-GD. a, Scatters of soil moisture, VPD, and canopy-level stomatal conductance (G_s) at site-GD were simulations from the *ecosys* model without irrigation impacts to reproduce the co-regulation pattern. The SDD soil moisture thresholds (blue curve including part 1, i.e. contour of critical $G_s = 0.007$ m/s, and part 2, i.e. the constant blue MAD

threshold with 35%), which were the transition points between water supply (light green region) and demand (light blue region) limitations, vary with VPD. **b**, the scatters of soil moisture and G_s when VPD equaled 3 kPa with the transition point 3; **c**, the scatters of soil moisture and G_s when VPD equaled 2 kPa with the transition point 2; and **d**, the scatters of soil moisture and G_s when VPD equaled 1 kPa with the transition point 1. MAD-55% (olive line), performed as the benchmark, denoted the traditional soil-moisture-based management allowable depletion (MAD) irrigation scheme with constant soil moisture threshold (45% of soil available water holding capacity).

L175-179: What is innovative is the use of a process-based model to set the varying threshold. It has long been recognized that water use rate should impact the soil water content threshold that is used to trigger irrigation. This was the case when the neutron probe was used for irrigation management. Suggest rephrasing and omitting the statement that MAD based irrigation scheduling uses a constant threshold. That is a simplistic view of MAD based irrigation management.

Reply: Thank you for the comments and suggestions. We have revised this part following your suggestions. Please see the revised texts in the manuscript below:

Line 177-184: The innovation of the SDD irrigation scheme was the dynamic soil moisture threshold based on the co-regulation of soil moisture and VPD on stomatal conductance. We compared its performance with a simple conventional soil-moisture-based irrigation scheme (MAD-55%, olive line in Fig. 4a). All parameters of the SDD (i.e. critical stomatal conductance under low VPD conditions and the soil moisture threshold under high VPD conditions) and MAD (i.e. soil moisture threshold) irrigation schemes were optimized to be either site-specific or universal for all the sites using simulated profit, which took total irrigation cost and economic gain from crop yields into account (see Methods, Fig. S6 and Table S2, S3).

L208: Shouldn't this be "simulated" not "tested"? Rephrasing to "We systematically simulated application and outcomes of the SDD and MAD irrigation schemes..." might be more to the point. A test would involve comparing the simulated results with some objective measure. There does not appear to be an objective measure that was obtained at any of the 12 sites. Indeed, the present study appears not to involve any actual test in the field.

Reply: We have revised it following your suggestions. Please see the revised texts in the manuscript below:

Line 212-214: We systematically simulated application and outcomes of the SDD and MAD irrigation schemes at 12 sites in Nebraska under current (2001-2019) and future (RCP-8.5, 2058-2076) climate conditions.

L220: Suggest removing "Besides". It does not seem to serve a purpose in this exposition.

Reply: We have deleted it.

L218-220: I do not see that this statement follows from the previous one. Why is the word "Thus" used here?

Reply: "Thus" here follows the previous two sentences. The first sentence described the benefits of SDD over MAD based on this study, and the second one described the MAD is already highly

efficient. Then, we came to our sentence “Thus the significant benefit of SDD over MAD demonstrated here provided a testimony for the improved performance of SDD”.

L223-224: This phrase is confusing. It appears that it is incompletely worded.

Reply: We have revised it. Please see the revised texts in the manuscript below:

Line 227-231: The absolute differences in irrigation water use and irrigation water productivity between the SDD and MAD irrigation schemes significantly decreased with increasing aridity index ($p < 0.05$) and also slightly decreased with increasing sand fraction; while the difference in crop yields between SDD and MAD maintained stable and negligible with aridity index and sand fraction.

L233-237: The climate change scenarios for Nebraska that I have seen do not predict such large changes in irrigation demand.

Reply: Thank you for the comment. Let us introduce more about the setting of *ecosys* model under climate change. The incremental change scheme without the acclimation of parameters was selected in *ecosys* model with the ensemble predictions of air temperature, precipitation, and [CO₂] from 15 Coupled Model Intercomparison Project phase 5 (CMIP5) models under Representative Concentration Pathway 8.5 (RCP-8.5). All the parameters of *ecosys* model under RCP-8.5 (2058-2076) kept the same as those under current climate (2001-2019). Thus, the growing season under RCP-8.5 would be shortened with earlier anthesis date and shorter grain-filling period as higher temperature under RCP-8.5 speeds maturation, resulting in maize yield decline under climate change.

The original setting of the irrigation season in the *ecosys* model was the period from planting date to harvest date, with no adjustments on the crop conditions. As the growing season under RCP-8.5 was shortened, we should also shorten the irrigation season in the *ecosys* model under RCP-8.5 based on the crop conditions. The simulated results under RCP-8.5 have been updated (Fig.6 and S11). Based on the updated results, the irrigation demand increased by 16.1% (28 mm) in Nebraska from 2001-2019 to 2058-2076 under RCP-8.5 scenario.

We have checked some references on the irrigation changes in Nebraska under climate change. For example, the irrigation demand increased 15% (~25 mm) under the high warming scenarios (RCP 8.5, +2.4 °C) from 1990 to 2050 in the northern High Plains aquifer, Platte River Basin, central Nebraska (Lauffenburger et al., 2018). Thus, we believe that the adjusted results under RCP-8.5 should be reasonable.

We have added more description on the setting of irrigation and climate change in the *ecosys* model. We have also updated all the results in the revised manuscript under RCP-8.5 scenario. Please see the revised texts in the manuscript below:

Line 484-492: For the irrigation setting in the *ecosys* model, irrigation was triggered when soil moisture was lower than soil moisture threshold of SDD (varying with VPD) and MAD (constant) at the daily scale during the growing season. Soil moisture was the weighted average of soil water content in different soil layers within the root-zone, which was varying with dynamic root-growth but no more than the top 9 soil layers with a depth of 0.92 m to reduce the impacts of deep wet soil.

Irrigation amount was determined by water required to fill current soil water content to field capacity, ignoring the constraints from irrigation infrastructures. Each irrigation event (last for 24 hours) was incorporated into the *ecosys* model at a daily time step in real time.

Line 434-437: *ecosys* model simulation under current climate (2001-2019) was extended through three additional 19-year cycles from 1 January 2020 to 31 December 2076 under the RCP-8.5 scenario using the incremental change scheme without the acclimation of parameters in *ecosys*, and the fourth cycle 2058-2076 was selected as the future climate to investigate the climate change effects.

Line 239-245: Under climate change conditions (RCP-8.5, 2058-2076), more concurrent soil water deficit and atmospheric dryness led to decreased stomatal conductance (Fig. 6a). Without considering the technological advances³³, such as seeds and fertilizer improvements, on crop yields, i.e. no technology yield trend, irrigation water use of MAD irrigation scheme under RCP-8.5 scenario (2058-2076) increased by 16.1%, accompanied with significant reductions in crop yield (-24.5%), economic profit (-54.1%), and irrigation water productivity (-29.2%), when compared with the current climate condition (2001-2019) (Fig. S11).

Line 250-253: Compared with MAD, SDD could still significantly reduce irrigation water use (-16.5%, -57.3 mm) and increase irrigation water productivity (+15.8%, 0.6 kg/m³), while it made negligible contributions to economic profits under future climate conditions (Fig. 6b).

Fig. 6 The impacts of climate change on irrigation water use and crop productivity. **a**, The simulated shifts of soil water supply (soil moisture), atmospheric evaporative demand (VPD), and

stomatal conductance (G_s) without irrigation under current climate (2001-2019) and over three continuous time periods (2020-2038, 2039-2057, and 2058-2076) under RCP-8.5 scenario. Each point showed the averages of daily soil moisture, VPD, and G_s during the peak growing season (July and August) in each scenario at 12 sites in Nebraska. Relative soil moisture, the ratio (in percentage) of available water to field capacity, was applied to avoid the impact of heterogeneous soil properties at 12 sites. **b**, The performances of the plant-centric SDD and soil-moisture-based MAD irrigation schemes with universal parameters under the RCP-8.5 scenario (2058-2076) (similar to Fig 5) without considering technological advances on crop yields. **c**, The schematic diagram of the main processes by which climate change affects irrigation water use and crop productivity.

Fig. S11 Performances of MAD irrigation scheme between current climate (2001-2019, baseline) and RCP-8.5 scenario (2058-2076). Relative (a) and absolute (b, c, d, and e) differences in irrigation amount, yield, profit, and irrigation water productivity with universal parameters between current climate (2001-2019) and RCP-8.5 scenario (2058-2076) across 12 sites in Nebraska.

L238-240: Just considering corn yields in places with much greater VPD and higher temperatures, one wonders how realistic these simulations are. For example, In the northwest corner of the Texas Panhandle, county wide corn yields approach 300 bushels per acre and are better than those in much of the corn belt. This in in a region with intensive irrigation due to its hot, semiarid to arid climate,

and lack of precipitation. In this region, irrigation quite adequately resolves yield loss under conditions that are already more severe.

Reply: Thank you for the comments. Let us introduce more about the setting of *ecosys* model under climate change. The incremental change scheme without the acclimation of parameters was selected in *ecosys* model with the ensemble predictions of air temperature, precipitation, and [CO₂] from 15 Coupled Model Intercomparison Project phase 5 (CMIP5) models under Representative Concentration Pathway 8.5 (RCP-8.5). All the parameters of *ecosys* model under RCP-8.5 (2058-2076) kept the same as those under current climate (2001-2019). Thus, the growing season under RCP-8.5 would be shortened with earlier anthesis date and shorter grain-filling period as higher temperature under RCP-8.5 speeds maturation, resulting in maize yield decline under climate change.

As you mentioned, maize in the Texas Panhandle could maintain high yield under hot, semiarid to arid climate due to intensive irrigation. There are several reasons for discussion: (1) the seeds used in Texas may be adapted to the hot, semiarid to arid climate, while the *ecosys* simulations under climate change used the same phenology parameters without acclimation from current climate, i.e. no genetic advances considered; (2) the existed cooling effects due to intensive irrigation in the field experiments was not considered in the *ecosys* model simulations under climate change; (3) the irrigation scheme with MAD under climate change was the same as that under current climate, i.e. no irrigation practice improvements considered.

Although we could apply the acclimation of parameters in *ecosys* under climate change to maintain the high maize yields under climate change, we think that the setting of the *ecosys* model without the acclimation of parameters under climate change is more reasonable. The reason is that we aimed to investigate the impacts of environmental variables under climate change, rather than other technological advances.

L250-251: The relative difference is undefined. Is it (result for SDD) minus (result for MAD), or is it (result for MAD) minus (result for SDD)?

Reply: The relative difference is (results for SDD) minus (results for MAD) divided by (results for MAD). We defined the relative and absolute differences in Methods (Eq. 11 and Eq. 12). We have added this information in the text to refer to it. Please see the revised texts in the manuscript below:

Line 503-504:

$$\text{Relative difference} = \frac{SDD - MAD}{MAD} \times 100\% \quad (11)$$

$$\text{Absolute difference} = SDD - MAD \quad (12)$$

Line 258-265: **Fig. 5 Performances of plant-centric SDD and soil-moisture-based MAD irrigation schemes across 12 sites in Nebraska under current climate (2001-2019).** **a**, The box plots of relative differences (see Methods) in irrigation amount, yield, profit, and irrigation water productivity (ratio between yield and irrigation) between SDD and MAD across 228 site-years. **b**, **c**, **d**, and **e**, The box plots of absolute differences in irrigation amount, yield, profit, and irrigation water productivity between SDD and MAD across 228 site-years. SDD and MAD irrigation schemes both used universal parameters across 12 sites in Nebraska. Boxes showed 25th–75th

percentiles, and the light green line and firebrick diamond denoted the median and mean for each box, respectively.

L274-275: It is only partially true that a co-regulation mechanism has never been applied in irrigation management. See the works of Wanjura, Mahan, Upchurch, Lascano, O'Shaughnessy, Andrade, etc. who go about it in other ways, combining different models and sensing of canopy temperature (as influenced by stomatal conductance) and soil water content (see review of this work by Evett et al. (2020)). The present manuscript should discuss the prior work along these lines and then draw the distinctions between the present work and the past and some ongoing work, rather than fail to recognize the earlier and ongoing work. The difference between the present paper and earlier ones is that in the present paper the focus is directly on stomatal conductance whereas in earlier works the focus has been on the properties related to changes in stomatal conductance. In many cases those properties are more easily sensed in the field than is stomatal conductance. See the following papers for plentiful citations of earlier work:

Evett, S.R., S.A. O'Shaughnessy, M.A. Andrade, P.D. Colaizzi, R.C. Schwartz, H.S. Schomberg, K.C. Stone, E.D. Vories, and R. Sui. 2020. Theory and development of a VRI decision support system: The USDA-ARS ISSCADA approach. *Trans. ASABE*. 63(4): <https://doi.org/10.13031/trans.13922>

Also, see:

O'Shaughnessy, S.A., M. Kim, M.A. Andrade, P.D. Colaizzi, and S.R. Evett. 2020. Site-specific irrigation of grain sorghum using plant and soil water sensing feedback - Texas High Plains. *Agric. Water Manage.* 240 (2020) 106273. <https://doi.org/10.1016/j.agwat.2020.106273>

Reply: Thank you for providing these important references. Current irrigation scheduling research mainly includes conventional soil-based (such as MAD using neutron probe), plant-based (such as ET-based and CWSI/TTT based on canopy temperature), and soil-plant hybrid (such as the ISSCADA-hybrid) irrigation scheduling. We agree that the plant-based and soil-plant hybrid irrigation scheduling considers the impacts of both soil water supply and atmospheric evaporative demand. The plant-based irrigation scheduling determines the irrigation decisions based solely on plant responses, while the plant-soil hybrid irrigation scheduling determines the irrigation decisions based on the plant and soil feedbacks. The difference between the proposed supply-demand dynamics (SDD) irrigation scheduling and the existing plant-based/hybrid scheduling is that SDD determines the dynamic soil moisture thresholds, varying with VPD, based on the co-regulation of soil moisture and VPD on stomatal conductance. Then, the dynamic soil moisture thresholds can be applied for irrigation scheduling directly with soil moisture and VPD measurements, without the need of considering the plant feedback. We have added this discussion and references in the Discussion sections. Please see the revised texts in the manuscript below:

Line 82-85: The innovation of this study is to apply the co-regulation pattern into irrigation management, and we find the new method has demonstrated a large improvement over the existing soil-moisture-only irrigation metrics and thus could have significant contributions to water sustainability.

Line 282-289: Our study proposed and implemented the plant-centric SDD irrigation scheme based on the plant water supply-demand dynamics, i.e. the co-regulations of soil moisture and VPD on stomatal conductance. This co-regulation mechanism has been widely demonstrated by observational evidence^{11,12,18,23,24}, but has seldom been applied in irrigation management directly³⁵. Although some newly developed irrigation practices were plant-based and/or plant-soil hybrid^{35,36},

such as evapotranspiration (ET)-based^{37,38} and canopy temperature-based^{35,39} irrigation, the plant-centric SDD irrigation scheme based on supply-demand dynamics was the first application in leveraging the co-regulation mechanism from plant physiology.

L278: Suggest changing "show" to "indicate", and suggest changing "can" to "may". The results are not definitive.

Reply: We have revised it. Please see the revised texts in the manuscript below:

Line 289-291: Our results indicated that the new SDD scheme may make significant contributions to water sustainability, compared with the already highly efficient MAD scheme, under both current and future climate conditions.

L281: Suggest changing "can" to "could". The purported benefits are not proven in this report. Use of the subjunctive in this and other statements if appropriate.

Reply: Thank you for the suggestions. We have revised this section with the subjunctive tone. Please see the revised texts in the manuscript below:

Line 292-298: Adopting the SDD irrigation scheme could provide economic incentives to producers. Our study has shown that SDD could lead to about \$13.8/ha increase of profit when compared with MAD under current climate conditions. Moreover, producers could further increase their net revenue by the emerging water right trading. For example, producers could sell or lease their saved water rights from the agriculture sector (with lower water value) to the industry sector (with higher water value) and to the federal government, state agencies, and/or non-governmental organizations for environmental conservation purposes⁴⁰⁻⁴².

L304-309: This description of the application of the SDD irrigation scheme is all in the realm of possibility but these have not been demonstrated so are conjecture, so please use the subjunctive tense – “Could” rather than “can”, “could enable” rather than “enables”.

Reply: We have revised this section with the subjunctive tone. Please see the revised texts in the manuscript below:

Line 316-323: This approach is a promising direction to enable the implementation of the SDD irrigation scheme at every field based on models constrained by satellite-derived, field-level ecohydrological variables⁵⁰. Additionally, real-time weather forecasts could be applied to provide forecasted ecohydrological variables for the SDD irrigation scheme, allowing producers to initiate irrigation days in advance in order to increase soil water supply and/or to avoid plant water stress due to the anticipated decrease in soil moisture and/or increase in VPD. We thus hope the SDD irrigation scheme could be potentially applied at the field scale and contribute to water sustainability in the future.

L309-310: At least this claim of practicality is presented as a belief, not a fact. However, applying a complex SPAC model such as ecosys to field situations is quite difficult. Much easier but still effective prior solutions to irrigation management have failed to be widely adopted yet could be

enabled more widely through the use of satellite remote sensing and associated simulation modeling. The SDD approach has yet to be demonstrated to be an effective tool, much less widely adopted.

Reply: We have revised this sentence as a promising vision. Please see the revised texts in the manuscript below:

Line 321-323: We thus hope the SDD irrigation scheme could be potentially applied at the field scale and contribute to water sustainability in the future.

L322-323: Which version of the Penman-Monteith formulation was used? Please give details of this inversion from ET data to stomatal conductance. Stomatal conductance changes with VPD, solar irradiance, and leaf water potential. How were leaf water potential and solar irradiance factored in? The Penman-Monteith formulation allows for solar irradiance but not for leaf water potential. This is far from an obvious inversion although it has been done before. Lascano and others have demonstrated a two-source (canopy and soil surfaces) implicit ET model and have shown how it can be inverted to compute stomatal conductance (Lascano et al., 2010a,b). Because it computes canopy and air temperatures separately, the Lascano et al. approach is more accurate than is inversion of the Penman-Monteith formulation under conditions where canopy and air temperatures vary substantially, which is the case in more arid, and highly irrigated regions. Because the Penman-Monteith assumes equality of air and canopy temperatures, inverting the Penman-Monteith will give approximately the same results as an implicit solution in a more humid climate (e.g., eastern Nebraska) but will give a different and less accurate stomatal conductance in regions where VPD and canopy-air temperature differences are both greater. In addition, there is some evidence that eddy covariance bias increases in more arid environments, which would add more to the bias of the results of this study.

Lascano, R.J., C.H.M. van Bavel, and S.R. Evett. 2010a. A field test of recursive calculation of crop evapotranspiration. *Trans. ASABE*. 53(4):1117-1126. <https://doi.org/10.13031/2013.32601>

Lascano, R.J., and S.R. Evett. 2010b. Calculation of canopy resistance with a recursive evapotranspiration model. Pp. 20-23 In Symposium 2.1.1 of 19th World Congress of Soil Science, Soil Solutions for a Changing World, 1-6 August 2010, Brisbane, Australia.

Reply: Thank you for introducing **the recursive evapotranspiration method** for the possible application in this research (Lascano and Evett, 2010; Lascano et al., 2010). Let us clarify different approaches to obtain the co-regulation pattern at three AmeriFlux sites (US-Ne1, Ne2, and Ne3 in eastern Nebraska) and 12 sites across Nebraska.

For three AmeriFlux sites (US-Ne1, Ne2, and Ne3), the canopy-level stomatal conductance was obtained by the inversion of the Penman-Monteith equation. As you mentioned, the inversion of the Penman-Monteith equation gave pretty good estimations in more humid regions (e.g. eastern Nebraska). Thus, we believed that the inverted canopy-level stomatal conductance at three AmeriFlux sites (US-Ne1, Ne2, and Ne3) using the Penman-Monteith equation was reliable.

For 12 sites across Nebraska, as there were no eddy covariance measurements for the sites located at central and western Nebraska, the canopy-level stomatal conductance at 12 sites from eastern to western Nebraska was the model simulation from the *ecosys* model, rather than inverting the Penman-Monteith equation. As we have described before, *ecosys* is a mechanistic ecosystem model with fully connected balances and interactions for water, energy, carbon and nutrient cycles in the soil-vegetation-atmosphere continuum.

We have added more details of the inversion of the Penman-Monteith equation in the Methods section. Solar radiation was considered, while leaf water potential was not considered in the inverted Penman-Monteith equation. This approach has been widely used to obtain canopy-level stomatal conductance (Kimm et al., 2020; Novick et al., 2016; Xiao et al., 2020). Please see the revised texts in the manuscript below:

Line 343-363:

The second set was eddy covariance measurements of maize cropping systems (seed: Pioneer 33P67/33B51) from 2001 to 2012 at three AmeriFlux sites (US-Ne1, Ne2, and Ne3). US-Ne1 and Ne2 were irrigated sites, with a continuous maize cropping system during 2001-2012 for US-Ne1 and with a maize-soybean rotation cropping system during 2001-2009 and then a continuous maize cropping system during 2010-2012 for US-Ne2. US-Ne3 was rainfed with a maize-soybean rotation cropping system during 2001-2012. The soil at three AmeriFlux sites was a deep silty clay loam consisting of four soil series: Yutan, Tomek, Filbert, and Filmore. The soil moisture data used here was from the top soil layer (10-25 cm). The canopy-level stomatal conductance (G_s) was derived by inverting the Penman-Monteith equation⁵² (Eqs. 1 and 2) from the eddy covariance measurements at the hourly scale^{18,23,53}, and the averaged value near midday (from 12:00-14:00) was applied as the daily canopy-level stomatal conductance to remove the diurnal cycle. This inversion was only conducted during peak growing season (July and August) to avoid the impact of LAI²³. The impact of evaporation from canopy interception and of low incoming shortwave radiation was removed by data filtering²³, i.e. excluding the data within two days following every precipitation and irrigation event, and periods of low incoming shortwave radiation conditions ($<500 \text{ Wm}^{-2}$).

$$\lambda E = \frac{\Delta(R_n - G) + \rho c_p g_a VPD}{\Delta + \gamma(1 + g_a/G_s)} \quad (1)$$

$$G_s = g_a \gamma / \left\{ \frac{\Delta(R_n - G) + \rho c_p g_a VPD}{\lambda E} - (\Delta + \gamma) \right\} \quad (2)$$

where Δ is the slope of the water vapor deficit; R_n and G are net radiation and soil heat flux, respectively; ρ is the air density; c_p is the specific heat capacity for dry air; g_a is aerodynamic conductance; γ is the psychrometric constant; and λE is evapotranspiration.

L335: Suggest changing "validated" to "tested". No model is ever "valid" but some are thoroughly tested with statistical reports indicating how well the simulated results compared with some objective measures.

Reply: We have revised it following your suggestion. Please see the revised texts in the manuscript below:

Line 369-371: *Ecosys* simulates coupled energy, water, carbon, and nutrient cycles⁵⁴⁻⁵⁷, and has been extensively tested in various agricultural ecosystems^{54-56,58,59}.

L337: Grant et al. (2007) indeed applied *ecosys* to an irrigated situation but did not describe how irrigation was handled by the model. Although *ecosys* has been used for a wide range of ecosystems, ranging from peat bogs, to forests, to agriculture, it appears that it has not been tested for simulation of ET and yield of maize and soybean beyond the one study. In the Grant et al. (2007) study, about 17% of variations in eddy covariance LE fluxes for irrigated maize were not explained by the model,

while irrigated grain yield was observed was close to that simulated. With only one year of maize data, the Grant et al. (2007) report is not a “rigorous” test. *ecosys* is currently being used in an AgMIP model intercomparison study using multiple years of LE data from eddy covariance and weighing lysimeter that may shed more light on its ability to simulate ET.

Reply: Thank you for the comments.

(1) *ecosys* model could incorporate the actual hourly irrigation records, and has the option of the auto-irrigation scheme based on soil moisture. For the *ecosys* simulations at three AmeriFlux sites (US-Ne1, Ne2, and Ne3), the actual hourly irrigation records were inputs of *ecosys* as the irrigation records (line 413-422); while for the *ecosys* simulations at 12 sites across Nebraska, we applied the auto-irrigation scheme for the irrigation impacts (line 445-447), as we do not have the actual irrigation records. We also have added more information about how irrigation is handled in the *ecosys* model for this research (line 483-491). Please see the revised texts in the manuscript below:

Line 415-422: Three AmeriFlux sites (US-Ne1, Ne2, and Ne3) had the complete data from 2001 to 2012 to test model performance, including the meteorological variables (such as surface air temperature, downward shortwave radiation, wind speed, and precipitation), eddy-covariance fluxes (such as GPP, net ecosystem exchange-NEE, and LE) from the FLUXNET2015 dataset⁶⁶ (<http://fluxnet.fluxdata.org/data/fluxnet2015-dataset/>) and detailed ground-based crop growth observations (i.e. planting/harvest date, irrigation/fertilization records, LAI, and yield) from the Carbon Sequestration Program (CSP) at University of Nebraska-Lincoln’s Agricultural Research and Development Center (<http://csp.unl.edu/Public/sites.htm>)⁶⁷.

Line 447-449: The auto-irrigation scheme in *ecosys* with the widely used soil-based MAD-50%⁷³ was applied to determine the irrigation scheduling at 12 sites to test the model performance.

Line 484-492: For the irrigation setting in the *ecosys* model, irrigation was triggered when soil moisture was lower than soil moisture threshold of SDD (varying with VPD) and MAD (constant) at the daily scale during the growing season. Soil moisture was the weighted average of soil water content in different soil layers within the root-zone, which was varying with dynamic root-growth but no more than the top 9 soil layers with a depth of 0.92 m to reduce the impacts of deep wet soil. Irrigation amount was determined by water required to fill current soil water content to field capacity, ignoring the constraints from irrigation infrastructures. Each irrigation event (last for 24 hours) was incorporated into the *ecosys* model at a daily time step in real time.

(2) We have updated the validated performance of *ecosys* model with maize-cropping systems at three AmeriFlux sites (US-Ne1, Ne2, and Ne3) during the period from 2001-2012 and 12 sites across Nebraska during the period from 2010-2019 (Table S5, Fig. S12-15). The R^2 of crop yields at three AmeriFlux sites (US-Ne1, Ne2, and Ne3) were 0.50 during the period 2001-2012. The regression line of maize yields is close to a 1-to-1 relationship. In addition, we have deleted the word “rigorously” in the whole manuscript.

L378: I recommend omitted the word “rigorously”. A test against eddy covariance ET data is not rigorous because eddy covariance ET can underestimate crop ET by up to 30% with common percentages of underestimation being around 15%. Are there soil water balance-based ET data against which to compare the eddy covariance ET data? I do not think so.

Reply: Thank you for the comments and suggestions. We have deleted the word “rigorously” in the revised manuscript. We only used the eddy covariance ET data at three AmeriFlux sites (US-Ne1, Ne2, and Ne3) in this study. The eddy covariance ET measurements at US-Ne1, Ne2, and Ne3 were obtained from FLUXNET2015 Dataset (<https://fluxnet.org/data/fluxnet2015-dataset/>), which has been adjusted for energy balance closure (Pastorello et al., 2020). In addition, the eddy covariance flux measurements at US-Ne1, Ne2, and Ne3 implied a fairly good closure of the energy balance (Suyker and Verma, 2009), which is further confirmed by our own calculation of the energy closure. Thus, we believe the ET eddy covariance dataset at US-Ne1, Ne2, and Ne3 are reliable.

L385-386: Figures S12 through S14 are nice figures but it would be more useful to the reader to place the statistical results in a table where they could be reviewed more easily. The use of the correlation coefficient (r) rather than the coefficient of determination (r^2) tends to inflate the perception of goodness of fit. The r^2 for yield is only 0.37, not convincing (the adjusted r^2 is closer to 0.35). Also, the RMSE and bias values are not very useful without mean values for comparison. More importantly, the regression slope and intercept values should be shown in the table along with the other statistics. In the case of yield, a regression on the data in the figure shows that *ecosys* tends to overestimate yield at the low end, which is consistent with figure S15b.

Reply: Thank you for the comments and suggestions. We have updated the validated performance of *ecosys* model with maize-cropping systems at three AmeriFlux sites (US-Ne1, Ne2, and Ne3) during the period from 2001-2012 and 12 sites across Nebraska during the period from 2010-2019 (Table S5, Fig. S12-15). The R^2 of crop yields at three AmeriFlux sites (US-Ne1, Ne2, and Ne3) were 0.50 during the period 2001-2012. The regression line of maize yields is close to a 1-to-1 relationship.

We have placed the statistical indexes in the Table S5. Two more statistical indexes (NRMSE and NBIase) were added to remove the impacts of mean values. Table S5 showed the statistical results of daily and monthly GPP, ET, LAI, and yield with flux towers/CPS observations with the maize cropping systems at three AmeriFlux sites (US-Ne1, US-Ne2, US-Ne3) during the growing seasons (June to September) of 2001-2012.

Besides, the correlation coefficient (r) has been replaced by the coefficient of determination (r^2) in the Fig. S12-S15 (shown before). We have deleted the word “rigorously” in the whole manuscript.

L387-390: If I understand correctly, meteorological data used to run *ecosys* were obtained from NLDAS-2 but ET data were obtained from site-specific eddy covariance systems. Is that correct? If so, then there is a problem mixing meteorological data that isn't from on-site sensors with ET data that are from on-site eddy covariance systems. This can be particularly important in the case of precipitation.

Reply: Thank you for the comments and suggestions. For three AmeriFlux sites (US-Ne1, Ne2, and Ne3), the meteorological data for *ecosys* and eddy-covariance ET data were obtained from eddy covariance flux tower for consistency. For the 12 sites across Nebraska, the meteorological data for model forcing was obtained from NLDAS2, and the canopy-level stomatal conductance was the model simulations, rather than the inversions of Penman-Monteith equation, as there was no

available eddy covariance fluxes in western and central Nebraska. We have added more details in the revised manuscript to make it clearer. Please see the revised texts in the manuscript below:

Line 415-422: Three AmeriFlux sites (US-Ne1, Ne2, and Ne3) had the complete data from 2001 to 2012 to test model performance, including the meteorological variables (such as surface air temperature, downward shortwave radiation, wind speed, and precipitation), eddy-covariance fluxes (such as GPP, net ecosystem exchange-NEE, and LE) from the FLUXNET2015 dataset⁶⁶ (<http://fluxnet.fluxdata.org/data/fluxnet2015-dataset/>) and detailed ground-based crop growth observations (i.e. planting/harvest date, irrigation/fertilization records, LAI, and yield) from the Carbon Sequestration Program (CSP) at University of Nebraska-Lincoln's Agricultural Research and Development Center (<http://csp.unl.edu/Public/sites.htm>)⁶⁷.

Line 466-472: The field measurements of soil moisture, VPD, and stomatal conductance at the daily scale from the greenhouse experiments and three AmeriFlux sites (US-Ne1, Ne2, and Ne3) were applied to investigate the co-regulation of soil moisture and VPD on stomatal conductance (Eq. 8) as observational evidences. With the validated *ecosys* model at three AmeriFlux sites (US-Ne1, Ne2, and Ne3) and 12 sites across Nebraska (Table S5, Fig. S12-15), the model simulations of daily soil moisture, VPD, and canopy-level stomatal conductance during the peak growing season (July and August) were applied to investigate the co-regulation patterns across 12 sites in Nebraska.

Use past tense when describing what was done. In this case, write "were" instead of "are". This applies to lines 387 through 409 at the very least and really to the entire methods section.

Reply: We have revised the text for what was done with past tense.

L406 and Figure S15b: I can't agree that the data match "well." In particular, the *ecosys* data are strongly skewed towards smaller yields than the NASS data with reduced probability of yields greater in the range from ~180 bu/acre to ~230 bu/acre. Could this bias the results towards stronger negative effects of climate change?

Reply: Thank you for the comments and suggestions.

(1) We updated the pdf results with the same parameters from the original period 2015-2019 (recent 5 years) to the period 2010-2019 (recent 10 years) (Fig. S15h). The NASS irrigated maize yields data was the county-level data, while the simulated maize yields data from *ecosys* was obtained with site-specific meteorological data from NLDAS2 and soil data from gSSURGO. Based on the updated results (Fig. S15h), the probability density functions of NASS matched well with that of *ecosys* with the similar shape and peak location.

(2) The negative effects on maize yields under climate change has been discussed before, rather than the bias of *ecosys* model.

L416-417: This filtering has the potential to bias the results. Particularly in more arid, irrigated environments, late afternoon and early evening ET can be a substantial portion of daily ET. See Tolk et al. (2006).

Tolk, J.A., T.A. Howell, and S.R. Evett. 2006. Nighttime evapotranspiration from alfalfa and cotton in a semiarid climate. *Agron. J.* 98:730–736. doi:10.2134/ agronj2005.0276

Reply: The data filtering, i.e. excluding the data within two days following every precipitation and irrigation event, and periods of low incoming shortwave radiation conditions ($<500 \text{ Wm}^{-2}$), was only applied for the eddy covariance measurements at three AmeriFlux sites (US-Ne1, Ne2, and Ne3) in eastern Nebraska (humid region). The data filtering could minimize bias due to evaporation from soil and leaf surface, plant surface dew formation/evaporation, and soil evaporation from ET, which has been applied in one of our co-author's work published in *Agricultural and Forest Meteorology* (Kimm et al., 2020). We have revised the description. Please see the revised texts in the manuscript below:

Line 350-358: The canopy-level stomatal conductance (G_s) was derived by inverting the Penman-Monteith equation⁵² (Eqs. 1 and 2) from the eddy covariance measurements at the hourly scale^{18,23,53}, and the averaged value near midday (from 12:00-14:00) was applied as the daily canopy-level stomatal conductance to remove the diurnal cycle. This inversion was only conducted during peak growing season (July and August) to avoid the impact of LAI²³. The impact of evaporation from canopy interception and of low incoming shortwave radiation was removed by data filtering²³, i.e. excluding the data within two days following every precipitation and irrigation event, and periods of low incoming shortwave radiation conditions ($<500 \text{ Wm}^{-2}$).

Line 463-464: The impacts of $[\text{CO}_2]$, nutrient, radiation, and temperature on stomatal conductance were not considered in this research.

L418: Is "screen out" what is meant here? It seems that "identify" is what is meant. If not, then this confused me.

Reply: We have revised "screen out" to "identify". Please see the revised texts in the manuscript below:

Line 473-474: In addition, Lindeman, Merenda, and Gold method (LMG)^{29,30} was used to identify the relative importance of soil moisture and VPD on stomatal conductance.

L425-and later: Use past tense when describing what was done. The use of present tense here causes confusion and misunderstanding.

Reply: We have revised the text for what was done with past tense.

L431-433: This is a simplistic and unrealistic way to use the MAD concept in irrigation management. Early season irrigation management and late season irrigation management typically do not strictly follow the MAD concept for good reasons. Reasons include holding off on early irrigation to force root growth into deeper soil layers and reducing irrigation during vegetative growth (by adjusting the MAD level). Also, the depth of soil considered in computing soil moisture varies with root growth in sophisticated irrigation management and may be greater than 0.92 m.

Reply: Thank you for the comments and suggestions.

(1) We fully agree that the traditional MAD irrigation scheme could be optimized through many settings, such as the growing stage-specific MAD. As a matter of fact, the MAD threshold level presented in the current study was already optimized by maximizing the economic profits as the benchmarks. We admit that the constant MAD level is a simple approach for testing.

Furthermore, since the optimal MAD threshold is not the focus of this study, the major goal of this study is to incorporate the acknowledged co-regulation pattern in plant physiology into irrigation management, and to assess its efficacy for irrigation guidance. The proposed supply-demand dynamics (SDD)'s threshold level varied with VPD, and we compared its performance with the already optimized MAD level to investigate the benefits of incorporating the co-regulation patterns into irrigation.

(2) For the soil depth, there are 12 soil layers (the depth of the bottom for each layer: 0.01m, 0.05m, 0.1m, 0.15m, 0.18m, 0.28m, 0.35m, 0.59m, 0.92m, 1.32m, 1.6m, 2.00m) with a maximum depth of 2.0 m in *ecosys*. Soil moisture for irrigation is the weighted average of soil water content in different soil layers within the root-zone, which is varying with dynamic root-growth. Crops could absorb water from deeper roots in the soil, but it takes a long time for water transport from deeper roots to reach the stem and then leaves. Thus, we add the constraint of the maximum depth of 0.92 m to reduce the impacts of deep wet soil. We have added more details of the soil moisture used for irrigation. Please see the revised texts in the manuscript below:

Line 440-442: There were 12 soil layers (the depth of the bottom for each layer: 0.01m, 0.05m, 0.1m, 0.15m, 0.18m, 0.28m, 0.35m, 0.59m, 0.92m, 1.32m, 1.6m, 2.00m) with a maximum depth of 2.0 m.

Line 486-489: Soil moisture was the weighted average of soil water content in different soil layers within the root-zone, which was varying with dynamic root-growth but no more than the top 9 soil layers with a depth of 0.92 m to reduce the impacts of deep wet soil.

Reviewer #3 (Remarks to the Author):

The study by Zhang et al. presented an important update to agricultural irrigation schemes. They first used observations from a controlled experiment and flux towers to demonstrate the relative importance of VPD and soil moisture. Then, they used a processed based model to test how much water can be saved without sacrificing productivity. Their findings could be of great interest from the landholders and local councils. I find the paper to be well-written with clear logic. Below are some minor points I suggest the authors to consider.

Reply: Thank you for your positive review and constructive comments. We have carefully revised this manuscript following your comments and suggestions.

The proposed plant-driven irrigation scheme is not evaluated with real data. I would imagine there will be farms in the same areas with different irrigation intensity. The authors could demonstrate the advantage by using the production data from farming using different irrigation intensity. Although the change of intensity is not necessarily the same as the plant driven scheme. Using real data is usual more convincing than just model predictions. Having said that, I know little about USA farm management so the idea may not be realistic.

Reply: Thank you for the comments and suggestions. We are now doing a 3-year field experiment to test SDD irrigation scheme at two sites in Nebraska starting from this year (one in eastern at Mead and one in central-western Nebraska). Though we are not reporting real-world testing results of SDD in the current manuscript, we believe our study on combining soil water supply and atmospheric evaporative demand for sustainable irrigation based on the validated hydraulically-driven ecosystem model, *ecosys*, is rigorous and has its unique and broader value to the community.

In addition, we have updated the validated performance of *ecosys* model with maize-cropping systems at three AmeriFlux sites (US-Ne1, Ne2, and Ne3) during the period from 2001-2012 and 12 sites across Nebraska during the period from 2010-2019 (Table S5, Fig. S12-15). The R^2 of crop yields at three AmeriFlux sites (US-Ne1, Ne2, and Ne3) were 0.50 during the period 2001-2012. The regression line of maize yields is close to a 1-to-1 relationship.

The difficulty of the application of the SDD irrigation scheme is to determine the dynamic threshold level, which varies with VPD based on the co-regulation of soil moisture and VPD on stomatal conductance. Once the dynamic threshold level of SDD was determined, SDD can be tested for irrigation with the measurements of soil moisture and VPD. As explained in the Discussion section, data-model fusion is a promising direction to achieve this goal.

Please see the revised texts in the manuscript below:

Table S5. The statistics indexes of *ecosys* simulated daily and monthly GPP, ET, LAI, and yield with flux towers/CPS observations with the maize cropping systems at three AmeriFlux sites (US-Ne1, US-Ne2, US-Ne3) during the growing seasons (June to September) of 2001-2012.

Variables	GPP					ET					LAI					Yield				
Indexes	RMSE (gC/m ²)	NRMSE (%)	Bias (gC/m ²)	NBias (%)	R ²	RMSE (mm)	NRMSE (%)	Bias (mm)	NBias (%)	R ²	RMSE (m ² /m ²)	NRMS E (%)	Bias (m ² /m ²)	NBias (%)	R ²	RMSE (t/ha)	NRMS E (%)	Bias (t/ha)	NBias (%)	R ²
Daily	3.82	29.6	0.26	2.0	0.79	1.04	27.7	-0.47	-12.5	0.73	1.10	32.6	-0.56	-16.6	0.74	1.31	13.2	0.12	1.2	0.50
Monthly	92.30	24.1	7.50	2.0	0.82	21.61	19.2	-14.14	-12.6	0.80	1.02	32.0	-0.58	-18.2	0.77					

* o_i and s_i are the observations and model simulations, respectively; RMSE is the Root Mean Square Error ($RMSE = \sqrt{\frac{1}{N} \sum_{i=1}^N (s_i - o_i)^2}$); NRMSE

is the Normalized Root Mean Square Error ($NRMSE = \frac{\sqrt{\frac{1}{N} \sum_{i=1}^N (s_i - o_i)^2}}{\bar{o}_i}$); Bias is the mean bias ($Bias = \frac{1}{N} \sum_{i=1}^N (s_i - o_i)$); NBias is the

normalized mean bias ($NBias = \frac{\frac{1}{N} \sum_{i=1}^N (s_i - o_i)}{\bar{o}_i}$); R^2 is the coefficient of determination ($R^2 = 1 - \frac{\sum_{i=1}^N (s_i - o_i)^2}{\sum_{i=1}^N (s_i - \bar{o}_i)^2}$).

Fig. S12 Performance of *ecosys* model on daily GPP, NEE, LE, and LAI with the statistical results (RMSE, Bias, and R^2) for continuous maize cropping systems during the period 2001-2012 at US-Ne1.

Fig. S13 Performance of *ecosys* model on daily GPP, NEE, LE, and LAI with the statistical results (RMSE, Bias, and R²) for maize (yellow regions) and soybean (green regions) cropping systems during the period 2001-2012 at US-Ne2.

Fig. S14 Performance of *ecosys* model on daily GPP, NEE, LE, and LAI with the statistical results (RMSE, Bias, and R²) for maize (yellow regions) and soybean (green regions) rotation cropping systems during the period 2001-2012 at US-Ne3.

Fig. S15 The performance of *ecosys* model with maize cropping systems at three AmeriFlux sites (US-Ne1, Ne2, and Ne3) during the period from 2001-2012 and 12 sites with a dramatic rainfall gradient across Nebraska during the period from 2010-2019. The colorbar shows the normalized gaussian kernel density estimation of the scatters. Black dashed lines indicate the 1-to-1 relationship. The red line is the regression line with the slope and intercept. The probability density function of the maize yields at the 12 sites is the gaussian kernel density estimation.

The empirical model (Equation 6) is quite different from what often used in other studies. For example, Sulman et al. (2016) used a linear model:

$$G_s = C_1 + C_2 \ln(VPD) + C_3 \text{SWC} + C_4 \ln(VPD) \text{SWC}$$

where C1-C4 are fitted parameters. This fitting could obtain the impacts (i.e., slopes) of VPD and SWC on G_s. However, the model used here is a multiple of VPD and SWC and thus difficult to separate the sensitivity of G_s to each factor. This study then use the R² rather than regression coefficient. I feel the authors could consider using a formula that separate the impact of each driver better. The linear model by Sulman could work or a more sophisticated model (e.g., Whitley et al., 2013) might be even better. I'm not saying the authors need to use those models but it makes sense to justify the model selection in the method or introduction.

References:

Sulman, B. N., Roman, D. T., Yi, K., Wang, L., Phillips, R. P., & Novick, K. A. (2016). High atmospheric demand for water can limit forest carbon uptake and transpiration as severely as dry soil. *Geophysical Research Letters*, 43(18), 9686–9695. <https://doi.org/10.1002/2016GL069416>
 Whitley, R., Taylor, D., Macinnis-Ng, C., Zeppel, M., Yunusa, I., O'Grady, A., Froend, R., Medlyn, B., & Eamus, D. (2013). Developing an empirical model of canopy water flux describing the common response of transpiration to solar radiation and VPD across five contrasting woodlands and forests. *Hydrological Processes*, 27(8), 1133–1146. <https://doi.org/10.1002/hyp.9280>

Reply: Thank you for providing these references about the stomatal conductance models.

(1) There are many approaches for modeling stomatal conductance, including empirical, such as Jarvis (Jarvis, 1976), semi-empirical, such as Ball-Berry (Ball et al., 1987), and mechanistic approaches, such as optimality theory (Medlyn et al., 2011). The empirical models for stomatal conductance can be complex (Jarvis, 1976; Whitley et al., 2013) or simple (Kimm et al., 2020; Sulman et al., 2016). As we focus on the co-regulation of soil moisture and VPD on stomatal conductance, the direct impacts of other environmental factors and photosynthetic rate are not considered in this research. There are mainly two types of empirical models for investigating the impacts of soil moisture and VPD on plants, including linear (Sulman et al., 2016) and nonlinear models (Kimm et al., 2020). Sulman et al. (2016) used a linear model with soil water potential and VPD for transpiration and GPP (Eq. 1). Soil water potential was calculated using the exponential relationship fitted in Wayson et al. (2006) (Eq. 2).

$$F = C_1 + C_2 \ln(VPD) + C_3 \psi_s + C_4 \ln(VPD) \psi_s \quad (1)$$

$$\psi_s = -1 \times \left[2.61 \times 10^8 \times 189^{(-2.51 \times \text{SWC}^{0.1386})} \right] \quad (2)$$

One of our co-author's work (Kimm et al., 2020) indicated that the linear (Sulman et al., 2016) and nonlinear (Kimm et al., 2020) empirical models have similar fitted performance (see Table S1 and Fig. 4 from Kimm et al. (2020), shown below). In addition, the exponent of VPD was close to 1 for croplands (Lin et al., 2018), and the linear function between stomatal conductance and soil water potential led to the logistic function between stomatal conductance and soil moisture. Thus, the nonlinear empirical model proposed by Kimm et al. (2020) (Eq. 8 in the revised manuscript) was applied in this research.

We have added more details on the empirical model selection for stomatal conductance in the Methods. Please see the revised texts in the manuscript below:

Line 456-465: Empirical nonlinear statistical model. There were linear and nonlinear empirical models available for modeling stomatal conductance^{20,23,73}. As the exponent of VPD was close to 1 for croplands²⁴ and the linear function between stomatal conductance and soil water potential led to the logistic function between stomatal conductance and soil moisture, we used an empirical nonlinear statistical model to describe the co-regulation of soil moisture and VPD on stomatal conductance (Eq. 8), including two sub-functions²³. The first one denoted an inverse proportional relationship between VPD and stomatal conductance^{24,74,75}, and the other represented the logistic function between soil moisture and stomatal conductance^{76,77}. The impacts of [CO₂], nutrient, radiation, and temperature on stomatal conductance were not considered in this research.

$$G_s = f(VPD, \theta) = \left(\frac{a_1}{VPD - a_2} + a_3 \right) \times \left(\frac{b_1}{1 + \exp(b_2(\theta - b_3))} \right) \quad (8)$$

Table S1. Model fitting and fractions of attribution (Kimm et al., 2020)

Site	Crop	Multiple linear regression				Non-linear model		
		R ²	log(VPD)	SWC	$\frac{SWC}{\log(VPD)}$	R ²	VPD	SWC
US-Bo1	Maize	0.44	0.85	-	0.15	0.41	0.84	0.16
	Soybean	0.59	0.79	0.21	-	0.64	0.75	0.26
US-Br1	Maize	0.15	0.85	0.01	0.14	0.13	1.00	0.00
	Soybean	0.45	0.94	0.06	-	0.45	0.82	0.18
US-Br3	Maize	0.20	0.63	0.37	-	0.18	0.68	0.32
	Soybean	0.32	0.91	0.09	-	0.30	1.00	0.00
US-Ne1	Maize	0.50	0.87	0.13	-	0.51	0.77	0.23
US-Ne2	Maize	0.61	0.95	0.04	-	0.60	0.83	0.17
	Soybean	0.53	0.97	0.03	-	0.52	1.00	0.00
US-Ne3	Maize	0.35	1.00	-	0.00	0.35	1.00	0.00
	Soybean	0.67	0.99	0.01	0.04	0.66	0.97	0.03
US-Ro1	Maize	0.32	1.00	0.00	-	0.35	1.00	0.00

Fig. 4. A surface of the fitted G_s models at Bo1-maize on three-dimensional space of VPD, SWC, and G_s with black dots representing observational data points. Color-coded lines on VPD-SWC plane are contours of G_s , and gray lines and dots are projection lines from the surface and projected points to the plane of VPD-SWC. Color scale and Z-axis redundantly represent G_s for visualization purpose. (Kimm et al., 2020)

(2) Lindeman, Merenda, and Gold method (LMG) was widely used to determine the relative importance of individual factor to the objective variable, i.e. the individual contributions of soil moisture and VPD on stomatal conductance. Please see the revised texts in the manuscript below:

Line 473-477: In addition, Lindeman, Merenda, and Gold method (LMG)^{29,30} was used to identify the relative importance of soil moisture and VPD on stomatal conductance. LMG decomposed the determination coefficients of a linear regression (R^2) to the contributions of soil moisture and VPD, i.e. to quantify the variation of stomatal conductance that can be explained by soil moisture and VPD, while taking the correlation between soil moisture and VPD into account.

My last point is with the ‘profit’ (Figures 5 and 6). The difference between estimated benefit of the new and old schemes has large variance (i.e., 25% quantile crossing zero). I wonder whether the variance is driven by the variation in sites and climate. If so, it may be better for the authors to present when/where the new scheme is better than the old one.

Reply: Thank you for the comments and suggestions. The profit is calculated based on revenue (maize yields and price) and costs (including irrigation costs and other fixed costs). Thus, the profit difference between the proposed SDD and traditional MAD irrigation scheme is mainly determined by the differences of maize yields and irrigation water use, and also the price of maize and irrigation. Based on the historical price in Nebraska 2019, we have investigated the impacts of climate (i.e. aridity index) and soil (i.e. sand fraction) properties to the differences of irrigation amount, crop yields, profits, and irrigation water productivity between SDD and MAD (Fig. S10). We also added more discussion on this point. Please see the revised texts in the manuscript below:

Line 225-238: The benefits of the SDD irrigation scheme over MAD varied with climate conditions (e.g. aridity index) and soil properties (e.g. sand fraction) (Fig. S10). The absolute differences in irrigation water use and irrigation water productivity between the SDD and MAD irrigation

schemes significantly decreased with increasing aridity index ($p < 0.05$) and also slightly decreased with increasing sand fraction; while the difference in crop yields between SDD and MAD maintained stable and negligible with aridity index and sand fraction. It indicated that the SDD irrigation scheme made larger contributions to water sustainability in wetter and/or less sandy regions than drier and/or more sandy regions. One of the reasons was that VPD in wetter regions was relatively lower than those in drier regions (Fig. S5), thus “water saving” occurred more frequently in wetter regions, resulting in more contributions to water sustainability. On the other hand, VPD had a larger impact on stomatal conductance in wetter regions than that in drier regions (Fig. 3b and S4a). Thus, we concluded that the SDD irrigation scheme could contribute more to water sustainability and economic profits at regions with lower VPD and/or larger constraints from VPD on agricultural drought.

Fig. S10 Variation of relative differences in irrigation amount, yield, profit, and irrigation water productivity between the SDD and MAD irrigation schemes with **a**, aridity index and **b**, sand fraction across 12 sites in Nebraska under current climate (2001-2019).

Detailed comments:

L157 (figure 3 caption). The MAP used here is only for the growing season. What about the impact of snow melt at early spring. My understanding is some region heavily depend on snow melt but not sure about these sites and region.

Reply: The soil profile is near field capacity in the early spring due to snow melt, while there is almost no snow melt after April in Nebraska, i.e. the beginning of the growing season (planting). Thus, the snow melt has little contributions to the available water for crops during the growing season.

L393. The authors chose RCP8.5 which is the most extreme pathway that is unlikely to be true. Some justification is needed for the choice.

Reply: We chose RCP 8.5 to investigate the high warming conditions with the highest greenhouse gas (GHG) emissions. We have added some explanation on the selection of RCP 8.5. Please see the revised texts in the manuscript below:

Line 428-433: For climate change, the ensemble predictions of air temperature, precipitation, and carbon dioxide concentration [CO₂] at the same irrigated sites in Nebraska were ensemble average projections from 15 Coupled Model Intercomparison Project phase 5 (CMIP5) models under Representative Concentration Pathway 8.5 (RCP-8.5)^{68,69} (Table S4). RCP-8.5 was selected as future scenario to investigate the high warming conditions with the highest greenhouse gas (GHG) emissions⁷⁰.

References:

- Ball, J. T., I. E. Woodrow, and J. A. Berry (1987), A model predicting stomatal conductance and its contribution to the control of photosynthesis under different environmental conditions, in *Progress in photosynthesis research*, edited, pp. 221-224, Springer.
- Jarvis, P. (1976), The interpretation of the variations in leaf water potential and stomatal conductance found in canopies in the field, *Philosophical Transactions of the Royal Society of London. B, Biological Sciences*, 273(927), 593-610.
- Kimm, H., K. Guan, P. Gentine, J. Wu, C. J. Bernacchi, B. N. Sulman, T. J. Griffis, and C. Lin (2020), Redefining droughts for the U.S. Corn Belt: The dominant role of atmospheric vapor pressure deficit over soil moisture in regulating stomatal behavior of Maize and Soybean, *Agric. For. Meteorol.* , 287, 107930.
- Lascano, R., and S. Evett (2010), Calculation of canopy resistance with a recursive evapotranspiration model, paper presented at Symposium.
- Lascano, R., C. H. M. van Bavel, and S. R. Evett (2010), A field test of recursive calculation of crop evapotranspiration, *Transactions of the ASABE*, 53(4), 1117-1126.
- Lauffenburger, Z. H., J. J. Gurdak, C. Hobza, D. Woodward, and C. Wolf (2018), Irrigated agriculture and future climate change effects on groundwater recharge, northern High Plains aquifer, USA, *Agricultural Water Management*, 204, 69-80.
- Lin, C., P. Gentine, Y. Huang, K. Guan, H. Kimm, and S. Zhou (2018), Diel ecosystem conductance response to vapor pressure deficit is suboptimal and independent of soil moisture, *Agric. For. Meteorol.* , 250-251, 24-34.
- Medlyn, B. E., R. A. Duursma, D. Eamus, D. S. Ellsworth, I. C. Prentice, C. V. Barton, K. Y. Crous, P. De Angelis, M. Freeman, and L. Wingate (2011), Reconciling the optimal and empirical approaches to modelling stomatal conductance, *Global Change Biology*, 17(6), 2134-2144.
- Miner, G. L., and W. L. Bauerle (2017), Seasonal variability of the parameters of the Ball–Berry model of stomatal conductance in maize (*Zea mays* L.) and sunflower (*Helianthus annuus* L.) under well-watered and water-stressed conditions, *Plant, cell & environment*, 40(9), 1874-1886.
- Najafi, E., N. Devineni, R. M. Khanbilvardi, and F. Kogan (2018), Understanding the Changes in Global Crop Yields Through Changes in Climate and Technology, *Earth's Future*, 6(3), 410-427.
- Novick, K. A., D. L. Ficklin, P. C. Stoy, C. A. Williams, G. Bohrer, A. C. Oishi, S. A. Papuga, P. D. Blanken, A. Noormets, and B. N. Sulman (2016), The increasing importance of atmospheric demand for ecosystem water and carbon fluxes, *Nature climate change*, 6(11), 1023-1027.
- Pastorello, G., et al. (2020), The FLUXNET2015 dataset and the ONEFlux processing pipeline for eddy covariance data, *Scientific Data*, 7(1), 225.
- Sulman, B. N., D. T. Roman, K. Yi, L. Wang, R. P. Phillips, and K. A. Novick (2016), High atmospheric demand for water can limit forest carbon uptake and transpiration as severely as dry soil, *Geophys. Res. Lett.* , 43(18), 9686-9695.
- Suyker, A. E., and S. B. Verma (2009), Evapotranspiration of irrigated and rainfed maize–soybean cropping systems, *Agric. For. Meteorol.* , 149(3), 443-452.
- Wayson, C. A., J. C. Randolph, P. J. Hanson, C. S. B. Grimmond, and H. P. Schmid (2006), Comparison of soil respiration methods in a mid-latitude deciduous forest, *Biogeochemistry*, 80(2), 173-189.
- Whitley, R., D. Taylor, C. Macinnis-Ng, M. Zeppel, I. Yunusa, A. O'Grady, R. Froend, B. Medlyn, and D. Eamus (2013), Developing an empirical model of canopy water flux describing the common response of transpiration to solar radiation and VPD across five contrasting woodlands and forests, *Hydrol. Process.*, 27(8), 1133-1146.
- Xiao, M., Z. Yu, D. Kong, X. Gu, I. Mammarella, L. Montagnani, M. A. Arain, L. Merbold, V. Magliulo, and A. Lohila (2020), Stomatal response to decreased relative humidity constrains the acceleration of terrestrial evapotranspiration, *Environ. Res. Lett.* .

Yang, Y., K. Guan, B. Peng, M. Pan, C. Jiang, and T. E. Franz (2020), High-resolution spatially explicit land surface model calibration using field-scale satellite-based daily evapotranspiration product, *J. Hydrol.*, 125730.

Zhang, J., et al. (2021), Challenges and opportunities in precision irrigation decision-support systems for center pivots, *Environ. Res. Lett.* .

REVIEWER COMMENTS

Reviewer #1 (Remarks to the Author):

The authors have addressed my concerns in a thoughtful manner. The manuscript is much improved.

Reviewer #2 (Remarks to the Author):

Lines 63-64: This statement is not true and misconstrues the information given in reference 14 (USDA NASS). The majority of irrigation scheduling practice has used forms of the Penman-Monteith equation and crop coefficients to estimate the crop evapotranspiration (water demand), which of course changes according to atmospheric aridity (i.e. vapor pressure deficit) among other things. This is the preferred method of determination crop water requirements for irrigation across the nation and in many other countries. The fact that P-M based ET values are then used to estimate the status of soil water does not mean that irrigation scheduling is based solely on soil moisture availability. Please correct this statement to present the actual state of practice.

Figure 3. The legend symbols do not clearly show the cross hatching in the figure, and so may be difficult for readers to interpret.

Line 324: I urge the authors to complete this section with a short paragraph acknowledging that an observational test comparing SDD with MAD has not yet been concluded, and describing their ongoing field test that compares SDD with MAD. Wording from their responses to comments are a good start.

Line 350: Please describe how soil moisture was determined in the field. Was a sensor used? What sensor, how many replicates, etc.?

Line 491: Here it states that each irrigation event lasted 24 hours. An irrigation event that lasts 24 hours is unusual. For example, a center pivot irrigation system will apply water to a point in the field over a span of time much less than 24 hours. Typically it only takes 15 minutes to an hour for the irrigation event to occur at any given point. Points closer to the pivot center experience the longer irrigation period and those at the far end of the sprinkler lateral may be irrigated in 15 minutes or less.

Lines 413-424: After looking at table S5 and the supplementary figures, I cannot agree with the words "matched very well." An r^2 value of 0.50 for yield is far from good agreement, no matter the fact that the regression line lay close to the 1:1 line. The r^2 of 0.75 for ET is also not very good in that it leaves 25% of the data variance unexplained, and Figure S15 shows that ET and LAI were both substantially underestimated. Because the crux of the results depends on yield and ET, the model results should be critically examined with regard to interpretation of the outcome.

Table S5: Please add the slope and intercept values for the regressions and the results of statistical tests on equality of slope with unity and equality of intercept with zero.

Figure S12 and similar figures: A major problem with these relationships is that they include all days of the year whereas the corn growing season is the pertinent time scale for comparison between modeled and measured data. Using the entire year tends to inflate the degree of agreement because of the many small values during the off season.

In the rebuttal, the authors write, "(1) We fully agree that the traditional MAD irrigation scheme could be optimized through many settings, such as the growing stage-specific MAD. As a matter of fact, the MAD threshold level presented in the current study was already optimized by maximizing the economic profits as the benchmarks. We admit that the constant MAD level is a simple approach for

testing. Furthermore, since the optimal MAD threshold is not the focus of this study, the major goal of this study is to incorporate the acknowledged co-regulation pattern in plant physiology into irrigation management, and to assess its efficacy for irrigation guidance. The proposed supply demand dynamics (SDD)'s threshold level varied with VPD, and we compared its performance with the already optimized MAD level to investigate the benefits of incorporating the co-regulation patterns into irrigation."

Adding a version of this text to the discussion could further elucidate the intent and limitations of the study with regard to sophisticated rather than simplistic use of MAD based irrigation scheduling. As it stands these ideas are ignored in the manuscript.

Also in the rebuttal, the authors write, "As you mentioned, maize in the Texas Panhandle could maintain high yield under hot, semiarid to arid climate due to intensive irrigation. There are several reasons for discussion: (1) the seeds used in Texas may be adapted to the hot, semiarid to arid climate, while the ecosys simulations under climate change used the same phenology parameters without acclimation from current climate, i.e. no genetic advances considered; (2) the existed cooling effects due to intensive irrigation in the field experiments was not considered in the ecosys model simulations under climate change; (3) the irrigation scheme with MAD under climate change was the same as that under current climate, i.e. no irrigation practice improvements considered. Although we could apply the acclimation of parameters in ecosys under climate change to maintain the high maize yields under climate change, we think that the setting of the ecosys model without the acclimation of parameters under climate change is more reasonable. The reason is that we aimed to investigate the impacts of environmental variables under climate change, rather than other technological advances."

Adding some version of this to the discussion would do a lot to set the results of the study in the broader context of current and future scenarios, and it would acknowledge that high yields are already obtained with irrigation in hot, arid climates, which implies that yield reductions simulated with ecosys are not the only possible outcome.

Steven R. Evett

Reviewer #3 (Remarks to the Author):

My comments were addressed in the revision and I would recommend this manuscript to be accepted for publishing.

One suggestion for the authors is with their choice of empirical models. As the authors are well aware of, the simple empirical function used to determine VPD and SWC impacts cannot separate the two factors. Although it might be accepted to use for now, under climate change, the fitted parameter values could become invalid. For example, future rainfall may remain the same but VPD could increase due to increase of temperature. The crops then may choose to use water differently and thus should use a different parameter value. This is not a key point of the study so no need to change here but maybe something to consider in the future.

Dear all,

We really thank the editor, associate editor, Dr. Steven R. Evett, and the two anonymous reviewers for their constructive comments and helpful suggestions to improve this manuscript. We have revised the manuscript and addressed these comments point by point, and we believe that we have significantly improved the manuscript. We hope that this revised manuscript could fulfill editors and reviewers' high standards for *Nature Communications*.

In the following responses, reviewer's comments appear in black, our response appears in blue, and the revised text in the manuscript appears in orange.

We look forward to your further feedback!

- Jingwen Zhang, Kaiyu Guan, Bin Peng, on behalf of all the authors

Dr Kasey Bolles, Associate Editor

Thank you again for submitting your manuscript "Sustainable irrigation based on co-regulation of soil water supply and atmospheric evaporative demand" to Nature Communications. We have now received reports from 3 reviewers and, on the basis of their comments, we have decided to invite a revision of your work for further consideration in our journal. Your revision should address all the points raised by our reviewers (see their reports below). In particular, please address the lingering concerns of Reviewer #2, not limited to: further discussion of the goal(s) of the study and broader contextualization in current and future scenarios (echoed by Reviewer #3's final comment). Please also provide clarification in the main text regarding minor line comments, such as a description of the ongoing field experiment.

Reply: Thank you for handling the review and helpful suggestions. We have revised the manuscript following the comments and suggestions from the second and third reviewers. Please see the detailed reply and revised manuscript.

REVIEWER COMMENTS

Reviewer #1 (Remarks to the Author):

The authors have addressed my concerns in a thoughtful manner. The manuscript is much improved.

Reply: Thank you for your approval.

Reviewer #2 (Remarks to the Author):

Dear Dr. Steven R. Evett, thank you for providing the review with the insightful comments and helpful suggestions for this manuscript. We have revised this manuscript following your comments and suggestions, and we hope that this version could solve your concerns.

Lines 63-64: This statement is not true and misconstrues the information given in reference 14 (USDA NASS). The majority of irrigation scheduling practice has used forms of the Penman-Monteith equation and crop coefficients to estimate the crop evapotranspiration (water demand), which of course changes according to atmospheric aridity (i.e. vapor pressure deficit) among other things. This is the preferred method of determination crop water requirements for irrigation across the nation and in many other countries. The fact that P-M based ET values are then used to estimate the status of soil water does not mean that irrigation scheduling is based solely on soil moisture availability. Please correct this statement to present the actual state of practice.

Reply: Thank you for correcting our statement. We have revised this statement in the manuscript following your suggestion. Please see the revised texts in the manuscript below:

Line 65-67: Current irrigation practices have primarily focused on soil water supply side, though we acknowledge that to calculate soil water balance to get soil moisture, evapotranspiration (ET) based on different methods is usually used in most methods¹⁴.

Figure 3. The legend symbols do not clearly show the cross hatching in the figure, and so may be difficult for readers to interpret.

Reply: We have enlarged the legend symbols in Figure 3. Please see the revised Figure 3 in the manuscript below:

Fig. 3 Spatial variability of the co-regulation pattern of soil moisture and VPD on stomatal conductance of maize in Nebraska, United States. **a**, Location of 12 sites with a large rainfall gradient (from 800 mm in the east to 400 mm in the west) in Nebraska. **b**, The relative importance metrics of soil moisture and VPD as drivers of variability in stomatal conductance (from *ecosys*) across 12 sites in Nebraska under current climate (2001-2019) and RCP-8.5 scenario (2058-2076). **c**, **d**, Variation of aridity index and sand fraction across 12 sites in Nebraska. Aridity index was calculated as the ratio of potential evapotranspiration³¹ (PET, mm) and precipitation (P, mm) during growing season from 2001 to 2019, i.e. PET/P.

Line 324: I urge the authors to complete this section with a short paragraph acknowledging that an observational test comparing SDD with MAD has not yet been concluded, and describing their ongoing field test that compares SDD with MAD. Wording from their responses to comments are a good start.

Reply: Thank you for the suggestions. We have added some discussion on the on-going field experiments. Please see the revised texts in the manuscript below:

Line 297-301: Multi-year field experiments comparing the SDD and MAD irrigation schemes are now underway at two sites in Nebraska starting from 2021: one site in eastern Nebraska with variable rate irrigation system and another site at North Platte in western-central Nebraska with center pivot irrigation system. Data collected from these experiments will be used to further validate the results reported here.

Line 350: Please describe how soil moisture was determined in the field. Was a sensor used? What sensor, how many replicates, etc.?

Reply: Thank you for the comments. We have added more detailed information about the soil moisture observations at three AmeriFlux sites (US-Ne1, Ne2, and Ne3) in the revised manuscript. Please see the revised texts in the manuscript below:

Line 358-364: There are three replicates with the soil moisture sensors (theta probes: ML2, Dynamax Inc.) installed horizontally with the profile of soil depth (10, 25, 50 and 100 cm) in the US-Ne1 and US-Ne2, and four replicates with soil moisture sensors (theta probes: ML2, Dynamax Inc.) installed horizontally with the profile of soil depth (10, 25, 50 and 100 cm) in the US-Ne3 (http://csp.unl.edu/public/G_moist.htm). The soil moisture data used here was from the top soil layer (10-25 cm).

Line 491: Here it states that each irrigation event lasted 24 hours. An irrigation event that lasts 24 hours is unusual. For example, a center pivot irrigation system will apply water to a point in the field over a span of time much less than 24 hours. Typically it only takes 15 minutes to an hour for the irrigation event to occur at any given point. Points closer to the pivot center experience the longer irrigation period and those at the far end of the sprinkler lateral may be irrigated in 15 minutes or less.

Reply: We totally agree with what you described here about the constraints (irrigation duration) of the irrigation systems, while we ignored the constraints from irrigation infrastructures for simplification in this research as it is complex and location specific. We have added this description for the duration of irrigation events in the revised manuscript. Please see the revised texts in the manuscript below:

Line 511-514: For simplification, we ignored the constraints on irrigation amount and duration from irrigation infrastructures. Irrigation amount was determined by water required to fill current soil water content to field capacity. Each irrigation event lasting 24 hours was incorporated into the *ecosys* model at a daily time step in real time.

Lines 413-424: After looking at table S5 and the supplementary figures, I cannot agree with the words "matched very well." An r^2 value of 0.50 for yield is far from good agreement, no matter the fact that the regression line lay close to the 1:1 line. The r^2 of 0.75 for ET is also not very good in that it leaves 25% of the data variance unexplained, and Figure S15 shows that ET and LAI were both substantially underestimated. Because the crux of the results depends on yield and ET, the model results should be critically examined with regard to interpretation of the outcome.

Reply: Thank you for the comments. We further re-tuned two parameters of *ecosys*, including fraction of leaf protein in bundle sheath chlorophyll (CHL4) and plant maturity group (GROUPX), at three AmeriFlux sites (US-Ne1, Ne2, and Ne3) manually during the period from 2001-2012. The validated performance of *ecosys* has been improved (Table S5, Figure S12-S15). The R^2 for the daily and monthly GPP during the growing season (May to October) are 0.87 and 0.91. The R^2 for the daily and monthly ET during the growing season are 0.80 and 0.88. The R^2 for the daily and monthly LAI during the growing season are 0.82 and 0.85. The R^2 for the crop yield has been improved to 0.56 (which is relatively higher compared with other similar modeling studies), and the regression line is close to 1:1 line. All the p-values from the F-tests with three null hypotheses

(slope=1, intercept=0, slope=1 and intercept=0) for crop yields are larger than 0.05 (significance level), and all the F-values from the F-tests are less than the F critical values (Table S5). These results indicate that all the three null hypotheses for crop yields cannot be rejected, then we accept it by default, denoting that the regression coefficients equal the given parameters significantly (slope equals 1 and intercept equals 0, see more details in Table S5). Thus, we think that current results could be used to show the performance of the *ecosys* model. In addition, we have changed “matched very well” to “matched well” in the manuscript.

Table S5. The statistics indexes of *ecosys* simulated daily and monthly GPP, ET, LAI, and yield with flux towers/CPS observations with the maize cropping systems at three AmeriFlux sites (US-Ne1, US-Ne2, US-Ne3) during the growing seasons (May to October) of 2001-2012.

Variables	Indexes	Daily	Monthly	
GPP	RMSE (gC/m ²)	3.32	77.58	
	NRMSE (%)	34.40	29.30	
	Bias (gC/m ²)	0.17	4.58	
	NBias (%)	1.80	1.70	
	R ²	0.87	0.91	
	Slope	0.97	1.01	
	Intercept	0.51	0.98	
	H0: Slope = 1	F value	36.47* < F _(0.05,4300,1) =254.31	0.26* < F _(0.05,152,1) =254.31
		p-value	<0.05	0.61*
	H0: Intercept = 0	F value	46.18* < F _(0.05,4300,1) =254.31	0.01* < F _(0.05,152,1) =254.31
		p-value	<0.05	0.92*
	H0: Slope = 1 and intercept =0	F value	24.19 > F _(0.05,4300,2) =19.50	0.40* < F _(0.05,152,2) =19.50
		p-value	<0.05	0.67*
	ET	RMSE (gC/m ²)	0.93	19.32
NRMSE (%)		31.30	21.70	
Bias (gC/m ²)		-0.35	-11.21	
NBias (%)		-11.90	-12.60	
R ²		0.80	0.88	
Slope		0.87	0.88	
Intercept		0.03	-0.34	
H0: Slope = 1		F value	413.04 > F _(0.05,4690,1) =254.31	21.88* < F _(0.05,154,1) =254.31
		p-value	<0.05	<0.05
H0: Intercept = 0		F value	2.13* < F _(0.05,4690,1) =254.31	0.02* < F _(0.05,154,1) =254.31
		p-value	0.15*	0.90*
H0: Slope = 1 and intercept =0		F value	639.93 > F _(0.05,4690,2) =19.50	55.57 > F _(0.05,154,2) =19.50
		p-value	<0.05	<0.05
LAI		RMSE (gC/m ²)	1.01	0.91
	NRMSE (%)	35.80	35.90	
	Bias (gC/m ²)	-0.45	-0.46	
	NBias (%)	-15.90	-18.00	
	R ²	0.82	0.85	
	Slope	0.78	0.78	
	Intercept	0.18	0.09	
	H0: Slope = 1	F value	99.02* < F _(0.05,267,1) =254.31	44.62* < F _(0.05,107,1) =253.02
		p-value	<0.05	<0.05
	H0: Intercept = 0	F value	5.14* < F _(0.05,267,1) =254.31	0.75* < F _(0.05,107,1) =253.02
		p-value	0.03	0.39*
	H0: Slope = 1 and intercept =0	F value	94.51 > F _(0.05,267,2) =19.50	47.87 > F _(0.05,107,2) =19.49
		p-value	<0.05	<0.05
	Yield	RMSE (gC/m ²)	1.25	
NRMSE (%)		12.58		

Bias (gC/m ²)		0.17
NBias (%)		1.67
R ²		0.56
Slope		1.01
Intercept		0.03
H0: Slope = 1	F value	0.01* < F _(0.05,22,1) =248.53
	p-value	0.94*
H0: Intercept = 0	F value	0* < F _(0.05,22,1) =248.53
	p-value	0.99*
H0: Slope = 1 and intercept =0	F value	0.20* < F _(0.05,22,2) =19.45
	p-value	0.82*

o_i and s_i are the observations and model simulations, respectively; RMSE is the Root Mean Square

Error ($RMSE = \sqrt{\frac{1}{N} \sum_{i=1}^N (s_i - o_i)^2}$); NRMSE is the Normalized Root Mean Square Error

($NRMSE = \frac{\sqrt{\frac{1}{N} \sum_{i=1}^N (s_i - o_i)^2}}{\bar{o}_i}$); Bias is the mean bias ($Bias = \frac{1}{N} \sum_{i=1}^N (s_i - o_i)$); NBias is the

normalized mean bias ($NBias = \frac{\frac{1}{N} \sum_{i=1}^N (s_i - o_i)}{\bar{o}_i}$); R² is the coefficient of determination

($R^2 = 1 - \frac{\sum_{i=1}^N (s_i - o_i)^2}{\sum_{i=1}^N (s_i - \bar{o}_i)^2}$). H0: Slope = 1 denotes the null hypothesis of slope equals 1 with no

constraints on the intercept. H0: Intercept = 0 denotes the null hypothesis of intercept equals 0 with no constraints on the slope. H0: Slope = 1 and intercept =0 denotes the null hypothesis of slope and intercept equal 1 and 0, respectively. The F-test is applied to investigate whether the regression coefficients equal the given parameters significantly or not (slope equals 1 and intercept equals 0). If the p-value from the F-test is less than the given significance level ($\alpha=0.05$) and the F-value from the F-test is larger than the critical value in the F distribution with given significance level ($\alpha=0.05$), the null hypothesis can be rejected, denoting that the regression coefficients equal the given parameters insignificantly. If we fail to reject the null hypothesis, we accept it by default, denoting that the regression coefficients equal the given parameters significantly (denoted with * in Table S5). $F_{(\alpha, df_1, df_2)}$ denotes the critical value in the F distribution with a given significance level ($\alpha=0.05$) with denominator degrees of freedom (df_1) and numerator degrees of freedom (df_2), which could be obtained from F distribution tables.

Fig. S15 The performance of *ecosys* model with maize cropping systems at three AmeriFlux sites (US-Ne1, Ne2, and Ne3) during the period from 2001-2012 and 12 sites with a dramatic rainfall gradient across Nebraska during the period from 2010-2019. The colorbar showed the normalized gaussian kernel density estimation of the scatters. Black dashed lines indicated the 1-to-1 relationship. The red line was the regression line with the slope and intercept. The probability density function of the maize yields at the 12 sites was the gaussian kernel density estimation.

Table S5: Please add the slope and intercept values for the regressions and the results of statistical tests on equality of slope with unity and equality of intercept with zero.

Reply: We have added the slope and intercept values of the regressions of daily and monthly GPP, ET, LAI, and crop yield in the Table S5. We have also added the statistical results from the F-tests of three null hypotheses (slope=1, intercept=0, slope=1 and intercept=0) for daily and monthly GPP, ET, LAI, and yield in the Table S5.

The F-test is applied to investigate whether the regression coefficients equal the given parameters significantly or not (slope equals 1 and intercept equals 0). If the p-value from the F-test is less than the given significance level ($\alpha=0.05$) and the F-value from the F-test is larger than the critical value in the F distribution with given significance level ($\alpha=0.05$), the null hypothesis can be rejected, denoting that the regression coefficients equal the given parameters insignificantly. If we fail to reject the null hypothesis, we accept it by default, denoting that the regression coefficients equal the given parameters significantly (denoted with * in Table S5). $F_{(\alpha,df_1,df_2)}$ denotes the critical value in the F distribution with a given significance level ($\alpha=0.05$) with denominator degrees of freedom (df_1) and numerator degrees of freedom (df_2), which could be obtained from F distribution tables. Please see more details in Table S5, which has been shown earlier in this response letter.

Figure S12 and similar figures: A major problem with these relationships is that they include all days of the year whereas the corn growing season is the pertinent time scale for comparison between modeled and measured data. Using the entire year tends to inflate the degree of agreement because of the many small values during the off season.

Reply: Thank you for the comments. We only kept the data (GPP, NEE, LE, and LAI) during the growing season (May to October) in the period 2001-2012 in Figure S12-S14. Please see the updated figures with updated statistical results below:

Fig. S12 Performance of *ecosys* model on daily GPP, NEE, LE, and LAI with the statistical results (RMSE, Bias, and R^2) for continuous maize cropping systems during the growing season (May to October) in the periods 2001-2012 at US-Ne1.

Fig. S13 Performance of *ecosys* model on daily GPP, NEE, LE, and LAI with the statistical results (RMSE, Bias, and R^2) for maize (yellow regions) and soybean (green regions) cropping systems during the growing seasons (May to October) in the period 2001-2012 at US-Ne2.

Fig. S14 Performance of *ecosys* model on daily GPP, NEE, LE, and LAI with the statistical results (RMSE, Bias, and R²) for maize (yellow regions) and soybean (green regions) rotation cropping systems during the growing seasons (May to October) in the period 2001-2012 at US-Ne3.

In the rebuttal, the authors write, “(1) We fully agree that the traditional MAD irrigation scheme could be optimized through many settings, such as the growing stage-specific MAD. As a matter of fact, the MAD threshold level presented in the current study was already optimized by maximizing the economic profits as the benchmarks. We admit that the constant MAD level is a simple approach for testing. Furthermore, since the optimal MAD threshold is not the focus of this study, the major goal of this study is to incorporate the acknowledged co-regulation pattern in plant physiology into irrigation management, and to assess its efficacy for irrigation guidance. The proposed supply demand dynamics (SDD)’s threshold level varied with VPD, and we compared its performance with the already optimized MAD level to investigate the benefits of incorporating the co-regulation patterns into irrigation.” Adding a version of this text to the discussion could further elucidate the intent and limitations of the study with regard to sophisticated rather than simplistic use of MAD based irrigation scheduling. As it stands these ideas are ignored in the manuscript.

Reply: Thank you for the suggestion. We have added some discussion on this topic in the revised manuscript. Please see the revised texts in the manuscript below:

Line 292-297: Our modeling results indicated that the new SDD scheme may make significant contributions to water sustainability, compared with the existing highly efficient MAD scheme with optimized constant threshold, under both current and future climate conditions. It should be noted

that the traditional MAD irrigation scheme could be further optimized through many other settings, such as the growing stage-specific thresholds, which is beyond the scope of this research.

Also in the rebuttal, the authors write, “As you mentioned, maize in the Texas Panhandle could maintain high yield under hot, semiarid to arid climate due to intensive irrigation. There are several reasons for discussion: (1) the seeds used in Texas may be adapted to the hot, semiarid to arid climate, while the *ecosys* simulations under climate change used the same phenology parameters without acclimation from current climate, i.e. no genetic advances considered; (2) the existed cooling effects due to intensive irrigation in the field experiments was not considered in the *ecosys* model simulations under climate change; (3) the irrigation scheme with MAD under climate change was the same as that under current climate, i.e. no irrigation practice improvements considered. Although we could apply the acclimation of parameters in *ecosys* under climate change to maintain the high maize yields under climate change, we think that the setting of the *ecosys* model without the acclimation of parameters under climate change is more reasonable. The reason is that we aimed to investigate the impacts of environmental variables under climate change, rather than other technological advances.” Adding some version of this to the discussion would do a lot to set the results of the study in the broader context of current and future scenarios, and it would acknowledge that high yields are already obtained with irrigation in hot, arid climates, which implies that yield reductions simulated with *ecosys* are not the only possible outcome.

Reply: Thank you for the suggestions. We have added some discussion on this topic in the revised manuscript. Please see the revised texts in the manuscript below:

Line 243-248: If we do not consider the technological advances³³, such as seeds, irrigation, and fertilizer improvements, on crop yields, i.e. no technology yield trend, irrigation water use of MAD irrigation scheme under RCP-8.5 scenario (2058-2076) increased by 16.1%, accompanied with significant reductions in crop yield (-24.5%), economic profit (-54.1%), and irrigation water productivity (-29.2%), when compared with the current climate condition (2001-2019) (Fig. S11).

Line 447-455: *ecosys* model simulation under current climate (2001-2019) was extended through three additional 19-year cycles from 1 January 2020 to 31 December 2076 under the RCP-8.5 scenario using the incremental change scheme without the acclimation of parameters in *ecosys*, and the fourth cycle 2058-2076 was selected as the future climate to investigate the climate change effects. It should be noted that the acclimation of parameters in *ecosys* could be applied to maintain high maize yields with increased air temperature under climate change, like the high maize yields under hot, semiarid to arid climate with intensive irrigation in the Texas Panhandle. As we aimed to investigate the impacts of environmental variables to maize under climate change, no acclimation of parameters in *ecosys* is suggested under climate change.

Reviewer #3 (Remarks to the Author):

My comments were addressed in the revision and I would recommend this manuscript to be accepted for publishing.

Reply: Thank you for the positive feedback.

One suggestion for the authors is with their choice of empirical models. As the authors are well aware of, the simple empirical function used to determine VPD and SWC impacts cannot separate the two factors. Although it might be accepted to use for now, under climate change, the fitted parameter values could become invalid. For example, future rainfall may remain the same but VPD could increase due to increase of temperature. The crops then may choose to use water differently and thus should use a different parameter value. This is not a key point of the study so no need to change here but maybe something to consider in the future.

Reply: Thank you for the comments. We have added some discussion in the revised manuscript about the parameters under current scenario and climate change. Please see the revised texts in the manuscript below:

Line 490-493: It needs to be noted that the fitted parameters under current climate (2001-2019) should be updated under RCP-8.5 scenario, as maize may respond differently under climate change due to increased air temperature and VPD.

REVIEWERS' COMMENTS

Reviewer #2 (Remarks to the Author):

I thank the authors for the revisions to their work, which I believe have substantially improved the report.

Dr Kasey Bolles, Associate Editor

Your manuscript entitled "Sustainable irrigation based on co-regulation of soil water supply and atmospheric evaporative demand" has now been seen again by our referees, whose comments appear below. In light of their advice I am delighted to say that we are happy, in principle, to publish a suitably revised version in Nature Communications under the open access CC BY license (Creative Commons Attribution 4.0 International License).

We therefore invite you to revise your paper one last time to address the remaining concerns of our reviewers and our editorial requests in the attached document(s). At the same time we ask that you edit your manuscript to comply with our policies and formatting requirements and to maximise the accessibility and therefore the impact of your work.

Reply: Thank you for handling the review. We have revised the manuscript following the comments and suggestions from the editorial requests. Please see the detailed reply in the Author_Checklist file and revised manuscript.

REVIEWERS' COMMENTS

Reviewer #2 (Remarks to the Author):

I thank the authors for the revisions to their work, which I believe have substantially improved the report.

Reply: Thank you for your approval.